

# Linking hydraulic traits to tropical forest function in a size-structured and trait-driven model (TFS v.1-Hydro)

Bradley O. Christoffersen[1,2], Manuel Gloor[3], Sophie Fauset[3], Nikolaos M. Fyllas[4], David R.
Galbraith[3], Timothy R. Baker[3], Lucy Rowland[2], Rosie A. Fisher[5], Oliver J. Binks[2], Sanna A.
Sevanto[1], Chonggang Xu[1], Steven Jansen[6], Brendan Choat[7], Maurizio Mencuccini[2,8], Nate G.
McDowell[1] and Patrick Meir[2,9]

*Correspondence to*: bradley@lanl.gov, ph: +1-505-665-9118

1. Earth and Environmental Sciences, Los Alamos National Laboratory, Los Alamos, New
Mexico, USA
2. School of GeoSciences, University of Edinburgh, Edinburgh, United Kingdom
3. School of Geography, University of Leeds, Leeds, United Kingdom
4. Department of Ecology and Systematics, University of Athens, Athens, Greece
5. Climate & Global Dynamics, National Center for Atmospheric Research, Boulder, Colorado,
USA
6. Institute of Systematic Botany and Ecology, Ulm University, Ulm, Germany
7. University of Western Sydney, Hawkesbury Institute for the Environment, Richmond, New
South Wales 2753, Australia
8. ICREA at CREAF, Cerdanyola del Vallès, Barcelona 08193, Spain
9. Research School of Biology, Australian National University, Canberra, Australia

**Abstract.** Forest ecosystem models based on heuristic water stress functions poorly predict
tropical forest response to drought because they do not capture the diversity of hydraulic traits
(including variation in tree size) observed in tropical forests. We developed a Richards'
equation-based model of plant hydraulics in which all parameters of its constitutive equations are
biologically-interpretable and measureable plant hydraulic traits (e.g., turgor loss point $\pi_{tlp}$, bulk
elastic modulus $\varepsilon$, hydraulic capacitance $C_{ft}$, xylem hydraulic conductivity $k_{s,max}$, water
potential at 50% loss of conductivity for both xylem ($P_{50,x}$) and stomata ($P_{50,gs}$), and the
leaf:sapwood area ratio $A_l : A_s$). We embedded this plant hydraulics model within a forest
simulator (TFS) that modeled individual tree light environments and their upper boundary
condition (transpiration) as well as provided a means for parameterizing individual variation in
hydraulic traits. We synthesized literature and existing databases to parameterize all hydraulic
traits as a function of stem and leaf traits wood density (*WD*), leaf mass per area (*LMA*) and
photosynthetic capacity (*A*max) and evaluated the coupled model's (TFS-Hydro) predictions



against diurnal and seasonal variability in stem and leaf water potential as well as stand-scaled sap flux.

Our hydraulic trait synthesis revealed coordination among leaf and xylem hydraulic traits and statistically significant relationships of most hydraulic traits with more easily measured plant

5     traits. Using the most informative empirical trait-trait relationships derived from this synthesis, the TFS-Hydro model parameterization is capable of representing patterns of coordination and trade-offs in hydraulic traits. TFS-Hydro successfully captured individual variation in leaf and stem water potential due to increasing tree size and light environment, with model representation of hydraulic architecture and plant traits exerting primary and secondary controls, respectively,

10     on the fidelity of model predictions. The plant hydraulics model made substantial improvements to simulations of total ecosystem transpiration under control conditions, but the absence of a vertically stratified soil hydrology model precluded improvements to the simulation of drought response. Remaining uncertainties and limitations of the trait paradigm for plant hydraulics modeling are highlighted.

15     Key words: vegetation modeling, plant hydraulics, size scaling, xylem embolism, capacitance, turgor loss point, water stress, soil-plant-atmosphere continuum



## 1 Introduction

Tropical forests harbor great biodiversity (Myers et al., 2000; ter Steege et al., 2013) and play an important role in regulating regional and global climate (Gash and Nobre, 1997; Silva Dias et al., 2002). However, climate change is inducing changes to the hydrological regime of tropical

forests (Feng et al., 2013; Fu et al., 2013; Gloor et al., 2013), with some consensus for a projected increase in drought frequency over the coming century via an intensification of precipitation seasonality (Boisier et al., 2015; Joetzjer et al., 2013) or an increase in El Niño events (Cai et al., 2014), even as the directional change of total precipitation remains highly uncertain (IPCC, 2007). Therefore, because of their intrinsic value and strong coupling to the

regional and global climate system, it is of paramount importance to have a predictive capability of tropical forest response to changes in water availability (Meir et al., 2015; Oliveira et al., 2014).

Evaluations of forest ecosystem models in the species-rich tropics indicate a poor predictive ability of these models to simulate tropical forest response to drought when plant hydraulics are

not represented (Galbraith et al., 2010; Joetzjer et al., 2014; Powell et al., 2013; Rowland et al., 2015b). This is due in large part to water stress being parameterized by a single heuristic "beta" function that downscales photosynthesis or stomatal conductance based solely upon soil water content or potential (discussed in Egea et al., 2011; Feddes et al., 1978; Verhoef and Egea, 2014; Xu et al., 2013). Additionally, water stress in many models includes relatively few plant

functional type (PFT)-dependent parameters. In contrast, explicit representation of plant hydraulics coupled with site-specific parameterization enabled high-fidelity simulation of tropical forest response to moisture (Fisher et al., 2007; Fisher et al., 2006).

There are a range of approaches for modeling plant hydrodynamics that can be understood in terms of how they represent hydraulic effects on stomata, xylem conductivity and its reduction

with increasing tension, and tissue water content or hydraulic capacitance. Some models represent the continuum with resistors and optional capacitors but which do not simulate within-tree reductions in hydraulic conductance (Alton et al., 2009; Bonan et al., 2014; Jarvis et al., 1981; Ogée et al., 2003; Williams et al., 1996). This approach has proven useful for modeling the effects of seasonal and experimental drought in tropical forests (Fisher et al., 2007; Fisher et

al., 2006; Williams et al., 1998). Improvements upon this approach involve implementing



variable xylem conductivity with water potential (the "percent loss of conductivity", or PLC, relationship) (Domec et al., 2012; Duursma and Medlyn, 2012; Hickler et al., 2006; Williams et al., 2001). Other models treat the plant continuum as a porous medium with constitutive equations defining water retention properties (the relationship between water potential and water

content) and xylem PLC, using Darcy's law to incorporate fluxes within the Richards' mass balance equation (Bohrer et al., 2005; Edwards et al., 1986; Mackay et al., 2015; Sperry et al., 1998)(Arbogast et al., 1993). This approach is attractive because it makes no assumptions about where within the continuum hydraulic transport is limiting, and also possesses the relevant hydraulic parameters for which large variation has been documented in the tropical literature.

An additional challenge for modeling tropical vegetation responses to climate lies in addressing the large and growing body of knowledge about the diversity of drought sensitivity and hydraulic traits within tropical forests. It has long been recognized that the enormous biodiversity of tropical forests mirrors a comparative functional diversity and associated ecological strategies for dealing with environmental variation (Corner, 1949; Hallé et al., 1978; Leigh Jr, 1999). This

diversity in tropical forests is prominently manifested in plant hydraulic traits such as cell turgor properties and capacitance, xylem conductivity, and vulnerability to embolism (Borchert, 1994; Lopez et al., 2005; Tobin et al., 1999; Zhu et al., 2013), which are implicated in the variation in species-level survival in both natural and experimentally-induced droughts (Engelbrecht and Kursar, 2003; Kursar et al., 2009; Meir et al., 2015; Moser et al., 2014; Nakagawa et al., 2000).

Survival in these studies was not only taxon-specific, but also stratified by tree size, usually with larger trees demonstrating significantly higher vulnerability to drought (da Costa et al., 2010; Nakagawa et al., 2000; Nepstad et al., 2007; Rowland et al., 2015a), a trend demonstrated to be pantropical in scope (Bennett et al., 2015; Phillips et al., 2010). Such a trend highlights that, in addition to hydraulic functional traits, the increasing hydraulic path length associated with tall

trees (and perhaps increased radiation loads) is an equally important determinant of drought sensitivity (McDowell and Allen, 2015). Differential drought sensitivity, in turn, is known to explicitly link to species distributions (Baltzer et al., 2008; Choat et al., 2007; Condit et al., 2013; Engelbrecht et al., 2007). Thus the implication of increases in drought frequency for tropical forests are shifts in growth form (Phillips et al., 2002; Schnitzer and Bongers, 2011), species

composition (Fauset et al., 2012; Meir et al., 2015) as well as size distribution. The implication for ecosystem models is that under-representation of diversity in functional traits and tree size in



tropical forests is hampering efforts to make accurate projections of tropical forest response to climate.

Because a complete representation of trait diversity in models is neither tractable nor desired, the challenge then becomes to represent the key dominant axes, or modes, of trait variation within a

plant hydraulics model. To date, syntheses of hydraulic traits have typically focused on a limited set of traits in isolation (Anderegg, 2015; Bartlett et al., 2012; Choat et al., 2012; Nardini et al., 2014) but see Mencuccini et al. (2015). The model we present necessitates a synthesis spanning a large range of hydraulic traits. It introduces multiple parameters to the forest model (Table 2); if all of these parameters varied independently of each other, representing trait diversity would

be intractable. If, however, these hydraulic traits coordinate with other aspects of plant economics, such as the leaf economics spectrum (Wright et al., 2004) or the stem economics spectrum (Chave et al., 2009), then we can represent the realistic plant water use strategies using a relative low number of axes of trait variation in our model. Arguments for and evidence of both strong (Mendez-Alonzo et al., 2012; Reich, 2014) and weak to nonexistent (Baraloto et al.,

2010) coupling between stem and leaf economics spectra in tropical forests have been put forward; therefore given the lack of consistent evidence for their coordination, we make no *a priori* assumptions in this regard, and ask how hydraulic traits coordinate with either leaf or stem economics traits either in conjunction or separately. An additional (not necessarily independent) constraint on trait variation might emerge via trade-offs among certain hydraulic traits, such as

the trade-off between xylem efficiency (hydraulic conductivity) and safety (embolism resistance) (Gleason et al., 2016; Sperry et al., 2008; Tyree et al., 1994) or the trade-off between drought avoidance (e.g., via leaf shedding) and tolerance (embolism resistance), mediated by sapwood capacitance (Borchert and Pockman, 2005; Meinzer et al., 2008; Pineda-Garcia et al., 2013). Here we investigate the possibility of predicting hydraulic traits from more commonly measured

leaf and stem traits. These quantitative relationships are analagous to pedotransfer functions that soil physicists have used to derive soil hydraulic properties from more easily measured properties such as soil texture; hence we refer to quantitative trait-trait relationships as 'ecotransfer' functions. A key aspect of this study, however, is to address how well such ecotransfer functions capture coordination and trade-offs among hydraulic traits wherever they

exist.



The objective of this study is to develop a plant hydraulics model rooted in established theory of plant physiology and allometry, and to implement this model within a broader model framework capable of parameterizing individual-level variation in plant hydraulic traits along key axes of hydraulic trait variation. Specifically, we first present a porous media representation of tree

water use that both resolves water potential gradients throughout the soil-root-stem-canopy continuum and uses first principles to capture the effect of increasing tree size on whole-tree hydraulic properties. Then, this hydraulic framework is incorporated into a size-structured and trait-driven model that allows for the representation of individual-level variation in leaf, stem and root hydraulic traits and tree size. To parameterize the model, we conduct an empirical synthesis

that both identifies how hydraulic traits are coordinated with more readily measured leaf and stem traits and assesses the evidence for and capability of the new model framework to represent coordination and trade-offs among hydraulic traits. We then demonstrate the model's capability to simulate different individual-level functional responses to environmental variation (either water or light), arising from hydraulic limitations imposed by size or hydraulic traits. Finally, we

evaluate the new model in terms of its ability to capture diurnal dynamics of water potential, individual-level variation in leaf water status, and the observed seasonal dynamics of whole-ecosystem water use under control and experimentally-induced soil water deficit.

## 2 Model Description: Plant Hydraulics and Host Individual Trait-Driven Forest Models

The plant hydraulics model developed here is integrated into a "host" individual tree trait-driven

forest model (TFS) (i.e., it simulates every tree in a stand > 10 cm DBH and each tree can possess a unique suite of traits). TFS supplies plant traits ($WD$, $LMA$, $N_L$, $P_L$), tree size, and canopy biophysics to the new plant hydraulics module (Fig. 1). Through synthesis of literature and existing databases, we develop empirical equations to estimate, for each simulated tree, tissue-level hydraulic traits from the input plant traits and, combined with tree size, whole-tree

hydraulic properties. The new plant hydrodynamics scheme passes to TFS a nondimensional multiplier ($FMC_{gs}$) (0,1] for downregulating stomatal conductance which is based on leaf water potential at the previous timestep. The TFS model uses $FMC_{gs}$ to estimate individual tree canopy-level transpiration fluxes at the current timestep, which it then passes back to the hydrodynamic model, which computes changes in water potentials and water contents

throughout the soil-plant continuum. Below we describe the components of the plant hydraulics



model (Section 2.1), and at the end of each subsection highlight the key hydraulic traits for which we seek to understand trait trade-offs and coordination via our empirical synthesis. Section 2.2 gives a description of our host trait-driven model within which the hydraulics scheme is embedded. The Supplement S1 gives a full technical description of the hydrodynamics

portion of our tree hydraulics model as it is implemented within TFS, and Figure S1.1 can be referred to as our hydraulics model schematic.

## 2.1 Plant Hydraulics Model

### 2.1.1 Overview

In this section we highlight the important developments we made to the model developed by J. S.

Sperry and described in Sperry et al. (1998; hereafter S98). S98 consists of a discretization of the soil-plant continuum as a series of water storage compartments with defined heights, volumes, conducting areas, water retention and conductivity properties, connected by elements with defined path lengths and conductances. Trees are divided into the four porous medium types of leaf, stem, transporting root and absorbing root, with the stem being divided into a

variable number of compartments and all other types consisting of a single compartment (Fig S1.1). The soil is radially discretized with a variable number of compartments, or cylindrical 'shells' around a characteristic absorbing root (the rhizosphere), and soil hydraulic properties are assumed constant across these compartments and across trees within a given soil type. The present scheme does not consider the vertical distribution of soil water or roots; this is left as a

component for subsequent model development.

With reference to the sections below, we modified S98 in three important ways by: 1) incorporating continuous functions of varying water potential with water content in tree stem and leaf tissues (i.e., non-constant capacitance; Section 2.1.2); 2) using first principles to scale length-independent hydraulic conductivities to hydraulic conductance over the entire stem

(Section 2.1.4); and 3) linking the hydrodynamics to stomatal conductance (Section 2.1.5), thus creating a fully prognostic and dynamic model of leaf water potential. The end result is a plant hydraulics model where the parameters of the constitutive equations (Eqns 1-5) are biologically interpretable and measureable plant hydraulic traits. We summarize these key plant-specific developments in the sections below.





With reference to the sections of the Technical Description in Supplement S1, a brief summary of the other components of the model is as follows: We used the van Genuchten (1980) and Mualem (1978) formulations for the soil water characteristic (Section 3.1.2) and unsaturated hydraulic conductivity curves (Section 3.2.2), respectively, which have been the basis for

pedotransfer functions developed for tropical soils (Hodnett & Tomasella 2002). Hydraulic compartment heights and sizes (volumes and lengths, in the case of absorbing roots) are linked to the tree allometry of TFS (Section 4). The numerical solution uses a non-iterative mass-based solution to the Richards' equation, thereby doing away with the need for an iterative (potentially computationally intensive) Newton-Raphson scheme (Section 5). Following the approach of

Siqueira et al. (2008), we separated out solutions for water movement in soil due to root water uptake (given in sections above) from those for vertical water movement due to infiltration and drainage, which were based on the mean water content across all rhizosphere shells (Section 6).

## 2.1.2 Tissue water relations

We used pressure-volume (PV) theory (Bartlett et al., 2012; Tyree and Hammel, 1972; Tyree and

Yang, 1990) to describe the relation between total water potential ($\psi_{tot}$, MPa) and relative water content ($RWC$, g $H_2O$ g$^{-1}$ $H_2O$ at saturation) in plant tissue. $\psi_{tot}$ is the sum of two components: solute potential $\psi_{sol}$ (MPa) which is negative due to the presence of solutes in living cells, and pressure potential $\psi_p$ (MPa) which is >= 0 due to cell wall turgor (but see Ding et al., 2014). PV theory assumes that xylem water (outside of living cells; the apoplast) is in equilibrium with

water in living cells (the symplast). It is usually applied to leaves, but can also apply to sapwood (Chapotin et al., 2006; Meinzer et al., 2008; Scholz et al., 2007) and we apply it here to all plant tissue. In addition, sapwood also stores capillary water (Tyree and Yang 1990), originating from void spaces and embolized conduits and which is not held under tension. Using PV theory as a formulation for tissue water relations in the model offers four main advantages: 1) it represents

an explicit expression for the relation between water content and water potential amenable to numerical solution techniques for porous media water flow; 2) its parameters have a mechanistic interpretation; 3) it avoids the use of a constant capacitance with water potential (and thereby potentially infinite water supply); and 4) it allows for species differences in physiological parameters which may govern differential drought tolerance under extreme drought beyond the

turgor loss point.





The general form for both leaf and sapwood PV curves describes three successive dehydration phases as:

$$\psi_{tot} = \begin{cases} \psi_0 - m_{cap}(1 - RWC) & RWC_{ft} \leq RWC \leq 1 \\ \psi_{sol}(RWC) + \psi_p(RWC) & RWC_{tlp} \leq RWC < RWC_{ft} \\ \psi_{sol}(RWC) & RWC_r \leq RWC < RWC_{tlp} \end{cases} \qquad (1)$$

The first phase represents drainage of capillary water and is assumed linear, characterized by a slope ($m_{cap}$) and a saturated water potential ($\psi_0$). This is followed by an elastic drainage region where both solute ($\psi_{sol}$) and pressure ($\psi_p$) potential are changing from the point at which elastic drainage begins ($RWC_{ft}$) up to the turgor loss point ($RWC_{tlp}$ and corresponding $\psi = \pi_{tlp}$), at which $\psi_p = 0$. In the final region, symplastic (cell water) and xylem water from embolized conduits is expressed up to the point at which $\psi_{ij}$ approaches $-\infty$ ($RWC_r$). $RWC_r$ is often referred to as the apoplastic fraction (Bartlett *et al.* 2012), but in light of the considerable amount of water released when vessels embolize in stems (Holtta et al., 2009; Tyree et al., 1991), in this context $RWC_r$ is best termed the residual fraction. Leaf PV curves as traditionally interpreted are a special case of Eqn (1) in which there is no capillary water ($RWC_{ft} = 1$). $\psi_{sol}$ and $\psi_p$ are respectively given by

$$\psi_{sol}(RWC) = \psi_{sol}(RWC^*) = \frac{-|\pi_o|}{RWC^*} \qquad (2)$$

$$\psi_p(RWC) = \psi_p(R^*) = |\pi_o| - \varepsilon R^* \qquad (3)$$

where $\pi_o$ (MPa) and $\varepsilon$ (MPa) are osmotic potential at full turgor and bulk elastic modulus, respectively, and $RWC^* = (RWC - RWC_r)/(RWC_{ft} - RWC_r)$ and $R^* = (RWC_{ft} - RWC)/(RWC_{ft} - RWC_r)$ are transformations representing $RWC$ and $R$ (relative water deficit; $1 - RWC$) of symplastic (cell) water only (Bartlett et al. 2012). The absolute mass ($W$; kg) of water in tissue is given by $W = \rho_w \theta_{sat} RWC$, where $\theta_{sat}$ (m$^3$ m$^{-3}$) is the maximum water content on a per volume basis (or porosity) and $\rho_w$ (kg m$^{-3}$) is the density of water. The constitutive equations used in the model are in terms of volumetric water content ($\theta$; m$^3$ H$_2$O m$^{-3}$ plant tissue), achievable by using the transformations given above (see Supplement S1). Volumetric capacitance ($C$; kg m$^{-3}$ MPa$^{-1}$) is defined at any point along the three regions as ($\rho_w \theta_{sat} \frac{dRWC}{d\psi}$), which for sapwood is highest in the capillary region (~200-400 kg m$^{-3}$ MPa$^{-1}$),





intermediate in the elastic region (20-200 kg m$^{-3}$ MPa$^{-1}$) (Tyree and Yang 1990), and after an initial increase beyond the turgor loss point, in theory will approach zero as $RWC \rightarrow RWC_r$. Henceforth and in all figures, we report $C$ at full turgor ($C_{ft}$), defined as the change in water mass per unit volume per unit change in water potential over the region where cell turgor ($\psi_p$) $\geq$

$0$ ($C_{ft} \equiv \rho_w \theta_{sat} \left. \frac{\Delta RWC}{\Delta \psi} \right|_{RWC=1}^{RWC=RWC_{tlp}}$). For these hydraulic traits and all others which follow, we denote specific reference to leaf and xylem (sapwood) tissue with the subscripts $l$ and $x$, respectively. Key hydraulic PV traits which we seek to determine as functions of more commonly measured plant traits are $\varepsilon$, $\pi_o$, $RWC_{tlp}$, $C_{ft}$, and $RWC_r$.

### 2.1.3 Embolism vulnerability

We use the inverse polynomial of (Manzoni et al., 2013a) for the xylem vulnerability curve, termed here the 'fraction of maximum conductivity' for xylem ($FMC_x$):

$$FMC_x(\psi_x) = \left[1 + \left(\frac{\psi_x}{P_{50,x}}\right)^{a_x}\right]^{-1} \qquad (4)$$

where $P_{50,x}$ is the water potential at which 50% of maximum conductance is lost and $a_x$ is a shape parameter (-). $FMC_x$ is defined at compartment nodes and is a nondimensional multiplier bounded on (0,1] limiting the maximum xylem conductance ($K_{max,i}$; kg s-1 MPa-1; FMC = 1 and

0 indicate no and complete xylem embolism, respectively) between any two compartments $i$ and $i+1$ (Section 1 of the Supplement outlines how $FMC_x$, defined at compartment nodes, limits $K_{max,i}$, which is defined at compartment boundaries). Critically, $K_{max,i}$ derives from plant hydraulic architecture and maximum xylem-specific conductivity ($k_{s,max,x}$). Because $k_{s,max,x}$ relates to the maximum rate at which water can be transported through the xylem, and $P_{50,x}$

quantifies the xylem water potential at which half of this transport capacity is lost, these can be thought as representing xylem efficiency and safety, respectively. "Safety" refers to a property of the xylem alone and is not to be confused with the hydraulic safety margin (HSM), which is field-observed or modeled minimum leaf water potential *relative to* $P_{50,x}$. As with PV traits, we seek to determine to what extent xylem efficiency and safety trade off with each other and/or

covary with other plant traits.





### 2.1.4 Scaling conductance with tree size

Tree size exhibits a first-order control over much variation in whole-plant hydraulic conductance (Sperry et al., 2008); therefore, capturing this effect in any hydraulics scheme is critical for modeling size-structured communities such as forests. The size effect is driven by an increase in

hydraulic path length with tree size (Mencuccini, 2002). Trees have evolved two main mechanisms to mitigate this negative effect of height: 1) the near-universal tendency for xylem conduits to increase in diameter within trees from stem tips to trunk base (referred to as xylem taper in the opposite direction) (Meinzer et al., 2010; Mencuccini et al., 2007; Olson et al., 2014; Olson and Rosell, 2013; Petit and Anfodillo, 2011) which increases whole-plant conductance

because of the Hagen-Poiseuille law and 2) the tendency for the leaf:sapwood area ratio ($A_l : A_s$) to decrease with increasing tree size, at least in temperate trees (McDowell et al., 2002), while some evidence for tropical forests has shown a tendency for $A_l : A_s$ to increase with tree size (Calvo-Alvarado et al., 2008). derive

We used these first principles in the model to derive a bottom-up estimate of whole-tree

maximum aboveground conductance ($K_{max,tree,ag}$; kg s$^{-1}$ MPa$^{-1}$) from xylem-specific conductivity ($k_{s,max,x}$; kg m$^{-1}$ s$^{-1}$ MPa$^{-1}$). We followed a three-step process, as follows: 1) $k_{s,max,x}$ was first standardized to the petiole ($k_{s,max,petiole}$; kg m$^{-1}$ s$^{-1}$ MPa$^{-1}$); 2) $k_{s,max,petiole}$ was then used to estimate whole-tree maximum aboveground conductance without xylem conduit taper ($K_{max,tree,notaper,ag}$; kg s$^{-1}$ MPa$^{-1}$); 3) $K_{max,tree,notaper,ag}$ was multiplied by a

nondimensional factor representing the ratio of theoretical whole-tree aboveground conductance with taper to that without taper ($\chi_{tap:notap,ag}$) to derive $K_{max,tree,ag}$. The primary purpose of this component of the model is to capture the size dependency of $K_{max,tree,ag}$, without resorting to a pre-specified relationship between $K_{max,tree,ag}$ and tree size because we wanted to make explicit the key hydraulic architecture traits ($A_l : A_s, \chi_{tap:notap,ag}$) governing this trend, which

may covary across species independently or in conjunction with other plant trait spectra. A taper exponent of 1/3 amounts to $\chi_{tap:notap}$ in the range of 23-50 for trees of heights 10-30 m; thus the benefit of xylem taper for increasing total plant conductance itself increases with tree height. $\chi_{tap:notap}$ is assumed constant across individuals in this study, but parameterizing variation in $A_l : A_s$ across species is an outcome of this study. The full details of this approach in addition to



the treatment of the belowground component of tree conductance ($K_{max,tree,bg}$) are outlined in Section 2 of the Technical Description (Supplement S1).

### 2.1.5 Hydraulic impacts on stomatal conductance

Finally, we introduced to TFS a formulation for the impact of water supply on stomatal

conductance ($g_s$), in which reductions in leaf water potential ($\psi_l$) cause a decline in $g_s$ (sensu Jarvis, 1976). Similar to the approach for the loss of xylem conductivity with water potential (Eqn 4), this formulation followed a "fraction of maximum conductivity" ($FMC_{gs}$) curve:

$$g_s = g_{s,max} \cdot FMC_{gs} = g_{s,max} \left[ 1 + \left( \frac{\psi_l}{P_{50,gs}} \right)^{a_{gs}} \right]^{-1} \qquad (5)$$

where $P_{50,gs}$ and $a_{gs}$ respectively represent leaf water potential at 50% stomatal closure and the slope of the curve at $\psi_l = P_{50,gs}$ (Manzoni et al., 2013b; Mencuccini et al., 2015), and $g_{s,max}$ is

stomatal conductance in the absence of water supply limitation and comes from the host model's stomatal conductance scheme (for TFS, we used Medlyn et al. (2011); see Eqn A25 of Fyllas et al. 2014).

Theory and data suggest that stomata operate in such a way so as to prevent catastrophic xylem embolism (Sperry and Love, 2015); this implies that $P_{50,gs} > P_{50,x}$, which is supported by global

datasets (Klein (2014) Manzoni et al. (2013c)). We used a 1:1 relationship between $P_{50,gs}$ and $P_{20,x}$, which is supported by data from a tropical dry forest (Brodribb et al., 2003). This current approach ignores the continuum in hydraulic safety (Klein, 2014; Martinez-Vilalta et al., 2014; Skelton et al., 2015) and future work needs to identify how such a continuum maps onto other plant traits.

$FMC_{gs}$ [0,1], which is updated at every timestep and is used to down-regulate non-water stressed stomatal conductance, is similar in this regard to traditional soil-based "beta" approaches (Powell et al., 2013). $FMC_{gs}$ encapsulates the hydraulic limitation of each tree and is the only variable passed from the hydraulics module to the host model TFS. While this approach might seem simplistic relative to the comparative rigor of the hydrodynamics scheme, we note that 1) the

present function (Eqn A5) is based on leaf water potential, a new prognostic, non-steady-state metric of tree water status that integrates individual-level differences in hydraulic traits, size, and



microenvironment (e.g., light) in addition to soil water potential, 2) there is no consensus as to the exact mechanism (direct or indirect) by which stomata respond to leaf water potential – recent theoretical developments on stomatal function are largely restricted to the effects of temperature and humidity alone (Buckley, 2005; Nikinmaa et al., 2013; Peak and Mott, 2011),

and 3) this approach simplifies the interface between the hydraulics module and the host model.

## 2.2 Host model TFS

The Trait Forest Simulator (TFS) (Fyllas et al., 2014) – is an individual tree model parameterized from plot-level observed tree size distribution and distributions of the four plant traits wood density ($WD$), leaf mass per area ($LMA$), leaf nitrogen ($N_L$), and leaf phosphorus ($P_L$). These

traits are assigned to each individual in a way that preserves the underlying observed joint distribution of all four traits (be it tight or weak coupling among traits), which are then used to parameterize plant physiological parameters for each tree in a stand. $WD$ in TFS v.1 drives differential rates of stem respiration and stem C allocation, and $LMA$, $N_L$ and $P_L$ jointly drive differential rates of leaf light-saturated photosynthesis ($A_{max}$) via parameterization of

photosynthetic parameters $V_{cmax}$ and $J_{max}$ (Domingues et al. 2010), in addition to respiration and leaf and stem C allocation. The construction of this model was motivated by a need to diagnose large-scale patterns of productivity and turnover in tropical forests which, among other drivers, implicate a link to soil fertility and more proximally, plant traits reflective of fertility differences (Baker et al., 2004; Fyllas et al., 2009; Johnson et al., 2016; Patiño et al., 2012; Quesada et al.,

2012). Because measurements of stand-level variation in easily measured plant traits are much more abundant that hydraulic traits, but may reflect ecological axes of variation common to hydraulic traits, we sought to parameterize hydraulic trait variation via empirical relationships with the four TFS traits (see Table 1 and Section 3.1 below).

A second key component of TFS is the representation of stand structure and hence individual-

level variation in tree crown light environments, which is an additional key determinant of ecosystem productivity (Stark et al., 2012) and, as we will show, individual-level variation in tree hydraulic function and limitation. It uses the perfect plasticity approximation (PPA) of Purves et al. (2007) updated to represent multiple canopy layers typical of tropical forests (Bohlmann et al., 2012) as well as basic gap dynamics (Fauset et al., in revision) to estimate

individual tree light environments.





In sum, TFS couples individual-level variation in plant traits and light environments with a full canopy biophysics scheme for estimating photosynthesis and transpiration for each tree > 10cm DBH in a stand. Transpiration, in turn is the necessary boundary condition for our new plant hydrodynamics module (Fig. 1). For further details on TFS biophysics, size structure, trait

parameterization, and light competition, see Fyllas et al. (2014) and references therein.

## 3 Methods

### 3.1 Hydraulic trait syntheses for parameterization of plant hydraulics model

We synthesized the literature and existing trait databases and asked whether hydraulic traits associated with pressure-volume (PV) curves in leaves ($\pi_{o,l}$, $\varepsilon_l$, $RWC_{tlp,l}$, and $RWC_{r,l}$) and

sapwood ($\pi_{o,x}$, $\varepsilon_x$, $RWC_{tlp,x}$, and $RWC_{r,x}$), xylem conductivity per unit sapwood area ($k_{s,max,x}$) or per unit leaf area ($k_{l,max,x}$), xylem resistance to embolism ($P_{50,x}$) and associated shape parameter ($a_x$) as well as hydraulic architecture ($A_l$:$A_s$) were coordinated with the leaf and stem traits employed as input by the TFS model. All analyses were carried out within R (R core team 2015). In all figures, the following asterisk codes denote statistical significance at α levels of

0.05, 0.01, 0.001, and 0.0001, respectively: '\*', '\*\*', '\*\*\*', '\*\*\*\*'.

For leaf PV traits and *LMA*, we started with the Bartlett et al. (2012) database subsetted for tropical ecosystems (111 individuals across 14 studies) and added substantial additional data (44 individuals across 12 studies) through further literature review, and joined to this dataset species-averaged values of wood density using the Global Wood Density Database (Chave et al., 2009;

Zanne et al., 2009). For sapwood PV traits, we generated our own database (33 specimens across 9 studies) by digitizing data points of $RWC_x$ vs. $\psi_x$ conducted on trunk or branch tissue from published figures using WebPlotDigitizer (Rohatgi, 2014), and fit curves to these plots following established methods for extracting PV parameters (Bartlett et al., 2012; Sack and Pasquet-Kok; Tyree and Hammel, 1972). We found significant differences in the shape of curve

and magnitude of capacitance for sapwood PV curves conducted on small trunk cores versus cut whole terminal branches, with trunk cores yielding capacitances 2-10 times that of branches, suggesting substantial artefacts associated measurements conducted on trunk cores, an issue recently highlighted by Wolfe & Kursar (2015). Using measurements made on the same species using both methods, we developed a correction procedure and applied it to data suffering this



bias (Christoffersen, unpublished manuscript), and provide versions of figures without the correction applied in the Supplement S2.

For analyzing relations between xylem efficiency and safety and other traits, we started with the Xylem Functional Traits (XFT) database (Choat et al., 2012; Gleason et al., 2016), subsetted to

tropical regions and added data from 15 additional species not originally present (Mendez-Alonzo *et al.* 2013). We then explored relationships of these traits with *WD* and $A_{\max}$, by prioritizing WD reported in the original publications and adding species-average *WD* data from Zanne *et al.* (2009) where it was unreported. There is substantial debate surrounding appropriate methods for determining embolism vulnerability (Choat et al., 2010; Cochard et al., 2010; Sperry

et al., 2012; Wang et al., 2014) (see Brodribb et al. *in press* for a brief summary). Because there is some consensus that the "gold-standard" for $P_{50,x}$ measurements involves bench dehydration (DH) on long stem segments (Jansen et al., 2015), we explored trait relationships with and without other measurement methods for $P_{50,x}$.

For hydraulic architecture, we used the only study of which we were aware for tropical trees

(Calvo-Alvarado et al., 2008) that reports independent measurements of individual total tree leaf area and sapwood area across a wide range of tree sizes to explore the relationship of $A_l : A_s$ with tree height as well as with *LMA* and verified the latter relationship using a much broader dataset of branch-level measurements of $A_l : A_s$ conducted across the Amazon basin (Patiño et al., 2012). Via literature survey, we also compiled an independent, extensive dataset of $A_s$ as it varied with

tree diameter at breast height (*DBH*, cm) in tropical forests.

Finally, we standardized the representation of the "biome" category across all databases, and defined the following categories based on the location of the study (not the species' home range): tropical flooded forest (if identified as such in original publication), tropical wet forest (no months where evaporative demand exceeds precipitation), tropical moist forest (at least 1 month

where evaporative demand exceeds precipitation; predominantly evergreen), tropical dry forest (drought-deciduous phenologies make up a substantial fraction of species), tropical savanna (identified as such in original publication), subtropical forest (absolute latitude exceeded 23 degrees), tropical mangrove, and greenhouse. Syntheses of traits were limited to studies conducted on species growing in native (non-greenhouse) environments in tropical (thus

excluding subtropical observations) upland (non-flooded) habitats. In some cases, it was also





necessary to match hydraulic traits for the same species given as multiple (different) records within a database. As a rule, however, we did not average to the species level, and thus some variation included in regressions is intraspecific, albeit small.

### 3.2 Model setup

We used Princeton downscaled meteorological forcing data (Sheffield et al., 2006), observed soil textural properties, and observed tree size and trait distributions (for $WD$, $LMA$, $N_L$ and $P_L$) in the Caxiuana National Forest of east-central Brazilian Amazonia, a seasonal evergreen forest receiving 2100-2500 mm rainfall annually, and the site of an ongoing throughfall exclusion experiment (da Costa et al., 2010; Meir et al., 2008; Meir et al., 2015; Rowland et al., 2015a;

Rowland et al., 2015c), to parameterize and run our model for one year. While actual soil depth at the site is ~10 m, we set soil depth in simulations to 4m, which reflects the effective depth over which the majority of water extraction occurs, based on previous model validation with soil moisture data (Fisher et al., 2007). This setup applied to all model simulations in this paper -- both the idealized experiments and the simulations for comparison with field data.

### 3.3 Idealized model experiments

We conducted two sets of idealized experiments designed to explore the impacts of the new plant hydraulics scheme on simulated tree transpiration. In both cases, the TFS model was run with (TFS v.1-Hydro) and without (TFS v.1) the new plant hydraulics scheme. In the first experiment, all trees were assigned identical trait values using plot mean values of ($WD$), leaf

mass per area ($LMA$), leaf nitrogen ($N_L$), and leaf phosphorus ($P_L$). Plant hydraulic traits were identical across individuals in TFS v.1-Hydro. Having eliminated plant hydraulic traits as a source of variation among simulated tree transpiration, we used this model experiment to explore the impact of plant hydraulics on transpiration dynamics as a function of tree height and canopy position. In the second experiment set, all trees were assigned different trait values for one trait

only (or two, in the case of $N_L$ and $P_L$), by resampling the observed plot trait distributions following the algorithm of Taylor & Thompson (1986) (see Fyllas et al. 2014 for a full description of this resampling method), keeping all other traits at the plot mean value. We then explored the dynamics of transpiration, for canopy trees (full sunlight) only, of three large (50-55 cm DBH) trees spanning the range of assigned trait values. In both model experiments, we





examined the simulated mean diurnal cycle of transpiration characteristic of the wet and dry season, defined as months where monthly precipitation exceeded or fell below 100 mm, respectively. Months at season boundaries were excluded to maximize seasonal differences.

### 3.4 Model evaluation

We performed three sets of model evaluation which assessed the model's ability to capture observed variations in transpiration and water potential at the individual tree level and the stand level. In the first evaluation set, we evaluated diurnal cycles of leaf and stem water potential across multiple individual trees spanning a wide range of size and canopy position which were also subject to seasonal and experimental (TFE) differences in water availability. We used

output from our idealized model experiment in which all individuals possessed identical (plot-mean) trait values, and ran an additional simulation of this type in which precipitation was reduced by 50%, similar to that of other model-data comparison exercises at this site (Powell et al. 2013, Joetzjer et al. 2014). We matched simulated trees to those measured by Fisher et al. (2006) on the basis of size (closest simulated tree within +/- 2 cm DBH) across replicate days in

the month matching the month in which observations were made (May for wet season, November for dry season), taking the standard error of the mean as the model error. A total of eight individual-level comparisons (four individuals in the control plot and four in the TFE plot) were made.

  The second evaluation set was similar to the first, except we focused on midday simulated leaf

water potentials. We conducted four distinct sets of model simulations, in which we explored how successive changes to model parameterization impacted model fidelity with the observations. The first model simulation assigned trees identical trait values, equal to the plot mean, and made no accounting of xylem conduit taper. The second model simulation was like the first, except this time accounting for xylem conduit taper (see Section 2.1.4 and Supplement

S1 section 2.1.1). The third simulation was as the second but adjusted the canopy position of two of the simulated individuals. The fourth simulation was as the third but assigned the four simulated individuals species-specific *WD* values.

  In the final set, we evaluated seasonal differences in stand-scaled sap flux under control and experimentally-induced drought (throughfall exclusion experiment; TFE). We used model

setups in which all individuals possessed identical hydraulic traits and where all individuals





possessed different traits according to the plot observed trait distributions in order to assess ecosystem-level consequences of plant trait variation. Each of these setups was run as TFS v.1 and TFS v.1-Hydro. For comparison with observations, all combinations of these setups were run with both control precipitation and a 50% reduction. Simulated individual total transpiration

(kg s$^{-1}$) was summed across individuals, divided by the plot area, and accumulated for each month of the year. Observed stand-level transpiration was derived following the methods outlined in Fisher et al. (2007).

## 4 Results

### 4.1.1 Hydraulic trait synthesis

All resultant empirical equations relating traits *WD*, *LMA*, or $A_{max}$ to hydraulic traits are given in Table 2.

For leaf PV traits (Fig. 2), we found highly significant (p < 0.0001) but weak negative relationships of $\pi_{o,l}$ (leaf osmotic potential at full turgor) with both *WD* (wood density) and *LMA* (leaf mass per area) ($r^2 = 0.28$ and 0.09, respectively). Because *WD* and *LMA* were not

significantly correlated in this dataset (p = 0.48) we used them as independent predictors of $\pi_{o,l}$ in multiple regression and were able to explain a higher fraction of the variance in $\pi_{o,l}$ ($r^2 = 0.44$; see Table 2). Surprisingly, $\varepsilon_l$ (leaf bulk elastic modulus) demonstrated no significant relationship with *LMA* but was marginally correlated with *WD* (p < 0.05; $r^2 = 0.10$; Fig 2c, d). While we found no significant relationship of the residual fraction ($RWC_{r,l}$) with *LMA* or *WD*, we did find

a significant relationship of $RWC_{r,l}$ with $\varepsilon_{leaf}$ ($r^2 = 0.32$; p = 0.002; Fig 2e). These relationships imply that leaf capacitance increased with both decreasing *WD* and *LMA*, which was corroborated by independent datasets (Fig 2f).

For bias-corrected sapwood PV traits (Fig. 3), we found that *WD* well-represented a single axis of variability along which $\pi_{o,x}$, $\varepsilon_x$, and $R_{tlp,x} = 1 - RWC_{tlp,x}$ were organized ($r^2 = 0.31$ to 0.44),

with increasing *WD* leading to exponentially increasing $\varepsilon_x$ and decreasing $\pi_{o,x}$ and $R_{tlp,x}$. The result was exponentially declining sapwood capacitance ($C_{ft,x}$) with *WD* ($r^2 = 0.84$; Fig. 3d). The same trends were generally supported by the uncorrected data (Fig. S2.1), most importantly for $C_{ft,x}$.





On the full dataset of $P_{50,x}$, we found a significant relationship with *WD* (p < 0.0001), but the explained variance was low ($r^2$ = 0.08) due to a large number of high wood density species with low $P_{50,x}$ values.. However, when limiting the dataset to measurements made using the bench dehydration method (DH), many values of high (less negative) $P_{50,x}$ at intermediate to high WD

were now excluded (the majority of which were obtained using the air injection method) and a stronger negative relationship emerged ($r^2$ = 0.34) (Fig 4a). In addition, we found that the slope ($a_x$) of the xylem FMC curve decreased significantly (p < 0.0001) with increasingly negative $P_{50,x}$ (Fig 4b), suggesting that threshold responses of embolism to increasing xylem tension were more pronounced in individuals with less resistant xylem.

The relationship of xylem efficiency on a leaf area basis ($k_{l,max,x}$) with *WD*, while highly significant (p < 0.0001), explained only 7% of the variance (Fig 5a). However, when examined in relation to reported values of light-saturated leaf photosynthesis rates ($A_{max}$), a greater fraction of the variance was explained ($r^2$ = 0.29; p < 0.0001) (Fig 5b). Because $A_{max}$ and *WD* were uncorrelated (Fig 5d), we used them as independent predictors (as $A_{max}/WD$) of $k_{l,max,x}$,

which now explained 42% of the variance (Fig 5c). Similar trends were observed for xylem efficiency on a sapwood area basis ($k_{s,max,x}$; Supplement Fig. S2.2).

While the Calvo-Alvarado et al. (2008) study showed no significant common relationship of $A_l : A_s$ with *H* or *DBH* across all individuals studied, it identified increasing $A_l : A_s$ with *H within* 4 out of 5 individuals (Fig 6a). When we took species-average $A_l : A_s$ from the Calvo-Alvarado

(2008) study and plotted the interspecific differences as a function of species-average *LMA*, the individuals largely fell on the same regression line as that of Patiño *et al.* (2012) (Fig. 6b). Therefore, it appears that for tropical forests leaf-level traits sets the range of $A_l : A_s$ about which variation occurs in response to height and/or light availability (Fig 6c).

### 4.1.2 Plant hydraulics parameterization

Figure 7 shows the parameterization of all components of the plant hydraulics model (combining Table 2 and plant hydraulics model constitutive equations 1-4) as a function of the TFS model input traits and tree height (*H*). Panels a-c demonstrate the distribution of leaf PV, sapwood PV and xylem vulnerability curves for a characteristic TFS model simulation in which each tree is assigned four trait values (*WD*, *LMA*, $N_L$ and $P_L$) according to the observed joint distribution





(Taylor & Thompson 1986 resampling algorithm) of these four traits at the focal field site
Caxiuana National Forest, which are used to parameterize each tree's constitutive equations
(using Table 2), with each tree plotted as separate lines colored by trait values. This forest is
representative of others in the region (Baraloto et al., 2010), showing decoupled leaf and stem

economics (i.e., lack of any significant correlation between *WD* and any of the three leaf traits
used as input for TFS). As a result, the model input does not prescribe any trade-off between
xylem efficiency and safety (Fig 7c). Panels d-f present 'idealized' scenarios in which the full
range of possible hydraulic trait inputs are presented, based on tight coupling of leaf and stem
economics (specifically, between *WD* and *LMA* or *WD* and $A_{max}$) over a wide range of input

values. We represented the dual dependency of $A_l : A_s$ on both *LMA* and tree size (Fig 6) as an
$A_l : A_s \sim H$ relationship with an *LMA*-dependent slope (Fig 7g). For a prescribed leaf biomass
allometry function (as $\sim DBH^2 H$; Lescure et al. 1980), this $A_l : A_s$ formulation predicted *LMA* as
the driver of differences in sapwood area ($A_s$) ~ DBH allometry, spanning the range of values
given by our independent literature compilation across many different studies, though this

relationship warrants further validation.

### 4.1.3 Ability of hydraulics model parameterization to represent observed trade-offs and coordination among hydraulic traits

Based on the pantropical xylem functional traits dataset, evidence for a trade-off between xylem
efficiency (as $k_{s,max,x}$) and safety ($P_{50,x}$) was insignificant (p = 0.14 on log-transformed data),

which was also the case when limiting the database to the DH method (p = 0.82; Fig. 8a).
However, when limiting the dataset to tropical dry forests (not savanna), a significant trade-off
between xylem efficiency and safety emerged (p < 0.01; $r^2$ = 0.17) which was even stronger (p <
0.0001; $r^2$ = 0.44) when considering two tropical dry forest studies (see circled points in Fig 8a
(Markesteijn et al., 2011a; Markesteijn et al., 2011b; Mendez-Alonzo et al., 2012). Our plant

hydraulics model parameterization is able to capture both the lack of a safety-efficiency tradeoff,
which occurs when stem and leaf economics are decoupled (Fig 7c), and a strong safety-
efficiency tradeoff when stem and leaf economics are tightly coupled (Fig 7f).

When we paired our sapwood PV data with data on xylem hydraulic safety, we found that
$C_{ft,x}$ (xylem capacitance) and $P_{50,x}$ demonstrated significant evidence (p = 0.02) for a trade-off

in drought avoidance (increasing capacitance) and drought tolerance (increasingly negative $P_{50,x}$)





(Fig 8b). The strength of this relationship became marginally insignificant (p = 0.05) when considering $P_{50,x}$ values obtained with the bench dehydration (DH) method only, but was unaffected by the correction factor that we applied to the sapwood PV curves, remaining significant (p = 0.02 and p = 0.01 for all data and DH-limited data, respectively; Supplement Fig.

S2.4). Because $C_{ft,x}$ and $P_{50,x}$ both derive directly from *WD*, our plant hydraulics model parameterization also follows this avoidance-tolerance tradeoff (cf thin line Fig 8b).

Finally, joining of the leaf PV database with the xylem functional traits database demonstrated significant evidence in support of coordination between leaf and xylem drought tolerance, as given by significant relationships between leaf drought tolerance traits $\pi_{o,l}$ and associated turgor

loss point $\pi_{tlp,l}$ with xylem drought tolerance traits $P_{50,x}$ and $P_{88,x}$, albeit with $R^2$ values no greater than 0.3 (Fig 8c-f). This cross-tissue coordination is also preserved by our model parameterization, with *WD* driving most of the variation in this coordination space and *LMA* generating residual variation in leaf drought tolerance (thin lines, Fig 8c-f).

### 4.2 Model idealized experiments

### 4.2.1 Impact of plant hydraulics on size- and light-dependency of transpiration

Figure 9 shows how mean +/- 1 s.d. diurnal cycles of transpiration change as a function of canopy position and wet/dry season for both TFS v.1 and TFS v.1-Hydro. Simulated transpiration was mainly driven by the variation in radiation depending on the wet/dry season and the canopy locations. Specifically, the seasonality of cloud cover associated with wet and

dry seasons at this equatorial site drives a large seasonal increase from the wet to dry season in incoming radiation, and consequently, absorbed photosynthetically active radiation (PAR) (Fig 9a-d) (Carswell, 2002; Fisher et al., 2007). At the level of individual tree crowns, canopy position is the dominant control over the amount of absorbed PAR (Fig 9a-d). In TFS v.1, the large increase in incoming radiation from the wet to dry season as well as the vertical gradient in

absorbed radiation drives comparatively large increases in transpiration per unit crown area (Fig 9e, f). These trends remain when considering total tree transpiration (Fig 9i, j), but variation in crown size adds significant variability in total water fluxes, especially for canopy trees (very small trees can still be in the canopy depending on subplot assignment). These seasonal and canopy gradient trends remained in TFS v.1-Hydro with some significant modification, however.



Hydraulic limitation imposed by stomatal closure ($P_{50,gs}$; see Eqn 5) was highest in canopy trees, intermediate in subcanopy trees, and nonexistent in understorey trees, but this limitation only occurred during the dry season when incoming radiation was sufficiently high to drive high potential evapotranspiration (Fig 9g, h, k, l), consistent with other studies (McDowell et al.,
5    2005).

### 4.2.2 Impact of plant hydraulics on the trait-dependency of transpiration

Figure 10 shows how mean +/- 1 s.e. diurnal cycles of transpiration per unit crown area of three large (50-55 cm DBH and 26-27 m height) individuals change as a function of a single plant trait ($P_L$, *WD* or *LMA*) and wet/dry season for both TFS v.1 and TFS v.1-Hydro. In most cases, the
seasonal wet to dry increase in transpiration associated with incoming radiation is the same as for the previous model experiment. In the absence of plant hydraulics, leaf nutrients drive large differences in photosynthetic parameters and rates, which, by model design, then drive large differences in $g_{s,max}$ and transpiration rates (Fig 10a, b). Because our plant hydraulic trait parameterization explicitly links hydraulic transport rates ($k_{s,max,x}$) to photosynthetic capacity,
this trend is preserved in TFS v.1-Hydro, except that all trees experience hydraulic limitation (Fig 10c, d). In contrast, large canopy trees with differences in *WD* and *LMA* in default TFS simulations demonstrated no differences in simulated transpiration dynamics (Fig 10e, f). In the case of *WD*, adding plant hydraulics caused hydraulic limitation across all three individuals but surprisingly little to no divergence in simulated dynamics (Fig 10g, h), despite large differences
in plant hydraulic traits which derive from *WD* (Table 2). This lack of a difference arises, however, because the effects of decreasing *WD* have opposing effects on midday leaf water potential via sapwood capacitance and xylem P50 because of the drought avoidance – tolerance tradeoff (Fig. 8b). In the case of *LMA*, dramatic differences emerged among the three individuals with differing *LMA*s (209, 95 and 47 g m-2) upon the inclusion of plant hydraulics
(Fig 10k, l). This was driven by large differences in total tree aboveground hydraulic conductance ($K_{max,tree,ag}$), which arose because of large differences in $A_l{:}A_s$ that ranged from 0.1 – 1.8 m$^2$ cm$^{-2}$ (itself arising from the *LMA* differences; see Fig 7g, h and Table 2). By Darcy's law, a reduction in conductance results in a comparative reduction in water flux for a given water potential gradient.



### 4.3 Model evaluation

TFS v.1-Hydro demonstrated capability to represent lags in water flux throughout the leaf-stem-soil continuum (Fig. S3.1), with sap flux lagging leaf-level transpiration, and root uptake continuing into the nighttime hours, consistent with diurnal dynamics of soil moisture shown in

other ecosystems. In addition, as expected, the model predicted that soil most adjacent to absorbing roots experiences significantly lower rates of volumetric root water uptake as compared to more distal rhizosphere shells (Fig S3.2), demonstrating capability for the model to represent rhizosphere hydraulic limitation if soil gets sufficiently dry.

### 4.3.1 Diurnal dynamics of leaf and stem water potential

Observed diurnal dynamics of leaf and stem water potential ($\psi_{leaf}$ and $\psi_{stem}$, respectively) indicated, for all trees regardless of size, predawn (~ 6am) values in the range of -0.1 to -0.4 MPa, whereas midday water potentials varied largely in concert with tree size (Fig 11). The model captured the observed diurnal trends in both $\psi_{leaf}$ and $\psi_{stem}$ reasonably well, especially given the absence of any model tuning or tissue-level hydraulic trait differences among

individuals. This suggests that the first-order control over variation in $\psi_{leaf}$ among individuals was primarily due to variation in radiation interception (both across seasons and within the canopy) and tree size, rather than tissue-level hydraulic traits. See Fig S3.4 for an alternate version of Fig. 11 showing how root water potentials vary diurnally for these individuals.

Despite large (upwards of -4 MPa) differences in midday and predawn leaf values, observed

$\psi_{stem}$ remained for the most part within -1 MPa of the observed $\psi_{leaf}$ values (Fig 11). For the larger trees, observed midday $\psi_{stem}$ during the dry season was in the range of -2 to -3 MPa. In contrast, modeled $\psi_{stem}$ rarely went more negative than -1 MPa, suggesting that the model parameterization overestimates sapwood capacitance ($C_{ft,x}$).

The data do not support the existence of an effect of TFE on water potential, either in the $\psi_{stem}$

or $\psi_{leaf}$ (compare dry season control and TFE columns in Fig. 11), at least within the first two years of the experiment when these measurements were collected. The model did not predict significant effects of the TFE treatment on water potential either, due to the fact that dry season $\psi_{leaf}$ approached $P_{50,gs}$ and were regulated near $P_{50,gs}$ via Eqn 5.



### 4.3.2 Impact of model parameterization on fidelity of simulated midday leaf water potential

Modeled dynamics of leaf water potential at the Caxiuana site captured observed variation at both diurnal and seasonal timescales (Supplement Fig S2.6). Throughout each day, reductions in leaf water potential lagged that of the stem and transporting root system, congruent with the
buffering role of canopy and stem capacitance, with refilling of distal tissues occurring at night (results not shown). The model also simulated reductions in predawn leaf water potential in the dry season due to reductions in soil water supply, and captured an increasing magnitude of diurnal variation driven largely by light-driven increases in canopy transpiration rates typical of equatorial Amazonian dry seasons (Fig. S6) (Carswell, 2002; Christoffersen et al., 2014; da
Rocha et al., 2009; Fisher et al., 2007; Hasler and Avissar, 2007).

Each of three successive model parameterization adjustments was able to make incremental improvements in the fidelity of modeled $\psi_{leaf}$, and these adjustments are informative for understanding the dominant controls over individual-level differences in $\psi_{leaf}$. First, when the model's treatment of xylem conduit taper is turned off, the model in most cases overestimated
the negative hydraulic impacts of tree height, with modeled midday $\psi_{leaf}$ falling below that which was observed (Fig 12a). Re-instating xylem taper ameliorated the negative effects of tree height for some trees by increasing $K_{max,tree,ag}$ and thus reducing the soil-leaf water potential gradient needed to maintain transpiration (Fig 12b). Next, adjusting the canopy position of two of the four individual trees from a subcanopy (canopy layer = 2) light environment to an
understorey (canopy layer = 3) light environment reduced the incoming radiation for these trees and hence their transpiration rates and midday $\psi_{leaf}$ values (Fig 12c). Finally, we found that the extremely negative midday $\psi_{leaf}$ values during the dry season of the largest tree (tree C3 of Fisher et al., 2006) could be better simulated by allowing its stomatal regulation to behave in a more anisohydric way by setting its $P_{50,gs}$ = -5.0 MPa.

**4.3.3 Fidelity of modeled stand-level transpiration under control and TFE conditions**

Without plant hydraulics, TFS v.1 had a large positive bias in simulated dry season stand-level transpiration rates (Fig 13a,b). Including plant hydraulics (TFS v.1-Hydro) largely eliminated this dry season positive bias when plot-mean trait values were used (Fig 13a). However, TFS v.1-Hydro tended to underestimate observed transpiration year-round when each tree was





assigned different trait values (Fig 13c) according to the observed plot distribution (according to Fig. 7a-c). As expected, the simple soil bucket models of TFS v.1 and TFS v.1-Hydro precluded both of them from capturing the observed pronounced reduction of transpiration under 50% TFE (Fig 13b, d).

**5 Discussion**

We present here a plant hydraulics model built from the bottom up. By "bottom-up" we mean that this model is parameterized at the level of plant tissues (leaf, stem, roots) and then scaled to the whole-tree level using established theory about how size and aboveground branching structures impact whole-tree function. We embedded this model within a size-structured forest

simulator (TFS) providing critical input information to the plant hydraulics model on individual tree light environments and community-level size and trait distributions (collectively these two models comprise TFS v.1-Hydro), but the plant hydraulics scheme is appropriate for inclusion within any demographic dynamic vegetation model. Our plant hydraulics model is also "trait-driven". By "trait-driven" we mean that individual trees take on distinct plant functional trait

values ($WD$, $LMA$, $N_L$, $P_L$) according to observed plot-level trait distributions and we have parameterized hydraulic traits from each tree's assigned traits according to a pantropical empirical synthesis. This synthesis documented widespread correlations between these plant hydraulic traits and those used in our model (Figs 2-6), and we additionally showed that even where such correlations were weak, the hydraulic trait parameterization for our model respected

observed ecological trade-offs and coordinations among hydraulic traits (Figs 7-8) known to be crucial to plant water use and survival. As a consequence, TFS v.1-Hydro has novel capability to simulate divergent functional responses to environmental variation (Figs 9-10) owing to the hydraulic impacts of tree size and coordinated shifts in hydraulic traits across individuals. This provides encouragement to the growing research agenda to utilize traits to parameterize Earth

System Models (Fisher et al., 2015; Sakschewski et al., 2015). The model also reasonably captures size- and light-related inter-individual variation in water potential across individuals (Figs 11-12) without need for significant model tuning. Finally, the model makes substantial improvements to TFS v.1 in terms of simulated transpiration rates under normal conditions (Fig 13a, c). The only place where the model demonstrably fails is under experimental drought (Fig





13b, d), a consequence of its simplified belowground soil hydrology scheme, which will be updated in subsequent version.

### 5.1 Underlying causes and implications of trait-trait relationships

The relationship of leaf $\pi_{o,l}$ (i.e., cell "saltiness") with *WD* is robust, and suggests some degree of correspondence between sapwood and leaf osmoregulation at the cellular level, itself potentially a function of soil potassium ($K^+$) availability (Quesada et al., 2012), although there was not significant enough species overlap in our leaf and sapwood databases to directly assess whether $\pi_{o,l}$ tracks $\pi_{o,x}$. An additional possibility is that species with a tendency to osmoregulate leaf tissue also have sapwood with higher *WD* and embolism resistance, though recent results challenge this explanation (Binks et al., 2016). The observation that leaf $\varepsilon_l$ (i.e., leaf cell "stiffness") was coordinated more closely with stem *WD* rather than *LMA* (Fig 2) may be due more to uncertainty in how $\varepsilon_l$ is calculated, either based on symplastic or total relative water content (*RWC** or *RWC*, respectively) (Bartlett et al., 2012), rather than strong evidence for stem over leaf traits governing leaf cell stiffness *per se*. There may be a mix of both methods used in calculating $\varepsilon_l$ in this dataset due to ambiguity in the original publications (M. Bartlett, personal communication), so the relationship of $\varepsilon_l$ with *WD* should be viewed as tentative.

The correspondence of sapwood PV parameters and associated capacitance with *WD* (Fig 3) was in some cases particularly tight ($r^2 = 0.84$ for sapwood capacitance $C_{ft,x}$). Given the uncertainty associated with the vessel cutting artefact for which we corrected (Supplement Notes S2), the exact relationships with *WD* should also be considered tentative, but the qualitative nature of *WD* as a 1st-order predictor of $C_{ft,x}$ should be robust. Nonetheless, the absence of data for sapwood PV traits for *WD* > 0.7 g cm$^{-3}$ highlights a real data need, which is particularly apparent for some regions of the tropics, such as eastern and north-eastern Amazonia, which have a large abundance of high wood density species (Baker et al., 2004; Quesada et al., 2012). 2nd-order predictors for sapwood capacitance have been identified in separate studies, which we have not explicitly dealt with here. Recently, Wolfe & Kursar (2015) identified a similar trend of $C_{ft,x}$ with *WD*, but additionally identified deciduousness as a modulator of $C_{ft,x}$, with evergreen species tending to have lower $C_{ft,x}$ at a given *WD*.



The increase of xylem efficiency ($k_{s,max,x}$) with $A_{max}$ we highlighted in our synthesis (Fig 5) has already been identified in individual studies (Brodribb et al., 2002; Santiago et al., 2004; Zhu et al., 2013), but to our knowledge, our study is new in demonstrating it across diverse tropical forests spanning wet, moist, and dry systems (and to some degree, savanna) when incorporating

*WD* as a secondary predictor, which increased by 13% the variance explained by $A_{max}$ alone. This relationship underscores the functional constraint of xylem "plumbing" on leaf gas exchange. This relationship potentially presents an important constraint for plant hydraulics modeling: given that *WD* is a poor predictor of the relevant wood anatomical characteristics mechanistically controlling xylem hydraulic conductivity in tropical angiosperms (Poorter et al.,

2010; Zanne and Falster, 2010; Zanne et al., 2010; Zieminska et al., 2015), $A_{max}$ serves as a powerful proxy, with *WD* acting as an additional constraint (albeit small) on the relevant anatomical variation. This is important in light of the relative ubiquity of leaf nitrogen and phosphorus measurements (which in turn are good proxies for $A_{max}$; Domingues et al. 2010) relative to wood anatomical measurements (Fichtler and Worbes, 2012) in tropical forests

worldwide.

Our synthesis of hydraulic traits suggests that a trade-off between drought avoidance (as given by $C_{ft,x}$) and tolerance ($P_{50,x}$) is more prominent than between xylem efficiency ($k_{s,max,x}$) and safety ($P_{50,x}$), though more data are needed to corroborate the limited data suggesting the former (Fig. 5). Partially responsible for this weak safety-efficiency tradeoff is methodological variation

in terms of how $k_{s,max,x}$ is measured (Gleason *et al.* 2016). However, there is undoubtedly a real component to this weak trade-off, as intervessel pit membrane thickness, which does not necessarily correlate with hydraulic conductivity, is a strong anatomical predictor of $P_{50,x}$ (Li et al. 2016). Perhaps of greater significance, however, is understanding patterns associated with the presence or absence of the efficiency-safety tradeoff in particular bioclimatic regions. The

correspondence of this tradeoff with tropical dry forests where stem economics traits closely track leaf economics traits (Markesteijn et al., 2011a; Markesteijn et al., 2011b; Mendez-Alonzo et al., 2012), in contrast to their wetter counterparts (Baraloto et al., 2010), suggests that the hydraulic safety-efficiency trade-off may be an underlying mechanism for the extent to which stem and leaf trait coupling is observed.





### 5.2 Model limitations

Our analysis reveals some limitations of our particular approach to trait-driven plant hydraulics modeling, namely that not all trait-trait relationships are of sufficient predictive accuracy to enable simulations with low uncertainty (most notably leaf drought tolerance traits and the

leaf:sapwood area ratio, see Figures 2 and 6). This is a likely outcome of the reality that emergent properties of biological systems arise from the combination of multiple traits and their associated trade-offs, thus no single trait is likely to explain the majority of an emergent process (Mencuccini et al., 2015). More work identifying general plant trait-hydraulic trait relationships is justified, but should consider incorporation of multiple traits for improved realism and

accuracy, and should emphasize understanding how trees as an integrated system function.

One consequence of using the empirical trait-trait correlations given here is to either under- or over-estimate the range of any given hydraulic trait in a simulated community of individuals, depending on the hydraulic trait of interest. For example, in our idealized model experiment in which we varied a single trait (Fig. 10), variation in xylem-specific hydraulic conductivity

($k_{s,max,x}$; Fig 10c, d), which was derived from the full range of observed $P_L$ (and hence, modeled $A_{max}$) at this site, was limited to only 1.1 - 2.9 kg m$^{-1}$ MPa$^{-1}$ s$^{-1}$, which is only ~15% of the range of observed $k_{s,max,x}$ values for tropical moist forests (cf. green points, Fig S3). Indeed it seems unlikely that the observed range of $k_{s,max,x}$ at this site would be this small. On the other hand, the range of $A_l:A_s$ predicted by *LMA* and tree height (Fig 6 and 7g) is likely overestimated;

we found that individuals with $A_l:A_s$ greater than the maximum observed $A_l:A_s$ (1.8 m$^2$ cm$^{-2}$) for whole trees in the Calvo-Alvarado et al. (2008) study were unviable (whole-tree conductance was insufficient for any appreciable tree transpiration to occur). Indeed, our result showing that a slight negative bias results when all individuals receive distinct trait values according to the community trait distribution versus a single community-mean value (Fig 13a vs 13c) follows as a

consequence. Furthermore, in contrast to our simple implementation of hydraulic architecture, $A_l:A_s$ may vary substantially throughout a tree's crown, at different branching levels, or different light environments (see Schuldt et al., 2011 as an example), given the deviations from self-similar, volume-filling branching and area-preserving $A_s$ that have been observed (Bentley et al., 2013; Smith et al., 2014; Whitehead et al., 1984).



The expected deficiency of TFS v.1-Hydro to capture experimentally induced soil water deficit (Fig 13b, d) relates to the absence of any significant soil drying within the root zone. The current model formulation does not vertically discretize the soil water or root distribution, whereas observations (Rowland et al., 2015a) indicate a strong vertical gradient in soil moisture, with the

driest soil occurring in shallow layers where roots are most abundant. The lack of sensitivity is not due to a lack of vegetation sensitivity (else TFS v.1-Hydro would not down-regulate transpiration relative to TFS v.1 under control conditions; cf Fig 13a, c), nor is it due to discrepancies in soil hydraulic parameters, as comparison of measured (Fisher et al., 2008) vs. pedotransfer-modeled van Genuchten parameters revealed no significant differences in soil water

holding capacity. The importance of capturing soil drying in the appropriate portions of the root zone for capturing drought response is an outcome that corroborates previous work showing that soil-root resistance alone (Williams et al., 2001) or some combination soil-root and within-root resistance are the key drivers of sharp reductions in water use under drought conditions (Fisher et al., 2007). We thus expect that extending our plant hydraulics model to incorporate vertical

variation in root length density and soil water content will make strong improvements in this regard.

### 5.3 Practical implications

Because TFS v.1-Hydro parameterized the majority of tree hydraulic traits at the scale of plant tissues and built the size-scaling into the model (rather than specifying hydraulic properties at the

whole-tree level), it has great potential for being used as a tool in data assimilation and other inverse approaches when in situ field data on sap flow, water potential, or water content data are available at one or multiple points within individual trees. This may aid in deciphering unobserved or unobservable processes, such as diagnosing where within the soil-plant continuum hydraulic limitation or embolism may be occurring (branches, stems, or fine roots) or how much

stored water is used for daily transpiration.

A second benefit to its method of construction is that the constitutive equations of the model (i.e., the relationships between water potential and water content, and between hydraulic conductivity and water potential), use formulations from plant physiology (such as PV theory) and as a consequence, have parameters which are empirically measurable and biologically

interpretable, such as saturated and residual tissue water contents $\theta_s$ and $\theta_r$, osmotic potential at





full turgor $\pi_o$, turgor loss point $\pi_{tlp}$, bulk elastic modulus $\varepsilon$, xylem water potential at 50% loss of conductivity $P_{50,x}$, and leaf water potential at 50% loss of maximum stomatal conductance $P_{50,gs}$. Hence, as more of these hydraulics data become available, this model can readily incorporate this information by either substituting the empirical trait relationships with measured
quantities, or updating the empirical hydraulic trait relationships.

### 5.4 Future directions

In light of the limitations to our empirical approach for parameterizing hydraulic traits via more commonly measured traits, one potential alternative for parameterizing community-level hydraulic trait variation is to do so independent of other traits employed by the model. One
approach is to use the Taylor & Thompson (1986) resampling algorithm to resample individuals or species with values in multiple trait databases which were used in this paper, but sparse database overlap may limit the utility of this approach. Alternatively, region-specific distributions of individual hydraulic traits from the databases presented here could be coupled with empirical relationships *among* hydraulic traits as given by our synthesis of coordination and
trade-offs among hydraulic traits (Fig 8). This analysis suggested that two independent hydraulic trait spectra may be sufficient: one safety-efficiency or avoidance-tolerance trait spectrum and potentially a stomatal hydraulic safety spectrum. More work is needed to clarify these hydraulic coordination and trade-off surfaces and in particular, to quantify how much residual variation is real or due to measurement biases.

From an empirical perspective, explaining the remaining 16 – 95% of unexplained variance in hydraulic traits shown here should be of high priority. We suggest some additional traits that show promise. The volumetric fractions of airspace, water, and solid material in leaves and stems (sensu (Roderick and Berry, 2001; Roderick et al., 1999)) are one such trait. For leaves, these volumetric fractions have been shown to explain much more variance in leaf PV traits than
*LMA* and *WD* (Bouche et al., 2015; Buckley, 2015; Sack et al., 2003; Scoffoni, 2015). While not as routine of a measurement in comparison to *LMA*, these measurements are not as intensive as the full PV curve. Second, identifying the strongest wood anatomical correlates of xylem vulnerability to embolism ($P_{50.x}$) should also be of high priority, such as the thickness of the intervessel pit membranes (Choat et al., 2008; Jansen et al., 2009), as well as improving
understanding of the air-seeding mechanism (Jansen and Schenk, 2015; Schenk et al.) Third,



and perhaps most important for whole-tree hydraulics, is more data on hydraulic architecture. Tuning hydraulic architecture in our dynamic model proved to be a key 'knob' for matching observations. Additional measurements of $A_l : A_s$, xylem taper, and crown branching patterns should be conducted not in isolation, but in tandem with each other and with other leaf and stem

economics traits (in particular *LMA*, as suggested by Fig 6b) to isolate physical constraints on structure and the dynamics of whole-tree transpiration (Smith et al., 2014). The paired relationship of $A_l : A_s$ with xylem taper as integrative measures of whole-tree hydraulic regulation is virtually nonexistent for tropical forests, although collocated measurements of these two traits exist (Horna et al., 2011; Zach et al., 2010).

For modeling hydraulic impacts on stomatal closure, we favored the use of a 'stomatal vulnerability function,' but this is one possibility among many. Specifically, multiple studies have found that branch $\psi$ rarely falls below that at which stem capacitance begins to decrease precipitously (Meinzer et al., 2008; Wolfe and Kursar, 2015), and stomatal regulation has also been observed to track leaf turgor loss points (Brodribb et al., 2003). Whether stomatal closure

in response to water stress in tropical forests forms an entirely distinct 'safety margin' axis (Martinez-Vilalta et al., 2014; Skelton et al., 2015), or maps on to other hydraulic traits remains to be elucidated. A recent review highlights the interacting roles of both hydraulic supply and vapor pressure deficit in controlling stomatal responses, which we have not yet accounted for (Sperry and Love, 2015). We also did not include into our model variability in leaf-level

hydraulic conductance, or comparative values for root traits. While terminal branches and leaves account for a majority of total plant resistance due in part to xylem taper, roots are more vulnerable to cavitation for the same reason and therefore represent a weak link in drought-induced mortality; these should be topics of future work.

**6 Conclusions**

We present a plant hydraulics model based on biologically interpretable and measureable plant hydraulic traits and rooted in established plant physiology theory. It is capable of scaling tissue-level hydraulic traits to whole-tree hydraulic function. Embedding it within an individual tree trait-driven model allowed us to explore how individual-level variation in hydraulic traits (including tree size) interacts with light environments to drive differences in hydraulic function



across individuals. Addition of plant hydraulics made substantial improvements to modeled transpiration fluxes during periods of high incoming radiation. Our synthesis of data for the tropics on a wide range of hydraulic traits needed to parameterize the model allowed us to develop empirical relationships among commonly measured plant traits and less common

hydraulic traits. These 'ecotransfer' functions, as we have termed them, are analogous to pedotransfer functions (Clapp and Hornberger, 1978; Cosby et al., 1984) developed for predicting soil hydraulic properties as a function of soil textural 'traits.' While they should be refined as more data become available, they are an important first step towards representing ecological dimensions of hydraulic trait variation in process ecosystem models . Critically, this

individual- and trait-driven framework provides a testbed for identifying both critical processes and functional traits needed for inclusion in coarse-scale Dynamic Global Vegetation Models, which will lead to reduced uncertainty in the future state of tropical forests under climate change.

**Code Availability**

The JAVA source code for TFS-Hydro can be obtained from the corresponding author upon
request.

**Data Availability**

The Supplement S4 describes supplementary data for this paper. Leaf PV data (Figs 2 and 8) that were newly extracted from publications are available as .csv data files as described under the heading "Leaf PV database". Leaf PV data originating from pre-existing leaf PV databases are
available by accessing the original articles cited under the heading "Leaf PV database." Sapwood PV data (Figs 3, S2.1, and S2.3) are available as .csv data files as described under the heading "Sapwood PV database." Sapwood area data (Fig 7h) are available as .csv data files as described under the heading "Sapwood Area database." Xylem hydraulic trait data (Figs 4, 5, 8, and S2.2) that were newly extracted from publications are available as .csv data files as
described under the heading "Xylem Functional Traits Database." Xylem hydraulic trait data originating from the TRY archive can be accessed there (www.try-db.org); the references of data from that archive used in analyses here are given under the heading "Xylem Functional Traits Database."

**Acknowledgments**



This research was supported in part by the European Union Seventh Framework Programme, under the project AMAZALERT (Grant Agreement No. 282664 to PM) and by the Next-Generation Ecosystem Experiments (NGEE-Tropics) project, which is funded by the Office of Biological and Environmental Research of the U.S. Department of Energy (DOE). PM also

acknowledges ARC support from FT110100457. This submission is under public release with the approved LA-UR-16-20338. We thank all colleagues who contributed data to the Xylem Functional Traits Database, as well as Rick Meinzer for sharing ancillary *LMA* data and Megan Bartlett for answering clarifying questions.  We additionally thank Brett Wolfe, Rafael Oliveira, Yadvinder Malhi and John Sperry for helpful discussions, Crystal Ng for suggesting the pithy

term 'ecotransfer function,' and Mark Decker and Guo-Yue Niu for helpful discussions regarding the use of the mass-based solution to the Richards equation.

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



**Tables**

**Table 1.** Input tree size and leaf and stem traits used by the TFS model.

| Trait | Symbol | Units |
|---|---|---|
| Tree height | $H$ | m |
| Tree diameter at breast height | $DBH$ | cm |
| Leaf mass per area | $LMA$ | g m$^{-2}$ |
| Leaf nitrogen | $N_L$ | mg g$^{-1}$ |
| Leaf phosphorus | $P_L$ | mg g$^{-1}$ |
| Light-saturated photosynthesis rate | $A_{max}$ | µmol m$^{-2}$ s$^{-1}$ |
| Wood density | $WD$ | g cm$^{-3}$ |





**Table 2**. hydraulic traits used by the model. Resultant empirical "ecotransfer" function which
estimates hydraulic traits as a function of input traits and its corresponding source.

| Parameter | Symbol | Units | Empirical "ecotransfer" function | Source |
|---|---|---|---|---|
| Leaf saturated water content | $\theta_{s,l}$ | m³ m⁻³ | $\dfrac{-2.32 * 10^4/LMA + 782}{\rho_w}\left(\dfrac{1}{-0.21 ln(10^4/LMA) + 1.43} - 1\right)$ | Stewart et al. (1990), L. Rowland & B. Christoffersen (unpublished data) |
| Leaf osmotic potential at full turgor | $\pi_{0,l}$ | MPa | $-0.04 - 1.51 WD - 0.0067 LMA$ | This study |
| Leaf osmotic potential at turgor loss | $\pi_{tlp,l}$ | MPa | $\dfrac{\varepsilon_x \pi_{0,1}}{\varepsilon_x + \pi_{0,1}}$ | Bartlett et al. (2012) |
| Leaf bulk elastic modulus | $\varepsilon_l$ | MPa | $2.5 + \dfrac{37.5}{1 + e^{-8.0 WD + 5.7}}$ | This study |
| Leaf relative water content at turgor loss | $RWC_{tlp,l}$ | (-) | $\dfrac{\left(\pi_{0,1}(1 - RWC_{r,l}) + \varepsilon_l\right)}{\varepsilon_l}$ | Bartlett et al. (2012) |
| Leaf residual fraction | $RWC_{r,l}$ | (-) | $0.01 \varepsilon_l + 0.17$ | This study |
| Leaf capacitance over $RWC_1 = 1$ to $RWC_{tlp,1}$ | $C_{ft,l}$ | g H₂O g⁻¹ dry weight MPa⁻¹ | $\dfrac{\rho_w \theta_{s,l}(1 - RWC_{r,l})(\varepsilon_l + \pi_{0,l})}{\varepsilon_l^2}$ | Bartlett et al. (2012) |
| Sapwood saturated | $\theta_{s,x}$ | m³ m⁻³ | $1 - \dfrac{WD}{1.54}$ | Siau 1984 |





| | | | | |
|---|---|---|---|---|
| water content | | | | |
| Sapwood RWC at which capillary reserves exhausted | $RWC_{ft,x}$ | (-) | $1 - 0.72(1 - RWC_{tlp,x})$ | This study |
| Fraction of $1 - r_{f,2}$ that is capillary in source | $f_{cap}$ | (-) | 0.07 | This study |
| Sapwood osmotic potential at full turgor | $\pi_{0,x}$ | MPa | $0.52 - 4.16 * WD$ | This study |
| Sapwood osmotic potential at turgor loss | $\pi_{tlp,x}$ | MPa | $\dfrac{\varepsilon_x \pi_{0,x}}{\varepsilon_x + \pi_{0,x}}$ | Bartlett et al. (2012) |
| Sapwood bulk elastic modulus | $\varepsilon_x$ | MPa | $\sqrt{1.02e^{8.5WD} - 2.89}$ | This study |
| Sapwood RWC at turgor loss | $RWC_{tlp,x}$ | (-) | $1 - \dfrac{(1 - 0.75 * WD)}{(2.74 + 2.01 * WD)}$ | This study |
| Sapwood residual fraction | $RWC_{r,x}$ | (-) | $\dfrac{\varepsilon_x(1 - f_{cap} - RWC_{tlp,x})}{\pi_{0,x}(1 - f_{cap})} + 1 - \varepsilon_x f_{cap}$ | This study |
| Sapwood capacitance over $RWC_1 = 1$ to $RWC_{tlp,1}$ | $C_{ft,x}$ | kg m$^{-3}$ MPa$^{-1}$ | $\dfrac{\rho_w \theta_{s,x}(1 - RWC_{r,x})(\varepsilon_x + \pi_{0,x})(\pi_{0,x}(1 - f_{cap}) - \varepsilon_x f_{cap})}{\pi_{0,x} \varepsilon_x{}^2}$ | This study |
| Xylem water potential at 50% loss of | $P_{50,x}$ | MPa | $-(3.57WD)^{1.73} - 1.09$ | This study |





| conductivity | | | | |
|---|---|---|---|---|
| Slope of xylem vulnerability curve at $P_{50}$ | $a_x$ | MPa$^{-1}$ *100 | $54.4(-P_{50,x})^{-1.17}$ | This study |
| Maximum xylem conductivity per unit leaf area | $k_{l,max,x}$ | kg m$^{-1}$ s$^{-1}$ MPa$^{-1}$ | $0.0021e^{-26.6WD/A_{max}}$ | This study |
| Maximum xylem conductivity per unit sapwood area | $k_{s,max,x}$ | kg m$^{-1}$ s$^{-1}$ MPa$^{-1}$ | $\dfrac{k_{l,max,x}}{A_l A_s}$ | This study |
| Leaf to sapwood area ratio | $A_l : A_s$ | m$^2$ cm$^{-2}$ | $546LMA^{-2.14}H$ | This study |
| Leaf to fine root absorbing surface area ratio | $A_l : A_r$ | m$^2$ m$^{-2}$ | 1 | This study |





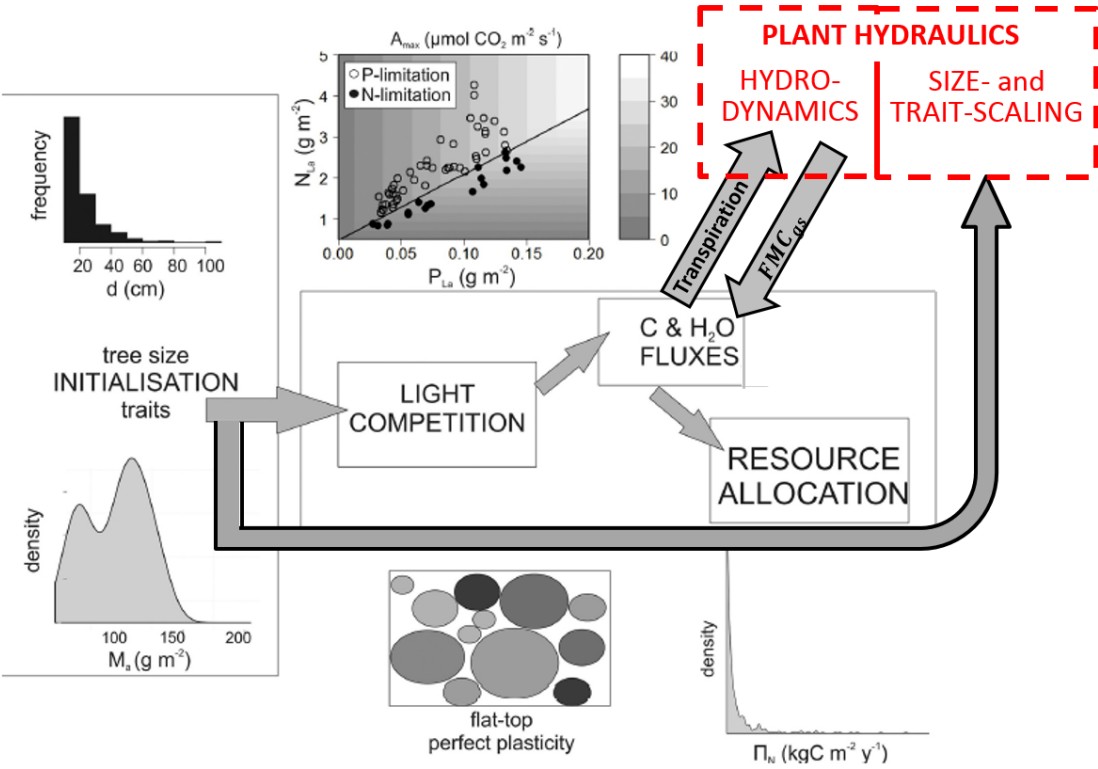

**Figure 1**. Integration of new plant hydraulics scheme (enclosed in dashed box) with host model TFS.
Arrows indicate main sources of information flow among the two models.  See Supplement Fig S1.1
for the structure of the hydrodynamic component of the plant hydraulics model.



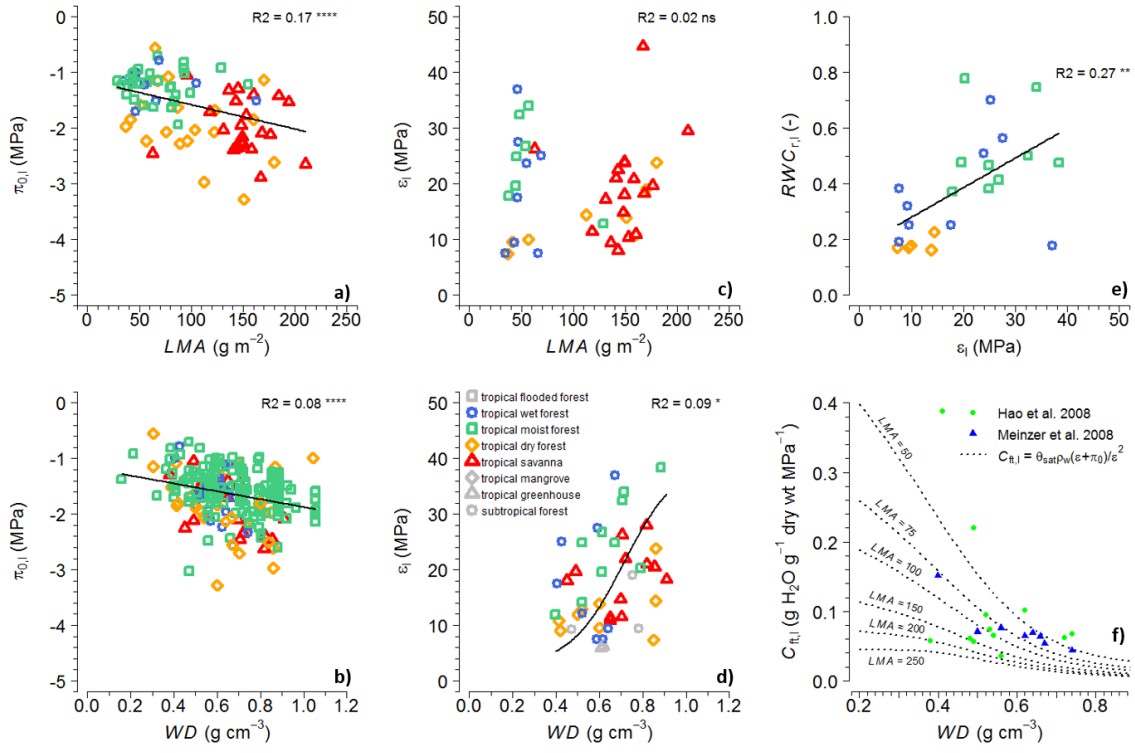

**Figure 2**. Tropical synthesis of leaf PV parameters in relation to *LMA*, *WD*, or each other. **a), b)** leaf osmotic potential at full turgor ($\pi_{o,l}$) in relation to *LMA* and *WD*; **c), d)** Leaf bulk elastic modulus ($\varepsilon_l$) in relation to *LMA* and *WD*; **e)** residual fraction ($RWC_{r,l}$) in relation to $\varepsilon_l$, and **f)** capacitance ($C_{ft,l}$) in relation to *LMA* (different lines) and *WD*. *LMA* and *WD* values come first from published values and are supplemented with species-average values from GLOPNET (Wright et al. 2004) and the global wood density database (Zanne et al. 2009). In this and all other figures, the following asterisk codes





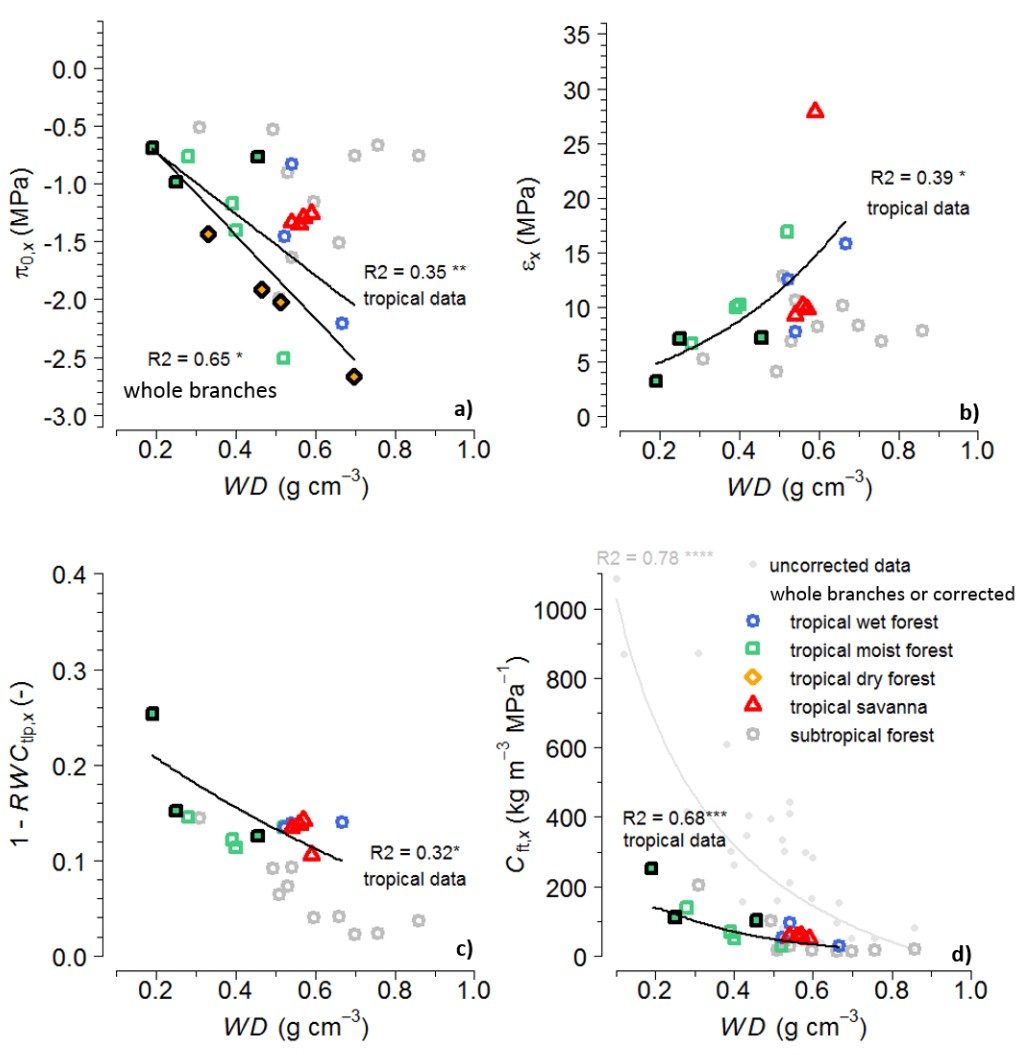

**Figure 3**. Tropical synthesis of sapwood PV parameters and resultant capacitance in relation to wood density (*WD*). **a)** osmotic potential at full turgor ($\pi_{o,x}$), **b)** bulk elastic modulus ($\varepsilon_x$), **c)** relative water deficit at turgor loss ($R_{tlp,x} = 1 - RWC_{tlp,x}$) and **d)** capacitance at full turgor ($C_{ft,x}$). Sapwood PV curves conducted on trunk cores, and their associated parameters and capacitances, are bias-corrected (see main text). Original data are shown in light gray in panel d) and Supplement Fig S2.1. Color scheme is given in the legend of panel **d)**. Asterisk codes for significance as in Fig. 2.





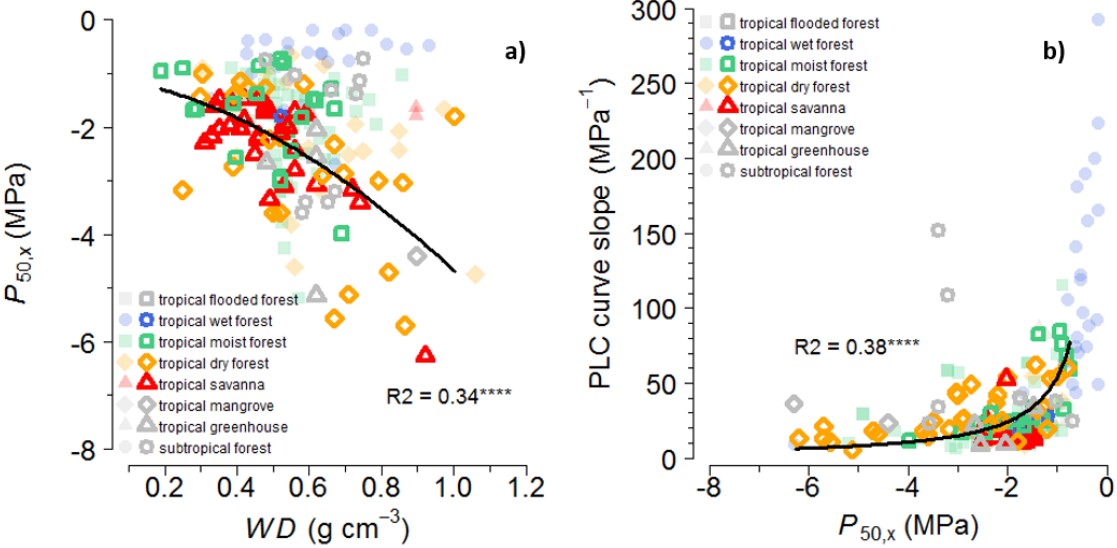

**Figure 4**. Tropical synthesis of xylem vulnerability. **a)** water potential at 50% loss of maximum
xylem conductivity ($P_{50,x}$). **b)** slope of PLC curve at ($P_{50,x}$) as a function of $P_{50,x}$.  Bold open
symbols: Bench dehydration method for $P_{50}$ measurement; light closed symbols: Air injection + all
other methods (see Methods 3.1 in main text).  Curves are fit through bench dehydration
measurements conducted on field-derived plant material in upland tropical forests & savannas only
(colored, bold open symbols). Asterisk codes for significance as in Fig. 2.





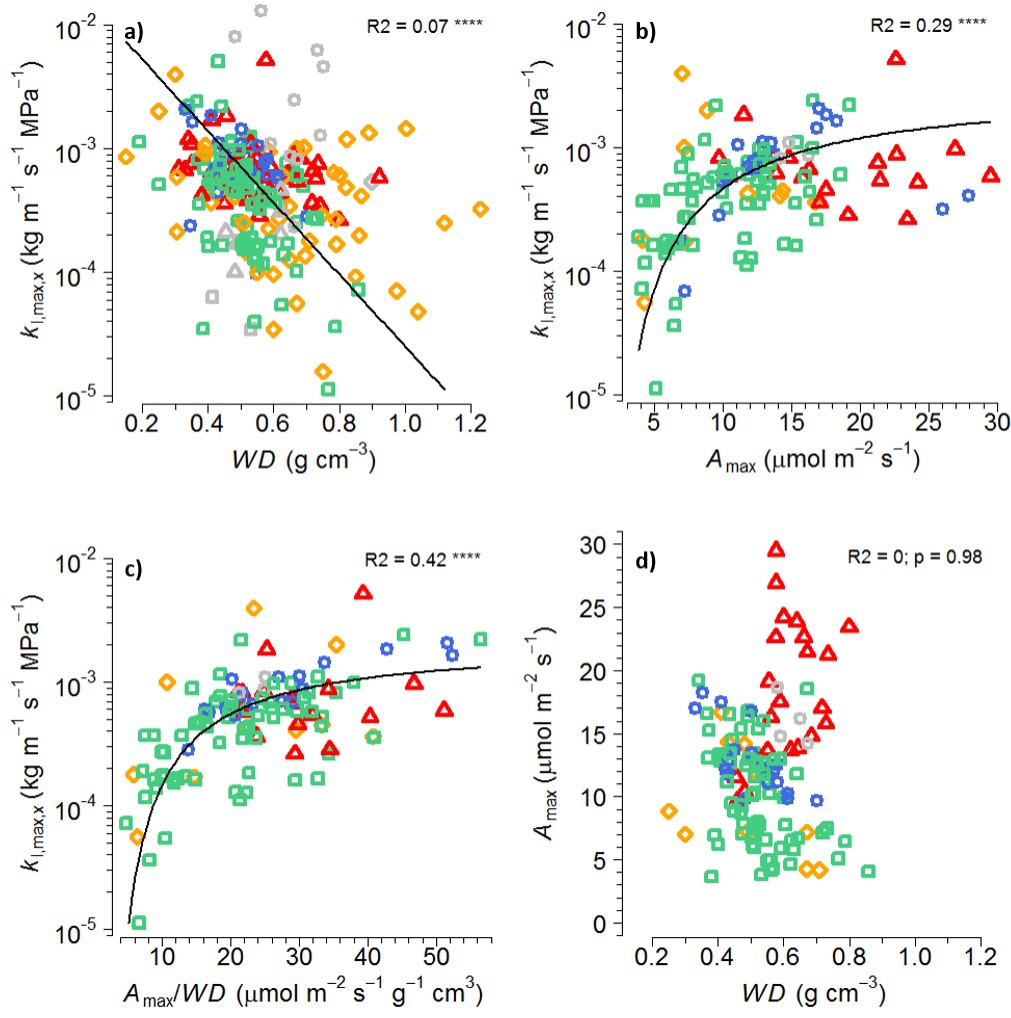

**Figure 5. a) – c).** Tropical synthesis of maximum xylem conductivity per unit leaf area ($k_{l,max}$) in relation to wood density ($WD$) and leaf maximum rate of photosynthesis ($A_{max}$). **d)** No significant correlation between $A_{max}$ and $WD$ exists in this dataset, justifying their use as independent simultaneous predictors of $k_{l,max}$. Symbols as in Figs. 2-3. See Supplement Fig. S2.2 for a version of this figure in terms of $k_{s,max}$. Asterisk codes for significance as in Fig. 2.





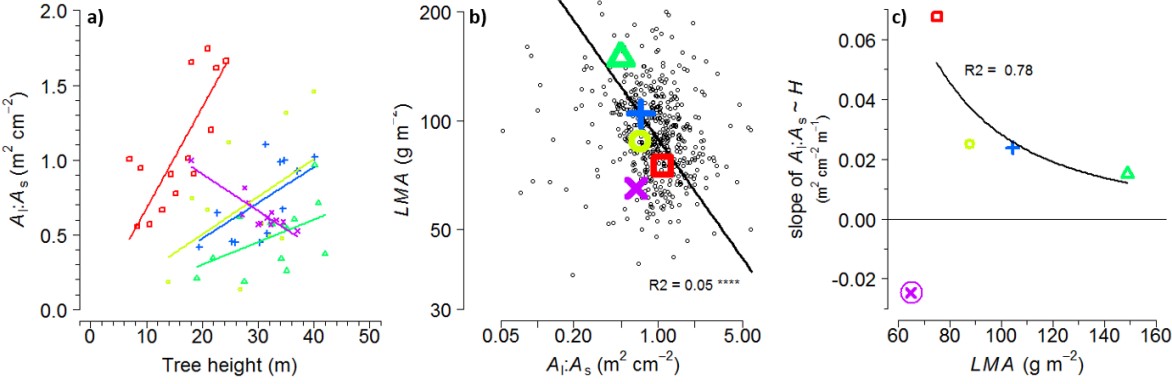

**Figure 6**. Ttropical synthesis showing the dual dependency of $A_l$:$A_s$ on both leaf *LMA* and tree size. **a)** $A_l$:$A_s$ versus tree height, replotted from Calvo-Alvarado et al. (2008). **b)** species-average *LMA* versus $A_l$:$A_s$ from a) overlain on a much broader dataset from the Amazon basin (replotted from Patino et al. 2012). **c)** The slope of the $A_l$:$A_s$ ~ H relationship in a) versus *LMA*. Circled datapoint is *Pentaclethra macroloba*, a compound-leaf species averaging 1600 leaflets / leaf and is excluded from the regression. The dependency of Al:As on LMA in b) is thus implemented in the model via variations in the slope of $A_l$:$A_s$ with *H* in c); see Fig 7g. Asterisk codes for significance as in Fig. 2.


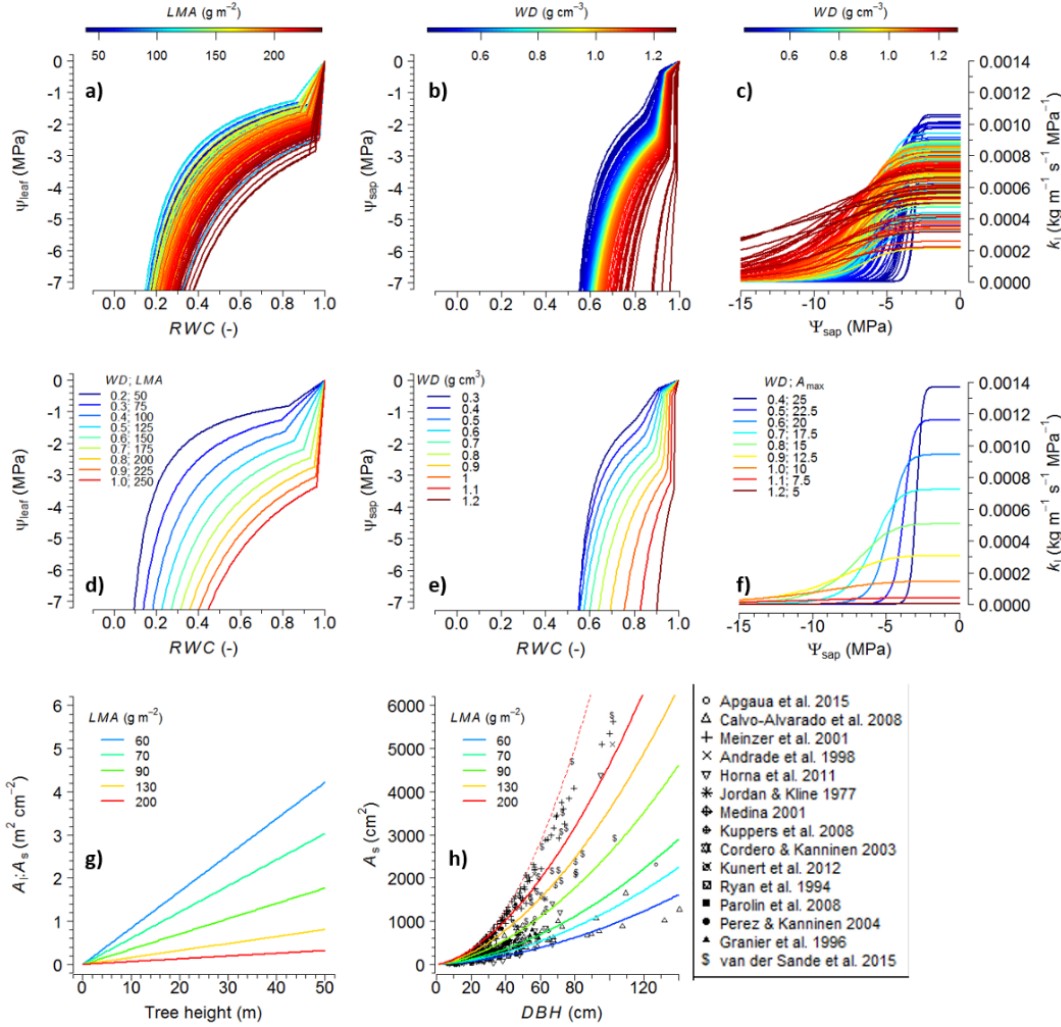

**Figure 7**. Plant hydraulics model parameterization as a function of plant traits (*LMA*, *WD*, $A_{max}$) and size (*H*) resulting from the syntheses presented in Figs. 2-6 and summarized in Table 2. **a), b)** Leaf and sapwood PV curves, respectively (Eqns 1-3) which relate tissue water potential to relative water content (RWC). **c)** sapwood xylem vulnerability curves (FLCx; Eqn 4) multiplied by maximum leaf-specific xylem conductivity based on a dataset of the joint distribution of *WD* and $A_{max}$ in an eastern Amazonian forest. Note that curves in a) and c) are also dependent on *WD* and $A_{max}$, respectively, in addition to the color scale shown. **d)-f)** same as a)-c) except for an idealized distribution of plant traits in which trait variation occurs over a single axis (*LMA* ~ *WD* ~ $A_{max}$). **g)** $A_l{:}A_s$ as a function of *LMA* and tree height. **h)** Sapwood area ($A_s$) as a function of *DBH* and *LMA*, overlain with synthesis of independent measurements of $A_s$. Dashed red line in h) represents theoretical maximum for $A_s$ (entirely sapwood).




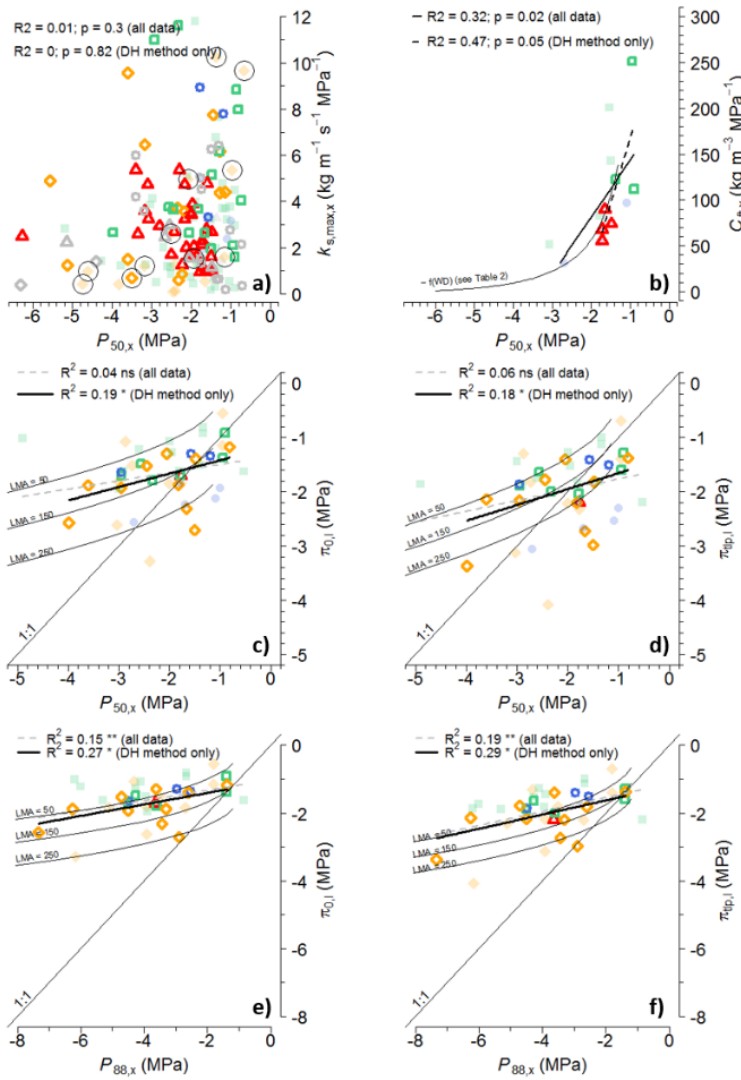

**Figure 8**. Tropical synthesis of trade-offs or coordination of various hydraulic traits with the water potential at 50% or 88% loss of maximum xylem conductivity ($P_{50,x}$ or $P_{88,x}$). **a)** the relationship between maximum xylem conductivity $k_{s,max,x}$ and $P_{50,x}$ does not support the notion of a trade-off between xylem efficiency and safety, except in tropical dry forests (circled points). **b)** The relationship between xylem capacitance ($C_{ft}$) and $P_{50}$ provides tentative evidence for a trade-off between between drought avoidance and tolerance. **c)-f)** Relationships of the leaf osmotic potential at full turgor ($\pi_{o,l}$) and the leaf turgor loss point ($\pi_{tlp,l}$) with $P_{88,x}$ support the hypothesis of coordination between leaf and xylem drought resistance, with less support for the hypothesis in terms of $P_{50,x}$. Thin lines in b)-f) correspond to the empirical equations in Table 2 over a wide range of input *WD* [0.2, 1.2] and *LMA* [50,250]. Symbols as in Fig. 4. See Supplement Fig. S2.3 for a version of b) in which no correction factor was applied to sapwood PV curves. Asterisk codes for significance as in Fig. 2.



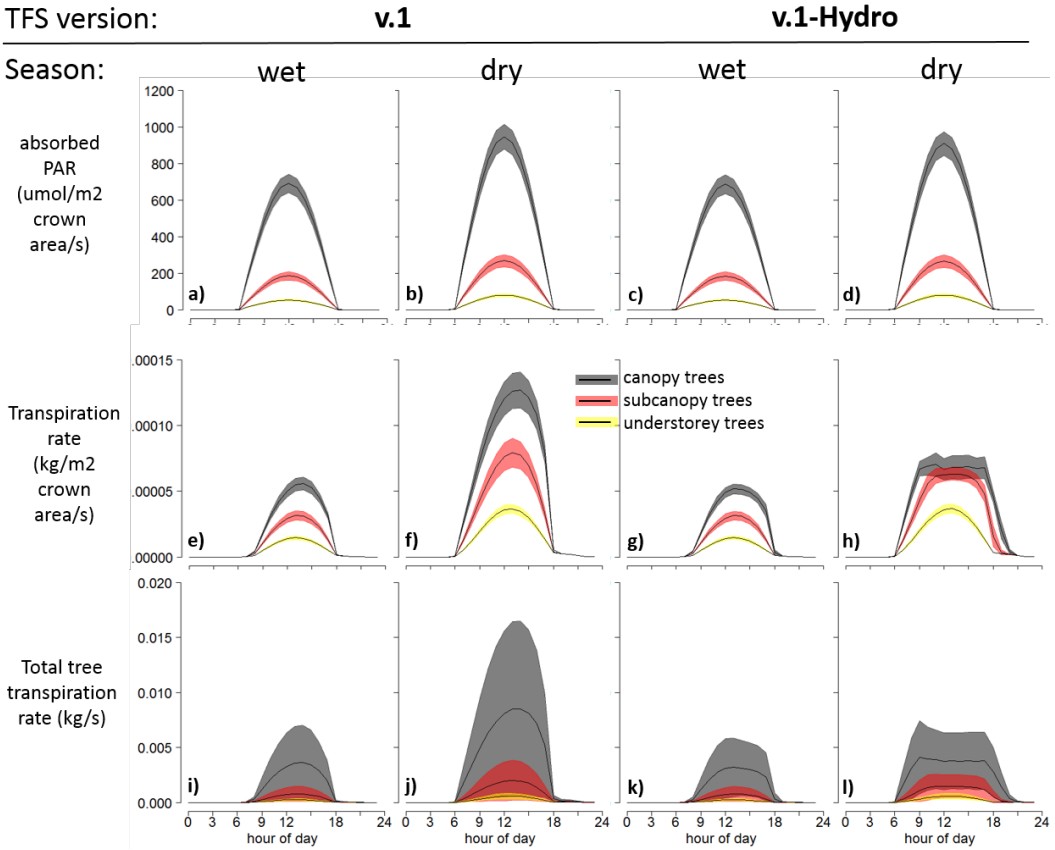

**Figure 9**. TFS simulated mean (+/- 1 s.d.) diurnal cycles across all individuals by canopy position, in wet and dry seasons with (v.1-Hydro) and without (v.1) plant hydraulics implemented, at Caxiuana, Brazil. **a)-d)** total absorbed photosynthetically active radiation, **e)-h)** transpiration rate per unit crown area, **i)-l)** total tree transpiration rate.





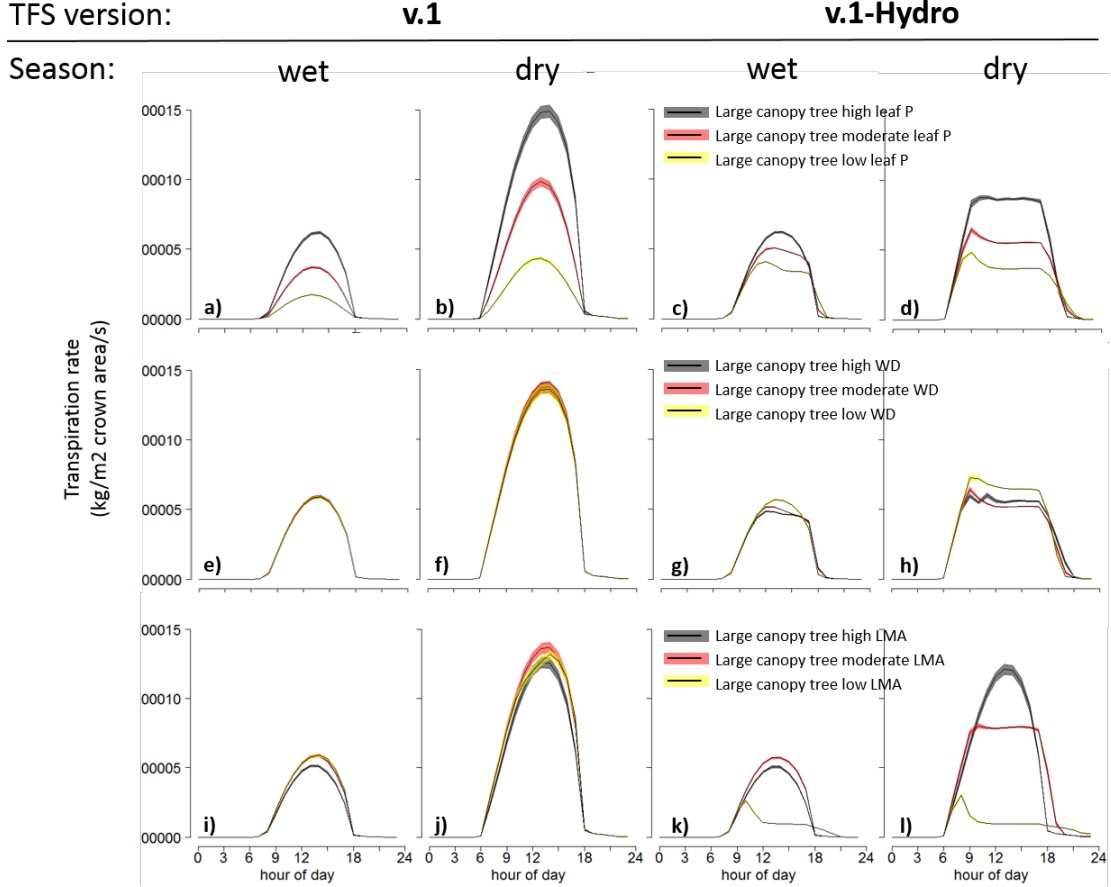

**Figure 10**. TFS simulated mean (+/- 1 s.e.) diurnal cycles of transpiration of three large (50-55 cm DBH) individual trees receiving full sunlight but differing in one plant trait, stratified by wet/dry season and TFS version with (v.1-Hydro) and without (v.1) plant hydraulics implemented, at Caxiuana, Brazil. **a)-d)** individuals differing in leaf P exhibit different $A_{max}$ and consequently $g_{s,max}$ and (in v.1-Hydro) $k_{s,max}$. **e)-h)** individuals differing in WD exhibit no difference in maximum transpiration rate per unit crown area in v.1, but different rates in v.1-Hydro emerge due to differences in xylem safety ($P_{50,x}$) and hence, stomatal control ($P_{50,gs}$). **i)-l)** individuals differing in LMA exhibit little difference in v.1, but large differences in v.1-Hydro due to large differences in $A_l$:$A_s$, and hence, total aboveground conductance ($K_{max,ag}$).




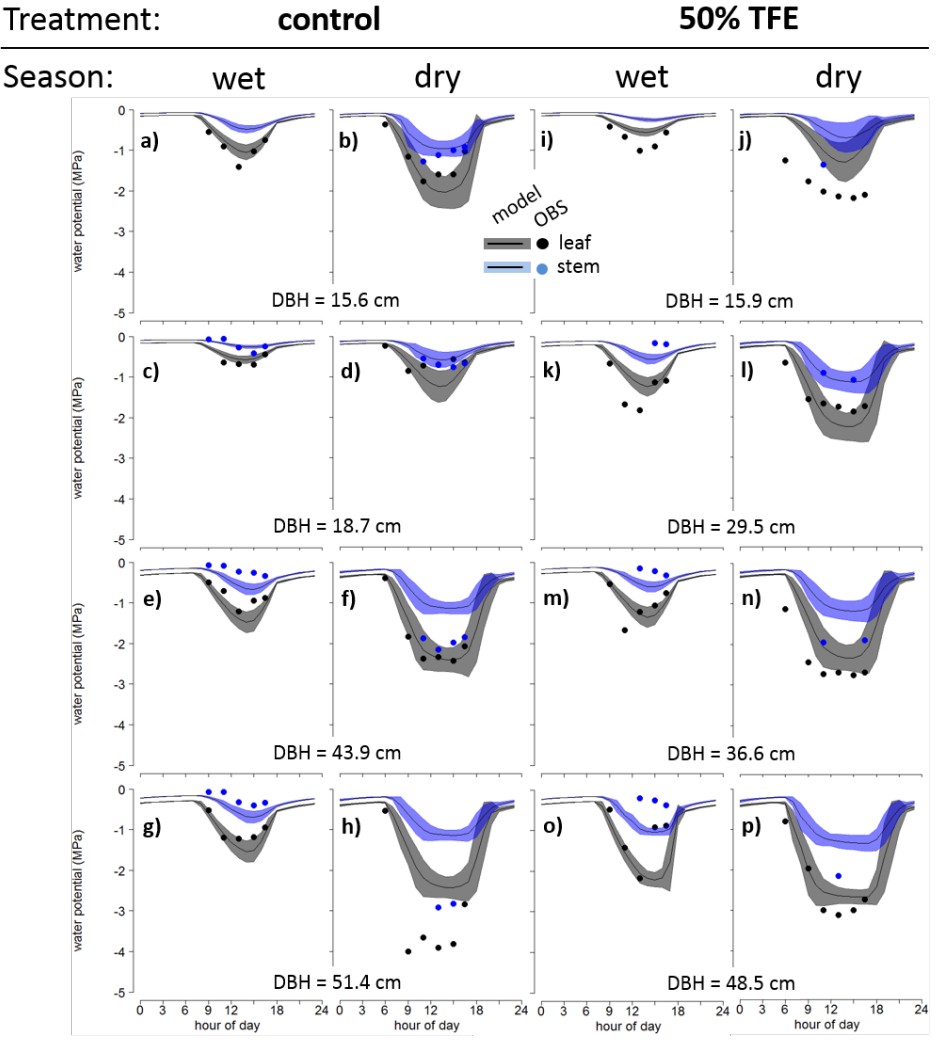

**Figure 11**. Simulated and observed diurnal variation in leaf and stem water potential for individual trees subject to seasonal and experimental variation in water availability at Caxiuana National Forest, Brazil. Simulated trees possess identical trait values equal to the plot mean and were matched with observed trees based on size (DBH). Trees are in order from smallest to largest, top to bottom, and are the following trees given in Fisher et al. (2006): **a)-b)** tree C1, **c)-d)** tree C2; **e)-f)** tree C4; **g)-h)** tree C3; **i)-j)** tree T1; **k)-l)** tree T2; **m)-n)** tree T3; **o)-p)** tree T4.





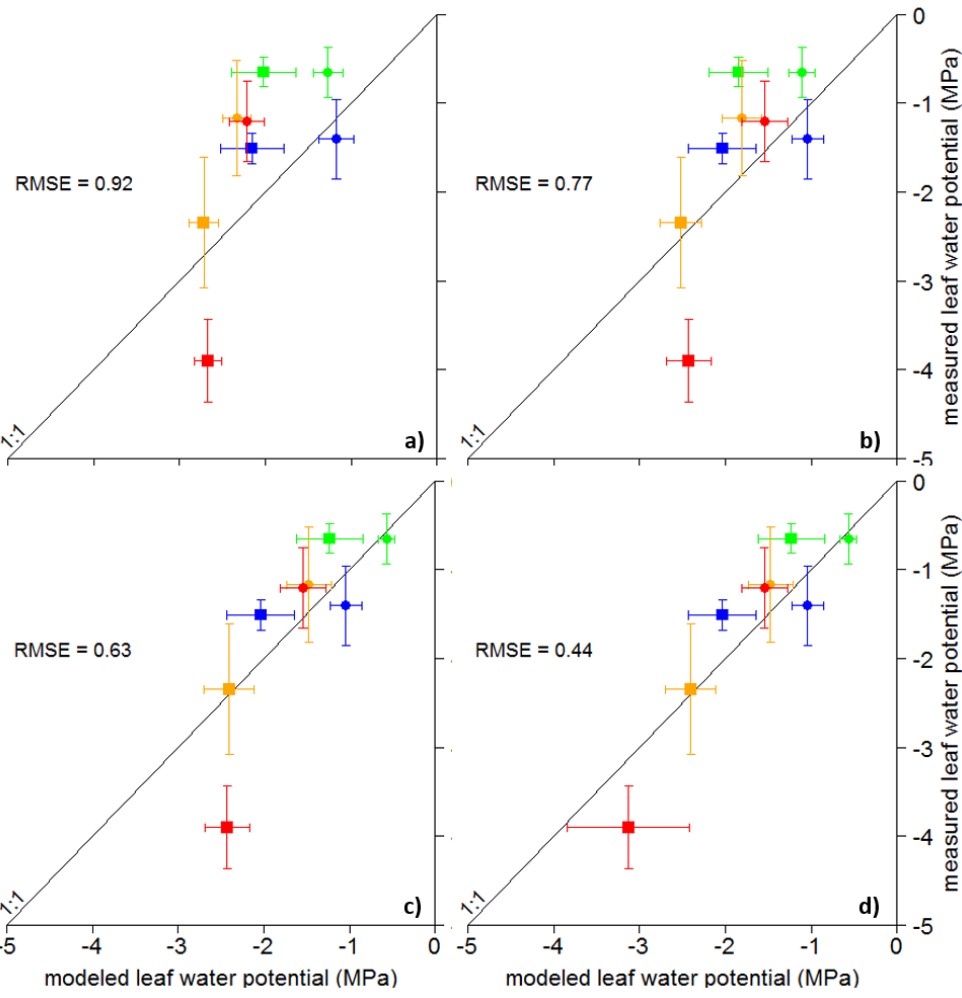

**Figure 12**. Simulated vs. observed individual-level variation in midday leaf water potential,
Caxiuana National Forest, Brazil.  **a)** without accounting for xylem taper and using mean plot
values for input plant traits and derived hydraulic traits from Table 2 (Savage et al. (2010) taper
exponent $p = 0.0$, $LMA = 96.1$ g m$^{-2}$, $WD = 0.73$ g cm$^{-3}$, $N_L = 20.9$ mg g$^{-1}$, $P_L = 0.59$ mg g$^{-1}$). **b)**
as in a) but accounting for xylem taper ($p = 1/3$). **c)** as in b) but after modifying canopy position
of individual trees (see Methods). **d)** as in c) but forcing tree C3 to be less isohydric by setting
$P50_{gs} = -5.0$ MPa ($P50_x$ remained the same at -3.2 MPa).  Different colors represent the
following four individuals from Fisher et al. (2006): green – tree C2 (18.7 cm DBH), blue – tree
C1 (15.6 cm DBH), orange – tree C4 (43.9 cm DBH), red – tree C3 (51.5 cm DBH). Filled
circles and squares represent wet and dry season values, respectively.





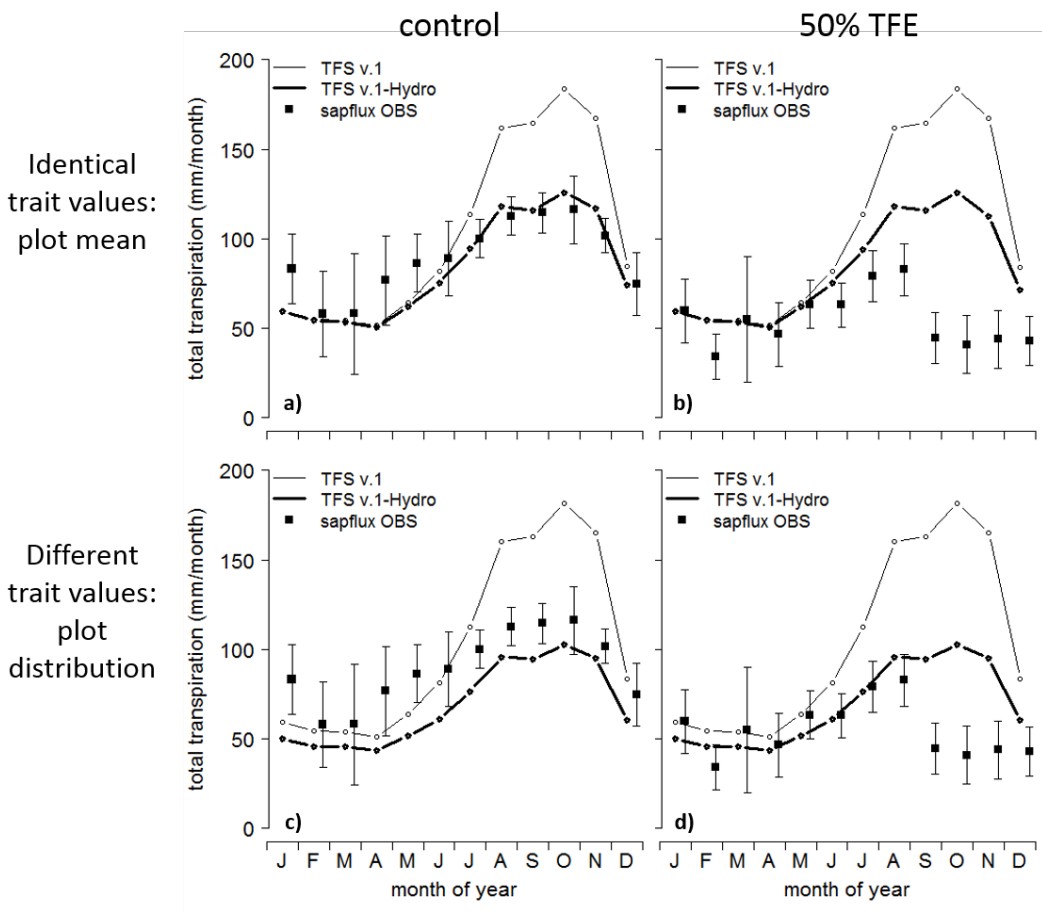

72

**Figure 13**. Simulated and observed stand-level total transpiration at Caxiuana National Forest, Brazil under a control (**a**), **c**)) and a 50% throughfall exclusion experiment (TFE; **b**), **d**)).  Model simulations were carried out where individuals were either **a**), **b**) forced to take on identical plot-mean trait values ($LMA$ = 96.1 g m$^{-2}$, $WD$ = 0.73 g cm$^{-3}$, $N_L$ = 20.9 mg g$^{-1}$, $P_L$ = 0.59 mg g$^{-1}$), or **c**), **d**) allowed to vary according to the plot-level trait distribution.