# Peer review of "Linking hydraulic traits to tropical forest function in a sizestructured and trait-driven model (TFS v.1-Hydro)"

_Geoscientific Model Development, 2016_

## Author Comment (AC1) · 15 Jun 2016

We inadvertently left out of the .zip file our two pdfs SI1 and SI2-4 which give the full technical description of the model, and supporting figures and data information, respectively. They are now both included in the updated .zip file.

Please also note the supplement to this comment:
http://www.geosci-model-dev-discuss.net/gmd-2016-128/gmd-2016-128-AC1-supplement.zip

---

## Referee Comment (RC1) · Anonymous Referee #1 · 12 Jul 2016

This paper seems a mine of parameters for land surface process models and resources for generating the parameters. My feeling is that the author(s) made a reasonable effort for conducting this research, and further, this paper should be published in GMD as soon as possible because I believe this paper would be of extreme value for developers and users of the land surface process and terrestrial ecosystem dynamics models. However, there are some small flaws in the present manuscript, and so before this paper is accepted, the author(s) must revise the manuscript according to the followings:

When an abbreviation/a symbol appears for the first time, write it out in full spelling: What is TFS (P1L33, P6L20), WD, LMA, NL and PL (P6L21)?

I suggest in Introduction section, you should state the temporal (apparently, ∼hourly

scale) and spatial (apparently single-plot/stand scale) scale of the model. This might be "readers-friendly". Further, you elaborated a plant hydraulic submodel in this research. As you note, solving Richards Equation is tough and needs heavy computation resourse. I do not think such plant hydraulic models are suitable for large-scale and long-term vegetation dynamics models. Thus, I also suggest in the Introduction section you should explicitly state why you elaborated the hydraulic models ("because such models can describe detailed plant water relations" is not enough) and future strategy of applications of the hydraulic models.

Concerning to P3L10, you need to read: Kumagai, T., Porporato, A. (2012) Drought-induced mortality of a Bornean tropical rainforest amplified by climate change. Journal of Geophysical Research -Biogeosciences, 117, G02032, doi:10.1029/2011JG001835.

Concerning to P4L3-7, you need to mention this pioneer paper: Kumagai, T. (2001) Modeling water transportation and storage in sapwood -model development and validation. Agricultural and Forest Meteorology, 109, 105-115. And "Arbogast et al., 1993" should be inserted in the former array of references P4L6-7.

Fugure 1: Subfigures involved in Fig 1 are incomprehensible and seem unnecessary because they have no explanation. Provide them with appropriate explanations if you want them to be involved in Fig 1.

P8L18: ">=" should be "ïĆş".

P10L13: "(-)" should be omitted or changed to "(unitless)".

P10L14: "-1" should be superscripts, and "FMC" should be "FMCx".

P10L15: "1" should not be italic.

P10L23: Why is "relative to" italic?

P11L13: Remove "derive".

P14L13: Insert "statistical" between "All" and "analyses". "R" should be "R software"

[Figure]

and "(R core team 2015)" is not enough information.

P14L14-15: You should not state this text "In all . . . . . . '****'." here. You should refer to statistical significance at each figure's caption (Figs 2-6, and 8).

3.2 Model setup: Give more detailed throughfall exclusion experiment (TFE) or remove all statements on the TFE in the manuscript (I will explain later).

P18L4: 50% reduction of what?

Captions of Figures 2 and 10: Give full spellings of the abbreviations such as LMA, WD, and so forth.

Figures 5, 6 and 8: Legends of symbols are needed.

Caption of Figure 6: Al:As needs to be italic and subscripts.

Caption of Figure 8: "ks,max,x" needs brackets.

4.2.1 Impact of plant. . . . . ...: Add further and more detailed explanation on why transpiration rate was higher in dry season than in wet season for both TFSs and why transpiration rate in dry season was inhibited with v.1-Hydro.

Figure 11: Mention clearly which type of TFS was used for this simulation in the caption. Further, you have to note that this analysis in Fig 11 cannot be any validation for the model because there was no difference in both observations and computations between control and 50%TFE. I recommend to omit all statements on both the TFE observations and computations (further recommendation and explanation later).

Figure 13 and 4.3.3 Fidelity of modeled. . . . . . . ...: Did you mean the simple soil bucket models of both TFSs v1 and v1-Hydro could not reproduce the temporal variation in soil moisture for the Control and the 50%TFE? If so, this is very critical problem because simulations of plant water relations such as stomatal behavior, sapflow, root water uptake and so forth must be conducted on the premises that the soil moisture environment is appropriately reproduced. In this case, I think discussion on simulation

of TFE would ruin this paper. So, in this case, I suggest omitting all statements about TFE in this paper. Note that without the TFE, the value of this paper would not change. If you are successful in reproducing soil moisture environment for both the Control and the TFE, show the time variations in modeled and simulated soil moisture and give more detailed and analytical explanation about why both TFSs v1 and v1-Hydro could not capture the observed reduction in transpiration under the TFE.

P25L24-25: How thinking plant hydraulic submodels important leads to developing ESM? Please give a concrete explanation.

P29L3: (If the models failed to reproduce the soil moisture environment) I guess "not vertically discretize the soil water or root distribution" is not big problem. This is simply problem caused by failures in mass balance equations because the models could not capture the soil moisture depletion induced from 50% rainwater reduction.

Conclusion: You should confess this paper's current model is too complex to incorporate it to coarse-scale DGVMs.

---

## Referee Comment (RC2) · Anonymous Referee #2 · 16 Aug 2016

This paper presents a plant hydraulic model and the parameterization of this model with plant traits of tropical forest trees. The study is interesting and in the scope of the journal Geoscientific Model Development. The hydraulic model itself is based on the model proposed by Sperry et al. (1998). But the parameterization with plant traits is comprehensive and valuable for model development, especially for modeling tropical forests. The equations in Supplement S1 have well explained the formulations of the model. And the figures and the equations in Table 2 are presenting the results well.

I have to say that I had a hard time reading this paper. I went through this paper many time in the past weeks and still didn't well get it. Fortunately, the equations in Supplemental material and Table 2 are clear and the figures of results are readable. I

can see it is a good work. But the writing of this paper should be substantially improved in its revised version.

Another concern is the interactions between the hydraulics model and the host model TFS. The authors only show that the hydraulics model makes the hourly predictions of water dynamics better (e.g., transpiration, water potential, etc.) at given forest structure in their results. But, how the hydraulics model affects the long-term predictions of TFS (decades to a century)? I ask this question because a plant hydraulics model may change the behavior of trees in competition and therefore change the long-term predictions of forest dynamics. I want to know to what extent it changes the host model (i.e., TFS).

Minor suggestions:

1. The section of "introduction" : I didn't get it why it is necessary to build this model and why this way works here from this section. I hope the authors can write a better introduction to make it easier to understand in a revised version.

2. In page 4, lines 3∼6 "Other models treat the plant continuum as a porous medium with constitutive equations defining water retention properties (the relationship between water potential and water content) and xylem PLC, using Darcy's law to incorporate fluxes 5 within the Richards' mass balance equation". I think this sentence is important because it describes what other models do. But please make this sentence clear.

3. Pages 4∼5, from line 10 in page 4 to line 30 in page 5: These two big paragraphs have a lot of facts and arguments. But they are too messy. I have read through them many times, trying to figure out the messages that the authors want to deliver. But, I still do not get them.

4. Page 7, lines 9∼10 "... the model developed by J.S. Sperry and described in Sperry et al. (1998)" can be shortened as " ... the model developed by Sperry et al. (1998)".

5. Page 7, line 21 "we modified S98 in three important ways ...". I prefer to say "we

modified S98 in three ways ..." by crossing out "important". I understand that these modifications are important. But here it's a description of the model and you don't have to evaluate your works here.

6. Page 8 lines 1∼12: This paragraph should be a summary of the model, not just what have been described in Supplement S1. The authors should expect the readers to get a picture of the model by their descriptions without reading S1.

7. Page 8, lines 22∼23: It took me a while to think what the authors want to tell in this sentence. If the authors just want to talk about "capillary water", I prefer a sentence like "sapwood also stores capillary water in its void spaces and embolized conduits". Then, I don't have to think about "tension theory".

8. Page 8 lines 23∼30: These arguments are not necessary because this section is to describe the model. And, the sentences in lines 14∼23 can be reorganized so that it's easier for readers to understand Eqn 1.

9. Page 9, line 6: "RWC" is not explained.

10. Page 10, line 13 "(-)": Does it mean ax is negative? If yes, I prefer to use "-ax" in the equation.

11. Page 11, lines 2∼13: This paragraph is supposed to describe some "first princi-ples" of the size effect of trees on plant hydraulics according to the second paragraph, but where are they? I saw "two main mechanisms". The second one describes two possibilities. Which one should be the "first principle" in the model?

12. Page 11, line 10: "because of the Hagen-Poiseuille law". It's a phenomenon of the Hagen-Poiseuille law, not because of it.

13. Page 12, lines 20∼23: I think the sentence "FMCgs ... is the only variable passed from the hydraulics module to the host model" is the message of this paragraph and therefore should be the first sentence.

14. Lines 23~26 in Pages 12 and lines 1~5 in page 13: These sentences can be moved to discussion. This section is to describe the model, it's not necessary to argue these issues here.

15. Page 13, line 28: Please also cite Strigul et al. 2008 for PPA.

16. Page 16, line 15: "Idealized model experiments". I would use "Model experiments" because any model experiments are always "idealized" somehow.

17. Page 17, lines 13~14 "We matched simulated trees . . ." It would be clearer if there is a table to show the settings of trees.

18. Figures 11 and 13: explain "50% TFE" in legends.

---

## Short Comment (SC1) · 18 Aug 2016

Referee #2, thank you for your thorough read and your comments. Our apologies that certain aspects of the paper were not easy to understand. I would like to request some further clarification about which parts of the paper were not readily understandable. It is very important to us that the revised version of the paper is clear and concise without leaving out important information.

From your comment's "Minor suggestions", the main areas you have highlighted that generate confusion are: - the motivation for the study and the particular type of model we are advocating (your #1) - the empirical evidence and arguments flowing from it in the intro (your #3) - too much detail given in the Tissue water relations section 2.1.2

(your #7)

My question is if the three bullets I have listed above encompass everything which you found difficult to understand. Are there either additional sections you found difficult to understand? Are there any further specifics about the bullets above which made these parts of the paper unclear?

————————————————————

---

## Author Response (AR1)

**General comment to the reviewer**: Thank you for taking the time to thoroughly read our manuscript and for your constructive comments, which have greatly improved the clarity and accessibility of our manuscript. Your original comments are highlighted in red and our responses are in black. When text is copied directly from the revised paper the words are italicized.

This paper seems a mine of parameters for land surface process models and resources for generating the parameters. My feeling is that the author(s) made a reasonable effort for conducting this research, and further, this paper should be published in GMD as soon as possible because I believe this paper would be of extreme value for developers and users of the land surface process and terrestrial ecosystem dynamics models.

Thank you for recognizing the value of our research, and again for reading our paper thoroughly.

However, there are some small flaws in the present manuscript, and so before this paper is accepted, the author(s) must revise the manuscript according to the followings: When an abbreviation/a symbol appears for the first time, write it out in full spelling: What is TFS (P1L33, P6L20), WD, LMA, NL and PL (P6L21)?

We have ensured that all acronyms are defined the first time they are used in the manuscript.

I suggest in Introduction section, you should state the temporal (apparently, hourly scale) and spatial (apparently single-plot/stand scale) scale of the model. This might be "readers-friendly".

We added two sentences in the revised introduction (P4 L4-8) to clarify this point:
*In this paper, we develop a continuous porous media approach intended for application at specific sites in the tropics to explore dynamics of water fluxes from hourly to seasonal timescales and at spatial scales ranging from individual trees to the stand-level scale. This intermediate-scale approach is a model testbed meant to inform implementations of plant hydraulics in coarse-scale forest ecosystem models.*

Prior to this insertion in the Introduction, we also clarified the scale of the various modeling approaches that are discussed (coarse-scale ecosystem models and fine-scale hydraulics models; see P3 L14 and P3 L27).

Further, you elaborated a plant hydraulic submodel in this research. As you note, solving Richards Equation is tough and needs heavy computation resourse. I do not think such plant hydraulic models are suitable for large-scale and long-term vegetation dynamics models. Thus, I also suggest in the Introduction section you should explicitly state why you elaborated the hydraulic models ("because such models can describe detailed plant water relations" is not enough) and future strategy of applications of the hydraulic models.

We have expanded the section outlining the various approaches to doing plant hydraulics (P3 L27-P5L8), in order to highlight why we elaborated (what we have now termed) the "continuous porous media" approach:

*a range of approaches exist for modeling plant hydraulics at fine scales (i.e., individual trees), all involve an extension of Darcy's law (Darcy, 1856) from the soil domain to include plants as well. Darcy's law states that water flux anywhere in the soil-plant continuum is proportional to the product of soil or plant hydraulic conductivity and a gradient in water potential. In order for these models to capture drought response, hydraulic conductivity within the soil-plant continuum must dynamically respond to changes in moisture. Three main approaches are distinguished in terms of how they represent the impact of declines in water potential on tissue water content and xylem hydraulic conductivity. A first class of models is the simplest and simulates moisture sensitivity of soil-root conductance but not xylem hydraulic conductance (Jarvis et al., 1981; Williams et al., 1996; Ogée et al., 2003; Alton et al., 2009; Bonan et al., 2014). This approach has proven useful for modeling the effects experimental drought in tropical forests (Williams et al., 1998; Fisher et al., 2006; Fisher et al., 2007), but it remains unclear whether this approach misattributes drought effects occurring within trees to the soil; therefore a second class of models implements variable xylem conductivity with xylem water potential (Williams et al., 2001; Hickler et al., 2006; Domec et al., 2012; Duursma and Medlyn, 2012; Xu et al., 2016). To simplify computational load, these two approaches do not explicitly track dynamic changes in the volume of plant water storage. Instead, a constant ratio of change in stored water per unit change in water potential, or stem hydraulic capacitance, is assumed, which may overestimate the buffering capacity of tree stored water under extreme drought conditions when small relative declines in stored water induce very large declines in water potential. An additional consequence of the design of these models is the inability to represent the bidirectional flow of water at the root-soil interface. Reverse flow of water from roots into soil is an important process in root hydraulic distribution (Oliveira et al., 2005), and may also mediate time to desiccation under drought (North and Nobel, 1997).*

*A convenient way to address these issues is in a third class of models (hereafter the "continuous porous media approach"), which simply extend the modeled mass balance of water from the soil domain into the plant by relating simulated changes in water content to water potential (and vice versa) everywhere within the plant-soil continuum (Edwards et al., 1986; Arbogast et al., 1993; Sperry et al., 1998; Kumagai, 2001; Bohrer et al., 2005; Mackay et al., 2015; Mirfenderesgi et al., 2016). While more computationally complex, the continuous porous media approach offers two main advantages in addition to addressing the issue of plant water storage and bidirectional root flow. First, the coupled plant-soil system is represented by a single mass balance equation, such that root water uptake or loss simply emerges from the solution of this equation, and needs not be ascribed post-hoc as is the case in the first two approaches. Second, this approach relies on an explicit description of the relationship between water content and water potential in plant xylem (the "pressure-volume", or PV curve), analogous to the water retention curves used in soil physics. As we will show, there is a wealth of information on PV hydraulic traits for leaves, and to a lesser degree, stems in tropical forests. Implementing PV curves in the model greatly increases the scope of data with which the model can be parameterized. In this paper, we develop a continuous porous media approach intended for application at specific sites in the tropics to explore dynamics of water fluxes from hourly to seasonal timescales and at spatial scales ranging from individual trees to the stand-level scale. This intermediate-scale approach is*

*a model testbed meant to inform implementations of plant hydraulics in coarse-scale forest ecosystem models.*

Concerning to P3L10, you need to read: Kumagai, T., Porporato, A. (2012) Drought-induced mortality of a Bornean tropical rainforest amplified by climate change. Journal of Geophysical Research -Biogeosciences, 117, G02032, doi:10.1029/2011JG001835.

We found this paper particularly relevant (thank you!) and have cited it in the place you suggested. We anticipate returning to this paper when considering deriving mortality functions from our plant hydraulics model.

Concerning to P4L3-7, you need to mention this pioneer paper: Kumagai, T. (2001) Modeling water transportation and storage in sapwood -model development and validation. Agricultural and Forest Meteorology, 109, 105-115. And "Arbogast et al., 1993" should be inserted in the former array of references P4L6-7.

This citation has now been included.

Fugure 1: Subfigures involved in Fig 1 are incomprehensible and seem unnecessary because they have no explanation. Provide them with appropriate explanations if you want them to be involved in Fig 1.

The figure caption has now been updated to explain the various other sub-panels in this figure:
*Left sub-panel: inputs are plot observations of the tree size distribution and distributions of each of four plant functional traits (WD, LMA, $N_L$, $P_L$). Middle bottom: the perfect plasticity approximation (PPA) orders trees by decreasing crown area. Middle top: TFS assigns physiological traits, such as $V_{c,max}$, to each tree based on functional traits. Middle: the PPA is used to estimate light environments of each tree, which influence fast timescale (hourly) biophysics of crown light interception, photosynthesis, and stomatal conductance. Assimilated carbon is allocated to leaves, stems, and fine roots daily. Top right (hydrodynamics): Soil-root-stem-canopy water fluxes interact with fast-timescale TFS biophysics by taking transpiration as a boundary condition and passing back the 'fraction of maximum conductance' ($FMC_{gs}$) for downregulating the next timestep's stomatal conductance based on leaf water potential. Top right (size- and trait-scaling): Hydraulic traits in leaves, stem, and roots are assigned based on each tree's height and trait values according to empirical equations and allometric theory described in this paper. Bottom right: TFS predicts a distribution of individual tree net primary productivities. See Supplement Fig S1.1 for the structure of the plant hydraulics model.*

P8L18: ">=" should be "ï´C ¸s".

Assuming the pdf conversion of your comment meant to say "≥", we updated the symbol accordingly

P10L13: "(-)" should be omitted or changed to "(unitless)".

Done.

P10L14: "-1" should be superscripts, and "FMC" should be "FMCx".

Done.

P10L15: "1" should not be italic.

Done.

P10L23: Why is "relative to" italic?

These italics have been removed

P11L13: Remove "derive".

Done.

P14L13: Insert "statistical" between "All" and "analyses". "R" should be "R software" and "(R core team 2015)" is not enough information.

"statistical" was inserted and "R" was changed to "R language," but we left the citation the same, as this is the suggested format for citing the R statistical program.

P14L14-15: You should not state this text "In all : : :: : : '****'." here. You should refer to statistical significance at each figure's caption (Figs 2-6, and 8).

This sentence has been deleted and the reference to statistical significance remains in the figure captions.

3.2 Model setup: Give more detailed throughfall exclusion experiment (TFE) or remove all statements on the TFE in the manuscript (I will explain later).

We have removed all analyses and references to the TFE per your suggestion (further explanation given below)

P18L4: 50% reduction of what?

Removed.

Captions of Figures 2 and 10: Give full spellings of the abbreviations such as LMA, WD, and so forth.

Done.

Figures 5, 6 and 8: Legends of symbols are needed.

Done.

Caption of Figure 6: Al:As needs to be italic and subscripts.

Done.

Caption of Figure 8: "ks,max,x" needs brackets.

Done.

4.2.1 Impact of plant…: Add further and more detailed explanation on why transpiration rate was higher in dry season than in wet season for both TFSs and why transpiration rate in dry season was inhibited with v.1-Hydro.

Done.  Revised text reads as follows, first on the seasonal differences (P22 L22-27):
*The large increase in simulated transpiration in the dry season for both TFS v.1 and TFS v.1-Hydro (Fig. 9e-f) is driven by the comparatively large increase in incoming solar radiation, and hence absorbed radiation, due to a reduction in cloud cover (Fig 9a-d) (Carswell, 2002; Fisher et al., 2007).  At the level of individual tree crowns, canopy position is the dominant control over the amount of absorbed PAR (compare different colored lines in Fig 9a-d).*

and on why dry season transpiration was inhibited with v.1-Hydro (P23 L1-4):
*The effect of including plant hydraulics in TFS v.1-Hydro was to limit late morning and afternoon transpiration via the depletion of stored water within the canopy and tree stem, which caused midday declines in leaf water potential (see Fig 10) and induced hydraulic limitation to water flux via the $FMC_{gs}$ term (see Eqn 8).*

Figure 11: Mention clearly which type of TFS was used for this simulation in the caption. Further, you have to note that this analysis in Fig 11 cannot be any validation for the model because there was no difference in both observations and computations between control and 50%TFE. I recommend to omit all statements on both the TFE observations and computations (further recommendation and explanation later).

Done. As explained below and per your suggestion, all analyses and references to TFE have been removed.

Figure 13 and 4.3.3 Fidelity of modeled …: Did you mean the simple soil bucket models of both TFSs v1 and v1-Hydro could not reproduce the temporal variation in soil moisture for the Control and the 50%TFE? If so, this is very critical problem because simulations of plant water relations such as stomatal behavior, sapflow, root water uptake and so forth must be conducted on the premises that the soil moisture environment is appropriately reproduced. In this case, I think discussion on simulation of TFE would ruin this paper. So, in this case, I suggest omitting all statements about TFE in this paper. Note that without the TFE, the value of this paper would not change. If you are successful in reproducing soil moisture environment for both the Control and the TFE, show the time variations in modeled and simulated soil moisture and give more

detailed and analytical explanation about why both TFSs v1 and v1-Hydro could not capture the observed reduction in transpiration under the TFE.

We agree on your suggestion to remove the TFE parts of this paper, because the current bucket soil hydrology is not amenable to simulating drought. We also revised a couple sentences (P8 L1-4) highlighting this current belowground structural deficiency for future development:
*The present scheme does not consider the vertical distribution of soil water or roots. We anticipate this to be a key component for future model development when we incorporate this scheme into host models with variable soil depths.*

P25L24-25: How thinking plant hydraulic submodels important leads to developing ESM? Please give a concrete explanation.

We have added a clause to the final sentence of this paragraph to this effect (P26 L26-27):
*Finally, the model makes substantial improvements to TFS v.1 in terms of simulated transpiration rates (Fig 13a, b), highlighting how plant hydraulics mediate the biosphere-atmosphere exchange of carbon and water.*

P29L3: (If the models failed to reproduce the soil moisture environment) I guess "not vertically discretize the soil water or root distribution" is not big problem. This is simply problem caused by failures in mass balance equations because the models could not capture the soil moisture depletion induced from 50% rainwater reduction.

The inability of the model to reproduce soil drought conditions is not related to the failure of the model to do soil water mass balance; all precipitation and evaporation is accounted for via changes in the total stored water in the system. Rather, it is a direct consequence of using a soil depth of 4 meters at this site and allowing roots to exploit this entire bucket. This soil bucket does not sufficiently deplete under the simulated drought conditions (in default TFS v.1 nor in TFS v1.-Hydro) because it represents an average soil moisture over the entire 4-meter domain. In contrast, a vertically stratified water balance model that accurately reproduces observed TFE soil moisture response at this site would, according to the observations (see Rowland et al. 2015 Extended Data Figure 2), have the largest a large TFE effect in shallow layers where most roots would be concentrated (see Fisher et al. 2007 Figure 6).

Conclusion: You should confess this paper's current model is too complex to incorporate it to coarse-scale DGVMs.

Done, although future work will need to identify where simplification is necessary and where complexity is still needed. The final sentence of the paper (P33 L5-8) highlights this:
*Likely some degree of simplification of the present approach will be required upon implementation in ESMs; nonetheless we expect that inclusion of trait-driven plant hydraulics*

*schemes will lead to reduced uncertainty in the future state of tropical forests under climate change.*

**References**

Fisher, R. A., Williams, M., da Costa, A. L., Malhi, Y., da Costa, R. F., Almeida, S., and Meir, P.: The response of an Eastern Amazonian rain forest to drought stress: results and modelling analyses from a throughfall exclusion experiment, Global Change Biology, 13, 2361-2378, 2007.

Rowland, L., da Costa, A. C., Galbraith, D. R., Oliveira, R. S., Binks, O. J., Oliveira, A. A., Pullen, A. M., Doughty, C. E., Metcalfe, D. B., Vasconcelos, S. S., Ferreira, L. V., Malhi, Y., Grace, J., Mencuccini, M., and Meir, P.: Death from drought in tropical forests is triggered by hydraulics not carbon starvation, Nature, 528, 119-122, 2015.

**General comment to the reviewer**: Thank you for taking the time to thoroughly read our manuscript and in particular, for highlighting the key parts of the paper which were unclear. We believe that our revisions make a substantial improvement to the accessibility of the description of the model and why we pursued our particular approach to a broad audience. Below, your original comments are highlighted in red and our responses are in black. When text is copied directly from the revised paper the words are italicized.

This paper presents a plant hydraulic model and the parameterization of this model with plant traits of tropical forest trees. The study is interesting and in the scope of the journal Geoscientific Model Development. The hydraulic model itself is based on the model proposed by Sperry et al. (1998). But the parameterization with plant traits is comprehensive and valuable for model development, especially for modeling tropical forests. The equations in Supplement S1 have well explained the formulations of the model. And the figures and the equations in Table 2 are presenting the results well.

Thank you for recognizing the value of our research.

I have to say that I had a hard time reading this paper. I went through this paper many time in the past weeks and still didn't well get it. Fortunately, the equations in Supplemental material and Table 2 are clear and the figures of results are readable. I can see it is a good work. But the writing of this paper should be substantially improved in its revised version.

We made substantial changes in four main places of the manuscript where the model is either introduced or described: The introduction (1.), the model overview (2.1.1), the tissue water relations (2.1.2), and scaling conductance with tree size (2.1.4). Several other minor modifications to text were also made. We describe these changes in detail in our response to your "Minor suggestions" #1, #2, #3 and #6 below.

Another concern is the interactions between the hydraulics model and the host model TFS. The authors only show that the hydraulics model makes the hourly predictions of water dynamics better (e.g., transpiration, water potential, etc.) at given forest structure in their results. But, how the hydraulics model affects the long-term predictions of TFS (decades to a century)? I ask this question because a plant hydraulics model may change the behavior of trees in competition and therefore change the long-term predictions of forest dynamics. I want to know to what extent it changes the host model (i.e., TFS).

We agree that the interactions between plant hydraulic traits and longer term demographic dynamics is a scientifically intriguing and important topic and is one we intend to pursue, but in future work. The current version of TFS we are using is not as of yet dynamic – mortality and recruitment processes are not yet fully developed. The model in its current form requires site-specific initialization and is intended to explore (and be validated by) various short-term demographically-structured outputs, such as productivity and (in the case of the hydraulics mode) water status of individual trees.

The validation exercises conducted by Fyllas et al. (2014) and this paper give us confidence to proceed with the analysis you suggest.  We highlight how the novel model capability described here fits within the broader scope of interactions between plant hydraulic traits and forest trait composition and dynamics in two new sentences at the end of the Discussion overview paragraph (page 26 line 27 – page 27 line 2):

*In sum, TFS v.1-Hydro represents a key and advanced model capability to represent differential performance of individual trees based on hydraulic traits, size and light environments (Figs 9-10). Future work coupling the present scheme with community dynamics (mortality, growth and recruitment) has the potential to predict shifts in community trait distributions under changing moisture regimes, as has been observed or implied in studies of tropical species distributions and forest community dynamics (Engelbrecht et al., 2007; Fauset et al., 2012).*

Minor suggestions:
1. The section of "introduction" : I didn't get it why it is necessary to build this model and why this way works here from this section. I hope the authors can write a better introduction to make it easier to understand in a revised version.

We have expanded the section outlining the various approaches to doing plant hydraulics (page 3 line 27 – page 5 line 8), in order to highlight why we elaborated (what we have now termed) the "continuous porous media" approach:

*While a range of approaches exist for modeling plant hydraulics at fine scales (i.e., individual trees), all involve an extension of Darcy's law (Darcy, 1856) from the soil domain to include plants as well.  Darcy's law states that water flux anywhere in the soil-plant continuum is proportional to the product of soil or plant hydraulic conductivity and a gradient in water potential. In order for these models to capture drought response, hydraulic conductivity within the soil-plant continuum must dynamically respond to changes in moisture. Three main approaches are distinguished in terms of how they represent the impact of declines in water potential on tissue water content and xylem hydraulic conductivity. A first class of models is the simplest and simulates moisture sensitivity of soil-root conductance but not xylem hydraulic conductance (Jarvis et al., 1981; Williams et al., 1996; Ogée et al., 2003; Alton et al., 2009; Bonan et al., 2014).  This approach has proven useful for modeling the effects experimental drought in tropical forests (Williams et al., 1998; Fisher et al., 2006; Fisher et al., 2007), but it remains unclear whether this approach misattributes drought effects occurring within trees to the soil; therefore a second class of models implements variable xylem conductivity with xylem water potential (Williams et al., 2001; Hickler et al., 2006; Domec et al., 2012; Duursma and Medlyn, 2012; Xu et al., 2016).  To simplify computational load, these two approaches do not explicitly track dynamic changes in the volume of plant water storage. Instead, a constant ratio of change in stored water per unit change in water potential, or stem hydraulic capacitance, is assumed, which may overestimate the buffering capacity of tree stored water under extreme drought conditions when small relative declines in stored water induce very large declines in water potential. An additional consequence of the design of these models is the inability to represent the bidirectional flow of water at the root-soil interface.  Reverse flow of water from roots into soil is an important process in root hydraulic distribution (Oliveira et al., 2005), and may also mediate time to desiccation under drought (North and Nobel, 1997).*

*A convenient way to address these issues is in a third class of models (hereafter the "continuous porous media approach"), which simply extend the modeled mass balance of water from the soil domain into the plant by relating simulated changes in water content to water potential (and vice versa) everywhere within the plant-soil continuum (Edwards et al., 1986; Arbogast et al., 1993; Sperry et al., 1998; Kumagai, 2001; Bohrer et al., 2005; Mackay et al., 2015; Mirfenderesgi et al., 2016). While more computationally complex, the continuous porous media approach offers two main advantages in addition to addressing the issue of plant water storage and bidirectional root flow. First, the coupled plant-soil system is represented by a single mass balance equation, such that root water uptake or loss simply emerges from the solution of this equation, and needs not be ascribed post-hoc as is the case in the first two approaches. Second, this approach relies on an explicit description of the relationship between water content and water potential in plant xylem (the "pressure-volume", or PV curve), analogous to the water retention curves used in soil physics. As we will show, there is a wealth of information on PV hydraulic traits for leaves, and to a lesser degree, stems in tropical forests. Implementing PV curves in the model greatly increases the scope of data with which the model can be parameterized. In this paper, we develop a continuous porous media approach intended for application at specific sites in the tropics to explore dynamics of water fluxes from hourly to seasonal timescales and at spatial scales ranging from individual trees to the stand-level scale. This intermediate-scale approach is a model testbed meant to inform implementations of plant hydraulics in coarse-scale forest ecosystem models.*

We have also better highlighted (page 5, line 28 – page 6, line 8) why plant hydraulics is important from a long-term demographic and trait-filtering perspective (which will give rise to the kinds of long-term investigations you highlight above):

*The implication for ecosystem models is that under-representation of diversity in functional traits and tree size in tropical forests is undermining efforts to make accurate projections of tropical forest response to climate. Model parameterization of hydraulic trait diversity should thus provide much-needed model capability to represent a diversity of responses to changes in moisture availability, laying the groundwork for representing trait-mediated differences in survival and subsequent shifts in forest trait composition. Shifts in trait composition are already occurring in some tropical forests (e.g., Enquist and Enquist, 2011; van der Sande et al., 2016), and such shifts (or the diversity of traits alone) have been shown to buffer ecosystems in the face of environmental change, and in some cases, are the difference between predicted complete loss of forest and forest persistence (Fauset et al., 2012; Levine et al., 2016; Sakschewski et al., 2016).*

2. In page 4, lines 36 "Other models treat the plant continuum as a porous medium with constitutive equations defining water retention properties (the relationship between water potential and water content) and xylem PLC, using Darcy's law to incorporate fluxes 5 within the Richards' mass balance equation". I think this sentence is important because it describes what other models do. But please make this sentence clear.

This description has been entirely re-worked and is given in the response to your comment immediately above

3. Pages 45, from line 10 in page 4 to line 30 in page 5: These two big paragraphs have a lot of facts and arguments. But they are too messy. I have read through them many times, trying to figure out the messages that the authors want to deliver. But, I still do not get them.

We have substantially restructured these paragraphs to highlight our important messages, by deleting some content and adding some clarifying sentences that link this section to the previous section summarizing modeling approaches, as follows (page 5, lines 9 – 15):

*Model parameterization leads us to the challenge of how to represent variation in plant hydraulic traits governing moisture sensitivity and water transport capacity. It has long been recognized that the functional trait diversity of tropical forests mirrors their large species diversity (Corner, 1949; Hallé et al., 1978; Leigh Jr, 1999), and diversity in plant hydraulic traits such as the water potential at turgor loss ($\pi_{tlp}$) and at 50% loss of conductivity ($P_{50}$), xylem-specific hydraulic conductivity ($k_{s,max}$) and the leaf-to-sapwood area ratio ($A_l : A_s$) is no exception (Borchert, 1994; Tobin et al., 1999; Lopez et al., 2005; Meinzer et al., 2008a; Zhu et al., 2013).*

We then proceed with our original summary (which has been reworded in some places) of various lines of evidence indicating how plant hydraulic traits can ultimately affect the distribution of species and their various hydraulic strategies across gradients in water availability (page 5, line 15 – page 6, line 8).

4. Page 7, lines 910 ": : : the model developed by J.S. Sperry and described in Sperry et al. (1998)" can be shortened as " : : : the model developed by Sperry et al. (1998)".

Done.

5. Page 7, line 21 "we modified S98 in three important ways : : :". I prefer to say "we modified S98 in three ways : : :" by crossing out "important". I understand that these modifications are important. But here it's a description of the model and you don't have to evaluate your works here.

We have eliminated value statements about our modeling approach from the Methods section.

6. Page 8 lines 112: This paragraph should be a summary of the model, not just what have been described in Supplement S1. The authors should expect the readers to get a picture of the model by their descriptions without reading S1.

We have substantially re-worked the model overview section (page 8, line 5 – page 9, line 9) to summarize the model for readers so that they do not have to read S1 to understand its structure.  We maintained references to the Supplement S1, however, so that readers can refer to it for further details not included in the main text:

*The fast-timescale dynamics of the hydraulics model are governed by three sets of constitutive relationships: 1) the relationship between water potential and water content, 2) the relationship between hydraulic conductivity and water potential, and 3) the relationship between a stomatal water stress multiplier and leaf water potential. The first two relations are applied to every compartment within the plant-soil continuum and have specific equations for plant and soil porous media types. The soil constitutive equations for the first two relations are given by, respectively, the van Genuchten (1980) and Mualem (1976) formulations. We chose these particular equations for the soil water characteristic and unsaturated hydraulic conductivity because extensive work has parameterized these formulations on tropical soils, which have been noted to have distinct hydraulic properties when compared to temperate soils of similar texture (Tomasella and Hodnett, 2002). These equations are given in the Supplement S1 Sections 3.1.2 and 3.2.2. The plant constitutive equations for the first two relations are formulated and described in Sections 2.1.2 and 2.1.3 below. The third relation for stomatal response to moisture stress is described in Section 2.1.5. All parameters of the constitutive relations for plant tissue are biologically interpretable and measureable plant hydraulic traits.*

*Several linkages are made between tree allometry and hydraulic properties (see Supplement S1 Section 4). Leaf, stem, transporting root, and absorbing root water storage compartment volumes derive respectively from the TFS-predicted leaf, stem, coarse root, and fine root biomasses using characteristic tissue densities. The heights of these components derive from tree height and rooting depth. The characteristic soil volume over which root uptake occurs is given by half the distance between absorbing roots, which decreases as total community root length (summed across all trees) increases. Total hydraulic conductance between adjacent plant water storage compartments is scaled from xylem hydraulic conductivity using first principles and plant allometric theory (Section 2.1.4). The model code only initializes these allometrically-dependent hydraulic properties; it does not (yet) implement functions to update them as trees grow. Neglecting the effects of growth has negligible effects on the results presented in this paper but will be necessary for application of the model at timescales longer than one year.*

*The numerical solution (see Supplement S1 Section 5) operates at every timestep and updates water contents and potentials throughout the plant-soil continuum (including root uptake or loss) due to transpiration. It uses a first-order Taylor series expansion about the water content term to linearize the Richards mass balance equation describing the 1-dimensional continuous array of plant and soil compartments. This results in a tridiagonal matrix that is solvable without iteration. Following the approach of Siqueira et al. (2008), infiltration and drainage are treated separately from the plant-soil fluxes due to transpiration (Supplement S1 Section 6).*

7. Page 8, lines 2223: It took me a while to think what the authors want to tell in this sentence. If the authors just want to talk about "capillary water", I prefer a sentence like "sapwood also stores capillary water in its void spaces and embolized conduits". Then, I don't have to think about "tension theory".

This sentence has been modified as suggested.

8. Page 8 lines 2330: These arguments are not necessary because this section is to describe the model. And, the sentences in lines 1423 can be reorganized so that it's easier for readers to understand Eqn 1.

We have eliminated value statements about our modeling approach from the Methods section. We deleted unnecessary details from the paragraph leading up to Eqn 1 and

added a sentence at the end to make it easier to understand Eqn 1. This paragraph now reads as:

*We used pressure-volume (PV) theory (Bartlett et al., 2012; Tyree and Hammel, 1972; Tyree and Yang, 1990) to describe the constitutive relation between total water potential ($\psi_{tot}$, MPa) and relative water content (RWC, g $H_2O$ $g^{-1}$ $H_2O$ at saturation) in the plant compartments (Eqn 1). $\psi_{tot}$ is the sum of two components: solute potential $\psi_{sol}$ (MPa) which is negative due to the presence of solutes in living cells, and pressure potential $\psi_p$ (MPa) which is $\geq 0$ due to cell wall turgor (but see Ding et al., 2014). PV theory is usually applied to leaves, but can also apply to sapwood (Chapotin et al., 2006; Meinzer et al., 2008b; Scholz et al., 2007); thus we apply it here to all plant tissue. Sapwood also stores capillary water (Tyree and Yang 1990) in its void spaces and embolized conduits. Consequently, this relation is described by three successive dehydration phases representing, respectively, capillary water (sapwood only), elastic cell drainage (positive turgor), and continued drainage after cells have lost turgor:*
*[equation 1]*

9. Page 9, line 6: "RWC" is not explained.

It is now explained.

10. Page 10, line 13 "(-)": Does it mean ax is negative? If yes, I prefer to use "-ax" in the equation.

"(-)" was changed to "(unitless)"

11. Page 11, lines 213: This paragraph is supposed to describe some "first principles" of the size effect of trees on plant hydraulics according to the second paragraph, but where are they? I saw "two main mechanisms". The second one describes two possibilities. Which one should be the "first principle" in the model?

The Section 2.1.5 including the paragraph to which you refer was substantially re-worked (page 11, line 19 – page 12, line 22) to make clear our scaling approach to the reader:

*Tree size exhibits a first-order control over much variation in whole-plant hydraulic conductance (Sperry et al., 2008) since hydraulic path length increases with tree height (H) (Mencuccini, 2002). For this reason, whole plant conductance is not a constant parameter in our model. Rather, first principles dictate that, to a first approximation, whole-tree maximum aboveground conductance ($K_{max,tree,ag}$; kg $s^{-1}$ $MPa^{-1}$) may be derived as a function of xylem conductivity ($k_{s,max,x}$; kg $m^{-1}$ $s^{-1}$ $MPa^{-1}$), sapwood area ($A_s$; $m^{-2}$), and H (m) as*

$$K_{max,tree,ag} = \frac{k_{s,max,x} A_s}{H} \qquad (5)$$

*This relation predicts the negative effects of H (increasing H means decreasing $K_{max,tree,ag}$). Whole-tree conductance per unit leaf area ($K_{l,max,tree,ag}$) will determine water status at the level of individual leaves, and thus hydraulic constraints on leaf-level gas exchange. Dividing through by leaf area ($A_l$; $m^2$) gives*

$$K_{l,max,tree,ag} = \frac{k_{s,max,x}A_s}{HA_l} \qquad\qquad (6)$$

*In this relation, the leaf:sapwood area ratio ($A_l : A_s$) emerges as a key plant trait controlling $K_{l,max,tree,ag}$. If $A_l : A_s$ decreases with tree height, as has been documented in many tree species (McDowell et al., 2002, but see Calvo-Alvarado et al., 2008), the negative effects of height can be partially overcome. In addition, the near-universal tendency for xylem conduits to increase in diameter within trees from stem tips to trunk base (referred to as xylem taper in the opposite direction) (Meinzer et al., 2010; Mencuccini et al., 2007; Olson et al., 2014; Olson and Rosell, 2013; Petit and Anfodillo, 2011) also mitigates the negative effects of height according to the Hagen-Poiseuille law. Neglecting the effects of xylem taper may thus overestimate the negative hydraulic effects of increasing path length in size-structured forests. Metabolic scaling theory (MST) makes baseline predictions about the optimal degree of xylem conduit taper in trees subject to the constraint of hydraulic safety (which decreases as conduits get larger) and has been validated against observations of conduit diameter across trees of different heights (Savage et al., 2010; West et al., 1999). We therefore use MST to include the effect of xylem taper on $K_{l,max,tree,ag}$ by modifying Equation 6 to include a xylem taper term ($\chi_{tap:notap,ag}$; unitless) representing the ratio of whole-plant conductance with taper to that without:*

$$K_{l,max,tree,ag} = \frac{k_{s,max,petiole}A_s}{HA_l}\chi_{tap:notap,ag} \qquad\qquad (7)$$

*where $k_{s,max,petiole}$ (kg m$^{-1}$ s$^{-1}$ MPa$^{-1}$) is used to reference $K_{l,max,tree,ag}$ to MST predictions. $\chi_{tap:notap,ag}$ is in the range of 23-50 for trees of heights 10-30 m; thus the benefit of xylem taper for increasing total plant conductance itself increases with tree height. $\chi_{tap:notap}$ is assumed constant across individuals in this study, but parameterizing variation in $A_l : A_s$ across species is an outcome of this study. The full details of this approach in addition to the treatment of the belowground component of tree conductance ($K_{max,tree,bg}$) are outlined in Section 2 of the Technical Description (Supplement S1).*

12. Page 11, line 10: "because of the Hagen-Poiseuille law". It's a phenomenon of the Hagen-Poiseuille law, not because of it.

This was changed to "according to"

13. Page 12, lines 2023: I think the sentence "FMCgs : : : is the only variable passed from the hydraulics module to the host model" is the message of this paragraph and therefore should be the first sentence.

We moved this sentence to be the first sentence.

14. Lines 2326 in Pages 12 and lines 15 in page 13: These sentences can be moved to discussion. This section is to describe the model, it's not necessary to argue these issues here.

Done – these sentences were moved to become the third paragraph in the 'Practical Implications' Section 5.3 of the Discussion (page 30, lines 17 – 27).

15. Page 13, line 28: Please also cite Strigul et al. 2008 for PPA.

Done.

16. Page 16, line 15: "Idealized model experiments". I would use "Model experiments" because any model experiments are always "idealized" somehow.

Done. Other references to "idealized model experiments" in the manuscript were also changed

17. Page 17, lines 1314 "We matched simulated trees : : :" It would be clearer if there is a table to show the settings of trees.

Great suggestion. We made a new Table 3 to this effect and now refer to it on page 18 line 5 and Figure 11 caption.  This new table is reproduced below:

**Table 3**. *Properties of simulated and observed trees given in Figure 11.*

| Figure 11 sub-panel | Tree ID* | Observed DBH (cm) | Simulated DBH (cm) | Simulated Canopy Layer | Simulated $A_l$:$A_s$ (m$^2$ cm$^{-2}$) | Simulated $K_{max,ag}$ (kg s$^{-1}$ MPa$^{-1}$) |
|---|---|---|---|---|---|---|
| a,b | C1 | 15.6 | 15.6 | 2 | 0.41 | 0.062 |
| c,d | C2 | 18.7 | 18.8** | 2, 3 | 0.45** | 0.064** |
| e,f | C4 | 43.9 | 43.9** | 1, 2 | 0.74** | 0.076** |
| g,h | C3 | 51.4 | 51.4 | 2 | 0.81 | 0.078 |

* as given in Fisher et al. (2006)

** two simulated trees were included in this size class. Value given is the average of the two trees.

18. Figures 11 and 13: explain "50% TFE" in legends.

Per the suggestion of Referee #1, we have removed the analyses related to TFE in this paper.

**References**

Fyllas, N. M., Gloor, E., Mercado, L. M., Sitch, S., Quesada, C. A., Domingues, T. F., Galbraith, D. R., Torre-Lezama, A., Vilanova, E., Ramírez-Angulo, H., Higuchi, N., Neill, D. A., Silveira, M., Ferreira, L., Aymard C, G. A., Malhi, Y., Phillips, O.

L., and Lloyd, J.: Analysing Amazonian forest productivity using a new individual and trait-based model (TFS v.1), Geoscientific Model Development, 7, 1251-1269, 2014.

**Linking hydraulic traits to tropical forest function in a size-structured and trait-driven model (TFS v.1-Hydro)**

Bradley O. Christoffersen[1,2], Manuel Gloor[3], Sophie Fauset[3], Nikolaos M. Fyllas[4], David R. Galbraith[3], Timothy R. Baker[3], Lucy Rowland[2], Rosie A. Fisher[5], Oliver J. Binks[2], Sanna A. Sevanto[1], Chonggang Xu[1], Steven Jansen[6], Brendan Choat[7], Maurizio Mencuccini[2,8], Nate G. McDowell[1] and Patrick Meir[2,9]

*Correspondence to*: bradley@lanl.gov, ph: +1-505-665-9118

1. Earth and Environmental Sciences, Los Alamos National Laboratory, Los Alamos, New Mexico, USA
2. School of GeoSciences, University of Edinburgh, Edinburgh, United Kingdom
3. School of Geography, University of Leeds, Leeds, United Kingdom
4. Department of Ecology and Systematics, University of Athens, Athens, Greece
5. Climate & Global Dynamics, National Center for Atmospheric Research, Boulder, Colorado, USA
6. Institute of Systematic Botany and Ecology, Ulm University, Ulm, Germany
7. University of Western Sydney, Hawkesbury Institute for the Environment, Richmond, New South Wales 2753, Australia
8. ICREA at CREAF, Cerdanyola del Vallès, Barcelona 08193, Spain
9. Research School of Biology, Australian National University, Canberra, Australia

**Abstract.** Forest ecosystem models based on heuristic water stress functions poorly predict tropical forest response to drought partly because they do not capture the diversity of hydraulic traits (including variation in tree size) observed in tropical forests. We developed a continuous porous media approach to model plant hydraulics in which all parameters of the constitutive equations are biologically-interpretable and measureable plant hydraulic traits (e.g., turgor loss point $\pi_{tlp}$, bulk elastic modulus $\varepsilon$, hydraulic capacitance $C_{ft}$, xylem hydraulic conductivity $k_{s,max}$, water potential at 50% loss of conductivity for both xylem ($P_{50,x}$) and stomata ($P_{50,gs}$), and the leaf:sapwood area ratio $A_l{:}A_s$). We embedded this plant hydraulics model within a 'trait forest simulator' (TFS) that model light environments of individual trees and their upper boundary condition (transpiration) as well as provid a means for parameterizing  variation in hydraulic traits among individuals. We synthesized literature and existing databases to parameterize all hydraulic traits as a function of stem and leaf traits including wood density (*WD*), leaf mass per area (*LMA*) and photosynthetic capacity (*A*~max~) and evaluated the coupled model's (called TFS v.1-Hydro)

predictions against observed diurnal and seasonal variability in stem and leaf water potential as well as stand-scaled sap flux.

Our hydraulic trait synthesis revealed coordination among leaf and xylem hydraulic traits and statistically significant relationships of most hydraulic traits with more easily measured plant traits.  Using the most informative empirical trait-trait relationships derived from this synthesis, the TFS-Hydro model parameterization is capable of representing patterns of coordination and trade-offs in hydraulic traits. TFS v.1-Hydro successfully captured individual variation in leaf and stem water potential due to increasing tree size and light environment, with model representation of hydraulic architecture and plant traits exerting primary and secondary controls, respectively, on the fidelity of model predictions.  The plant hydraulics model made substantial improvements to simulations of total ecosystem transpiration under control conditions, but the absence of a vertically stratified soil hydrology model precluded improvements to the simulation of drought response.  Remaining uncertainties and limitations of the trait paradigm for plant hydraulics modeling are highlighted.

Key words: vegetation modeling, plant hydraulics, size scaling, xylem embolism, capacitance, turgor loss point, water stress, soil-plant-atmosphere continuum

**1 Introduction**

Tropical forests harbor great biodiversity (Myers et al., 2000; ter Steege et al., 2013) and play an important role in regulating regional and global climate (Gash and Nobre, 1997; Silva Dias et al., 2002). However, climate change is inducing changes to the hydrological regime of tropical forests (Feng et al., 2013; Fu et al., 2013; Gloor et al., 2013), with some consensus for a projected increase in drought frequency over the coming century via an intensification of precipitation seasonality (Joetzjer et al., 2013; Boisier et al., 2015) an increase in El Niño events (Cai et al., 2014), and chronically rising atmospheric moisture demand (McDowell and Allen, 2015), even as the directional change of total precipitation remains highly uncertain (IPCC, 2007). Therefore, because of their intrinsic value and strong coupling to the regional and global climate system, it is of paramount importance to have a predictive capability of tropical forest response to changes in water availability (Kumagai and Porporato, 2012; Oliveira et al., 2014; Meir et al., 2015).

Evaluations of many coarse-scale forest ecosystem models in the species-rich tropics indicate a poor predictive ability of these models to simulate tropical forest response to drought  (Galbraith et al., 2010; Powell et al., 2013; Joetzjer et al., 2014; Rowland et al., 2015b). While most of these models treat soil water fluxes mechanistically, soil-root and internal plant water fluxes are not mechanistically modeled; consequently, dynamic changes in plant moisture stress in these models are driven by changes in soil water status alone (discussed in Feddes et al., 1978; Egea et al., 2011; Xu et al., 2013; Verhoef and Egea, 2014). Additionally, water stress in many models includes relatively few plant functional type (PFT)-dependent parameters; thus models are largely ignorant of plant traits  that may control response to moisture. In contrast, a mechanistic treatment of  plant hydraulics allows water stress to be driven by changes in leaf water status (Sperry and Love, 2015). Coupled with site-specific parameterization, the plant hydraulics approach enables high-fidelity simulation of tropical forest response to moisture (Fisher et al., 2006; Fisher et al., 2007).

While  a range of approaches exist for modeling plant hydraulics at fine scales (i.e., individual trees), all involve an extension of Darcy's law (Darcy, 1856) from the soil domain to include plants as well.  Darcy's law states that water flux anywhere in the soil-plant continuum is proportional to the product of soil or plant hydraulic conductivity and a gradient in water potential. In order for these models to capture drought response, hydraulic conductivity within the soil-plant continuum must dynamically respond to changes in moisture. Three main approaches are distinguished in terms of how they represent the impact of declines in water potential on tissue water content and xylem hydraulic conductivity . A first class of models is the simplest and  simulates moisture sensitivity of soil-root conductance but not  xylem hydraulic conductance (Jarvis et al., 1981; Williams et al., 1996; Ogée et al., 2003; Alton et al., 2009; Bonan et al., 2014).  This approach has proven useful for modeling the effects  experimental drought in tropical forests (Williams et al., 1998; Fisher et al., 2006; Fisher et al., 2007), but  it remains unclear whether this approach misattributes drought effects occurring within trees to the soil; therefore a second class of models  implemented variable xylem conductivity with xylem water potential (Williams et al., 2001; Hickler et al., 2006; Domec et al., 2012; Duursma and Medlyn, 2012; Xu et al., 2016).  To simplify computational load, these two approaches do not explicitly track dynamic changes in the volume of plant water storage. Instead, a constant ratio of change in stored water per unit change in water potential, or stem hydraulic capacitance, is assumed, which may overestimate the buffering capacity of tree stored water under extreme drought conditions when small relative declines in stored water induce very large declines in water potential. An additional consequence of the design of these models is the inability to represent the bidirectional flow of water at the root-soil interface.  Reverse flow of water from roots into soil is an important process in root hydraulic distribution (Oliveira et al., 2005), and may also mediate time to desiccation under drought (North and Nobel, 1997).

A convenient way to address these issues is in a third class of models (hereafter the "continuous porous media approach"), which simply extend the modeled mass balance of water from the soil domain into the plant by relating simulated changes in water content to water potential (and vice versa)

everywhere within the plant-soil continuum    (Edwards et al., 1986; Arbogast et al., 1993; Sperry et al., 1998;

5    Kumagai, 2001; Bohrer et al., 2005; Mackay et al., 2015; Mirfenderesgi et al., 2016).  While more computationally complex, the continuous porous media approach offers two main advantages in addition to addressing the issue of plant water storage and bidirectional root flow. First, the coupled plant-soil system is represented by a single mass balance equation, such that root water uptake or loss simply emerges from the solution of this equation, and needs not be

10   ascribed post-hoc as is the case in the first two approaches.  Second, this approach relies on an explicit description of the relationship between water content and water potential in plant xylem (the "pressure-volume", or PV curve), analogous to the water retention curves used in soil physics.  As we will show, there is a wealth of information on PV hydraulic traits for leaves, and to a lesser degree, stems in tropical forests.  Implementing PV curves in the model greatly

15   increases the scope of data with which the model can be parameterized.     In this paper, we develop a continuous porous media approach intended for application at specific sites in the tropics to explore dynamics of water

20   fluxes from hourly to seasonal timescales and at spatial scales ranging from individual trees to the stand-level scale. This intermediate-scale approach is a model testbed meant to inform implementations of plant hydraulics in coarse-scale forest ecosystem models.

Model parameterization leads us to the challenge of how to represent variation in plant hydraulic traits governing moisture sensitivity and water transport capacity.

25       It has long been recognized that the functional trait diversity of tropical forests mirrors their large species diversity   (Corner,

30   1949; Hallé et al., 1978; Leigh Jr, 1999), and diversity in   plant hydraulic traits such as the water potential at turgor loss ($\pi_{tlp}$) and at 50%

loss of conductivity ($P_{50}$), xylem-specific hydraulic conductivity ($k_{s,max}$) and the leaf-to-sapwood area ratio ($A_l : A_s$)cell turgor properties and capacitance, xylem conductivity, and vulnerability to embolism is no exception (Borchert, 1994; Tobin et al., 1999; Lopez et al., 2005; Meinzer et al., 2008a; Zhu et al., 2013)., Interspecific variation in these traits is thought to mediate which are implicated in the variation in species differences level in survival in both natural and experimentally-induced droughts (Nakagawa et al., 2000; Engelbrecht and Kursar, 2003; Kursar et al., 2009; Moser et al., 2014; Meir et al., 2015). Differential drought sensitivity, in turn, is known to explicitly link to species distributions (Choat et al., 2007; Engelbrecht et al., 2007; Baltzer et al., 2008; Condit et al., 2013). Mechanistic plant hydraulics also allows for diversity in tree height, already included in size-structured forest ecosystem models, to act as a hydraulic trait because it influences the path length over which water must traverse to reach the canopy. L Survival in these studies was not only taxon-specific, but also stratified by tree size, usually with larger trees often demonstrateing significantly higher vulnerability to drought (Nakagawa et al., 2000; Nepstad et al., 2007; da Costa et al., 2010; Rowland et al., 2015a), a trend demonstrated to be pantropical in scope (Phillips et al., 2010; Bennett et al., 2015). Such a trend highlights that, in addition to hydraulic functional traits, the increasing hydraulic path length associated with tall trees (and perhaps increased radiation loads) is an equally important determinant of drought sensitivity (McDowell and Allen, 2015).

Thus the implication of increases in drought frequency for tropical forests are shifts in growth form (Phillips et al., 2002; Schnitzer and Bongers, 2011), species composition (Fauset et al., 2012; Meir et al., 2015) as well as size distribution. The implication for ecosystem models is that under-representation of diversity in functional traits and tree size in tropical forests is hampering undermining efforts to make accurate projections of tropical forest response to climate. Model parameterization of hydraulic trait diversity should thus provide much-needed model capability to represent a diversity of responses to changes in moisture availability, laying the groundwork for representing trait-mediated differences in survival and subsequent shifts in forest trait composition. Shifts in trait composition are already occurring in some tropical forests (e.g., Enquist and Enquist, 2011; van der Sande et al., 2016), and such shifts (or the diversity of traits alone) have been shown to buffer ecosystems in the face of environmental change, and in some cases, are the difference between predicted complete loss of forest and forest persistence (Fauset et al., 2012; Levine et al., 2016; Sakschewski et al., 2016).

Because a complete representation of trait diversity in models is neither tractable nor desired, the challenge is to identify and represent the dominant dimensions of trait variation within our plant hydraulics model. To date, syntheses of hydraulic traits have

5 typically focused on a limited set of traits in isolation (Bartlett et al., 2012; Choat et al., 2012; Nardini et al., 2014; Anderegg, 2015) but see Mencuccini et al. (2015). In contrast, here we synthesize how a large suite of hydraulic traits represented in our model vary with better quantified dimensions of plant functional trait variation, such as the

10  leaf (Wright et al., 2004) andstem (Chave et al., 2009) economics spectra. We make no *a priori* assumptions about the coordination (or lack thereof) between leaf and stem economics, given the conflicting

15 evidence  (Baraloto et al., 2010; Mendez-Alonzo et al., 2012; Reich, 2014). In addition, we assess the evidence for trade-offs and coordination among hydraulic traits, independent of leaf and stem economics traits.

20 ~~economics spectra in tropical forests have been put forward; therefore given the lack of consistent evidence for their coordination, we make no *a priori* assumptions in this regard, and ask how hydraulic traits coordinate with either leaf or stem economics traits either in conjunction or separately. An additional (not necessarily independent) constraint on trait variation might emerge via trade-offs among certain hydraulic traits, such as the trade-off between xylem~~

25 ~~efficiency (hydraulic conductivity) and safety (embolism resistance) (Gleason et al., 2016; Sperry et al., 2008; Tyree et al., 1994) or the trade-off between drought avoidance (e.g., via leaf shedding) and tolerance (embolism resistance), mediated by sapwood capacitance (Borchert and Pockman, 2005; Meinzer et al., 2008b; Pineda-Garcia et al., 2013). Here we investigate the possibility of predicting hydraulic traits from more commonly measured leaf and stem traits.~~

30

In what follows,  we first describe the  continuous porous media hydraulics module that simulates the fast-timescale (hourly) processes of tree water flux and  water potential gradients throughout the soil-plant continuum . Second, via a pantropical synthesis of hydraulic traits, we derive empirical relationships for parameterizing hydraulic trait variation in the plant hydraulics model. . Third we  demonstrate the model's capability to simulate different individual-level functional responses to environmental variation (either water or light), arising from hydraulic limitations imposed by tree size or hydraulic traits. Finally, we evaluate the new model in terms of its ability to capture diurnal dynamics of water potential, individual-level variation in leaf water status, and the observed seasonal dynamics of ecosystem-level water use.

**2 Model Description: Plant Hydraulics and Host Individual Trait-Driven Forest Models**

The plant hydraulics model developed here is integrated into a "host" individual tree trait-driven forest model (TFS). TFS  simulates every tree in a stand > 10 cm DBH driven by plot observations of the tree size distribution, and each tree can possess a unique set of values of the four functional .  wood density (*WD*), leaf mass per area

(*LMA*), leaf nitrogen (*N*L,) and leaf phosphorus (*P*L).

5  The modification of transpiration fluxes in TFS by the plant hydraulics module can be summarized as follows (see Figure 1): The hydraulics module passes to TFS a nondimensional multiplier (*FMC*gs) (0,1] for each tree , which is based on each tree's leaf water potential at the previous timestep.  TFS  uses *FMC*gs  within its biophysics calculations to estimate

10 transpiration fluxes for each tree at the current timestep, and the transpiration of each tree is  passes back to the  hydraulics model, which computes changes in water potentials and water contents throughout the soil-plant continuum (including root water uptake or loss) due to transpiration. The plant hydraulics module also assigns hydraulic traits based on each tree's size and pre-assigned plant functional traits. Below we describe the

15 components of the plant hydraulics model (Section 2.1), and at the end of each subsection highlight the key hydraulic traits for which we seek to understand trait trade-offs and coordination via our empirical synthesis. Section 2.2 gives a description of the host TFS model . The Supplement S1 gives a full technical description of the  hydraulics module as it is

20 implemented within TFS, and Figure S1.1 can be referred to as the schematic of our hydraulics module .

**2.1 Plant Hydraulics Model**

**2.1.1 Overview**

In this section we highlight the important developments we made to the model developed by

25  Sperry et al. (1998; hereafter S98). S98 consists of a discretization of the soil-plant continuum as a series of water storage compartments with defined heights, volumes, conducting areas, water retention and conductivity properties, connected by elements with defined path lengths and conductances. Trees are divided into the four porous medium types of leaf, stem, transporting root and absorbing root, with the stem being divided into a

30 variable number of compartments and all other types consisting of a single compartment (Fig

S1.1).  The soil is radially discretized with a variable number of compartments, or cylindrical 'shells' around a characteristic absorbing root (the rhizosphere), and soil hydraulic properties are assumed constant across these compartments and across trees within a given soil type.  The present TFS v.1-Hydro scheme does not consider the vertical distribution of soil water or roots. We anticipate this to be a key this is left as a component for subsequent future model development when we incorporate this scheme into host models with variable soil depths.

The fast-timescale dynamics of the hydraulics model are governed by three sets of constitutive relationships: 1) the relationship between water potential and water content, 2) the relationship between hydraulic conductivity and water potential, and 3) the relationship between a stomatal water stress multiplier and leaf water potential. The first two relations are applied to every compartment within the plant-soil continuum and have specific equations for plant and soil porous media types. The soil constitutive equations for the first two relations are given by, respectively, the van Genuchten (1980) and Mualem (1976) formulations. We chose these particular equations for the soil water characteristic and unsaturated hydraulic conductivity because extensive work has parameterized these formulations on tropical soils, which have been noted to have distinct hydraulic properties when compared to temperate soils of similar texture (Tomasella and Hodnett, 2002).  These equations are given in the Supplement S1 Sections 3.1.2 and 3.2.2. The plant constitutive equations for the first two relations are formulated and described in Sections 2.1.2 and 2.1.3 below.  The third relation for stomatal response to moisture stress is described in Section 2.1.5. All parameters of the constitutive relations for plant tissue are biologically interpretable and measureable plant hydraulic traits.

Several linkages are made between tree allometry and hydraulic properties (see Supplement S1 Section 4). Leaf, stem, transporting root, and absorbing root water storage compartment volumes derive respectively from the TFS-predicted leaf, stem, coarse root, and fine root biomasses using characteristic tissue densities. The heights of these components derive from tree height and rooting depth. The characteristic soil volume over which root uptake occurs is given by half the distance between absorbing roots, which decreases as total community root length (summed across all trees) increases. Total hydraulic conductance between adjacent plant water storage compartments is scaled from xylem hydraulic conductivity using first principles and plant allometric theory (Section 2.1.4).  The model code only initializes these allometrically-dependent

hydraulic properties; it does not (yet) implement functions to update them as trees grow. Neglecting the effects of growth has negligible effects on the results presented in this paper but will be necessary for application of the model at timescales longer than one year.

The numerical solution (see Supplement S1 Section 5) operates at every timestep (hourly) and updates water contents and potentials throughout the plant-soil continuum (including root uptake or loss) due to transpiration. It uses a first-order Taylor series expansion about the water content term to linearize the Richards mass balance equation describing the 1-dimensional continuous array of plant and soil compartments. This results in a tridiagonal matrix that is solvable without iteration. Following the approach of Siqueira et al. (2008), infiltration and drainage are treated separately from the plant-soil fluxes due to transpiration (Supplement S1 Section 6). With reference to the sections below, we modified S98 in three important ways by: 1) incorporating continuous functions of varying water potential with water content in tree stem and leaf tissues (i.e., non-constant capacitance; Section 2.1.2); 2) using first principles to scale length-independent hydraulic conductivities to hydraulic conductance over the entire stem (Section 2.1.4); and 3) linking the hydrodynamics to stomatal conductance (Section 2.1.5), thus creating a fully prognostic and dynamic model of leaf water potential. The end result is a plant hydraulics model where the parameters of the constitutive equations (Eqns 1-5) are biologically interpretable and measureable plant hydraulic traits. We summarize these key plant-specific developments in the sections below.

With reference to the sections of the Technical Description in Supplement S1, a brief summary of the other components of the model is as follows: We used the van Genuchten (1980) and Mualem (1978) formulations for the soil water characteristic (Section 3.1.2) and unsaturated hydraulic conductivity curves (Section 3.2.2), respectively, which have been the basis for pedotransfer functions developed for tropical soils (Hodnett & Tomasella 2002). Hydraulic compartment heights and sizes (volumes and lengths, in the case of absorbing roots) are linked to the tree allometry of TFS (Section 4). The numerical solution uses a non-iterative mass-based solution to the Richards' equation, thereby doing away with the need for an iterative (potentially computationally intensive) Newton-Raphson scheme (Section 5). Following the approach of Siqueira et al. (2008), we separated out solutions for water movement in soil due to root water

**2.1.2 Tissue water relations**

We used pressure-volume (PV) theory (Tyree and Hammel, 1972; Tyree and Yang, 1990; Bartlett et al., 2012) to describe the constitutive relation between total water potential ($\psi_{tot}$, MPa) and relative water content ($RWC$, g H$_2$O g$^{-1}$ H$_2$O at saturation) in the plant compartments (Eqn 1). $\psi_{tot}$ is the sum of two components: solute potential $\psi_{sol}$ (MPa) which is negative due to the presence of solutes in living cells, and pressure potential $\psi_p$ (MPa) which is $\geq$ 0 due to cell wall turgor (but see Ding et al., 2014). PV theory  is usually applied to leaves, but can also apply to sapwood (Chapotin et al., 2006; Scholz et al., 2007; Meinzer et al., 2008b); thus  we apply it here to all plant tissue. Sapwood also stores capillary water (Tyree and Yang 1990) in its void spaces and embolized conduits . Consequently, this relation is described by three successive dehydration phases representing, respectively, capillary water (sapwood only), elastic cell drainage (positive turgor), and continued drainage after cells have lost turgor~~ Using PV theory as a formulation for tissue water relations in the model offers four main advantages: 1) it represents an explicit expression for the relation between water content and water potential amenable to numerical solution techniques for porous media water flow; 2) its parameters have a mechanistic interpretation; 3) it avoids the use of a constant capacitance with water potential (and thereby potentially infinite water supply); and 4) it allows for species differences in physiological parameters which may govern differential drought tolerance under extreme drought beyond the turgor loss point.~~

:

$$\psi_{tot} = \begin{cases} \psi_0 - m_{cap}(1 - RWC) & RWC_{ft} \leq RWC \leq 1 \\ \psi_{sol}(RWC) + \psi_p(RWC) & RWC_{tlp} \leq RWC < RWC_{ft} \\ \psi_{sol}(RWC) & RWC_r \leq RWC < RWC_{tlp} \end{cases} \qquad (1)$$

The first phase (capillary water)  is assumed linear, characterized by a slope ($m_{cap}$) and a saturated water potential ($\psi_0$). The second elastic drainage  phase has both solute ($\psi_{sol}$) and pressure ($\psi_p$) potential  changing from the relative water content at which elastic drainage begins ($RWC_{ft}$) up to the turgor loss point ($RWC_{tlp}$ and corresponding $\psi = \pi_{tlp}$), at which $\psi_p = 0$. In the final post-turgor loss phase, symplastic (cell water) and xylem water from embolized conduits are expressed up to the point at which $\psi_{tot}$ approaches $-\infty$ ($RWC_r$). $RWC_r$ is often referred to as the apoplastic fraction (Bartlett *et al.* 2012), but in light of the considerable amount of water released when vessels embolize in stems (Tyree et al., 1991; Holtta et al., 2009),  $RWC_r$ is best termed the residual fraction. Leaf PV curves as traditionally interpreted are a special case of Eqn (1) in which there is no capillary water ($RWC_{ft} = 1$). $\psi_{sol}$ and $\psi_p$ are respectively given by

$$\psi_{sol}(RWC) = \psi_{sol}(RWC^*) = \frac{-|\pi_o|}{RWC^*} \qquad (2)$$

$$\psi_p(RWC) = \psi_p(R^*) = |\pi_o| - \varepsilon R^* \qquad (3)$$

where $\pi_o$ (MPa) and $\varepsilon$ (MPa) are osmotic potential at full turgor and bulk elastic modulus, respectively, and $RWC^* = (RWC - RWC_r)/(RWC_{ft} - RWC_r)$ and $R^* = (RWC_{ft} - RWC)/(RWC_{ft} - RWC_r)$ are transformations representing $RWC$ and $R$ (relative water deficit; $1 - RWC$) of symplastic (cell) water only (Bartlett et al. 2012). The absolute mass ($W$; kg) of water in tissue is given by $W = \rho_w \theta_{sat} RWC$, where $\theta_{sat}$ (m³ m⁻³) is the maximum water content on a per volume basis (or porosity) and $\rho_w$ (kg m⁻³) is the density of water. The constitutive equations used in the model are in terms of volumetric water content ($\theta$; m³ H₂O m⁻³ plant tissue), achievable by using the transformations given above (see Supplement S1). Volumetric capacitance ($C$; kg m⁻³ MPa⁻¹) is defined at any point along the three regions as ($\rho_w \theta_{sat} \frac{dRWC}{d\psi}$), which for sapwood is highest in the capillary region (~200-400 kg m⁻³ MPa⁻¹), intermediate in the elastic region (20-200 kg m⁻³ MPa⁻¹) (Tyree and Yang 1990), and after an initial increase beyond the turgor loss point, in theory will approach zero as $RWC \rightarrow RWC_r$. Henceforth and in all figures, we report $C$ at full turgor ($C_{ft}$), defined as the change in water mass per unit volume per unit change in water potential over the region where cell turgor ($\psi_p$) $\geq$

$0$ ($C_{ft} \equiv \rho_w \theta_{sat} \frac{\Delta RWC}{\Delta \psi} \Big|_{RWC=1}^{RWC=RWC_{tlp}}$). For these hydraulic traits and all others which follow, we

denote specific reference to leaf and xylem (sapwood) tissue with the subscripts $l$ and $x$,

respectively. Key hydraulic PV traits which we seek to determine as functions of more

commonly measured plant traits are $\varepsilon$, $\pi_o$, $RWC_{tlp}$, $C_{ft}$, and $RWC_r$.

**5  2.1.3 Embolism vulnerability**

We use the inverse polynomial of (Manzoni et al., 2013a) for the xylem vulnerability curve,

termed here the 'fraction of maximum conductivity' for xylem ($FMC_x$):

$$FMC_x(\psi_x) = \left[ 1 + \left( \frac{\psi_x}{P_{50,x}} \right)^{a_x} \right]^{-1} \tag{4}$$

where $P_{50,x}$ is the water potential at which 50% of maximum conductance is lost and $a_x$ is a

shape parameter (-unitless). $FMC_x$ is defined at compartment nodes and is a nondimensional

10  multiplier bounded on (0,1] limiting the maximum xylem conductance ($K_{max,i}$; kg s⁻¹ MPa⁻¹;

$FMC_x$FMC = 1 and 0 indicate no and complete xylem embolism, respectively) between any two

compartments $i$ and $i$+1. (Section 1 of the Supplement outlines how $FMC_x$, defined at

compartment nodes, limits $K_{max,i}$, which is defined at compartment boundaries). Critically,

$K_{max,i}$ derives from plant hydraulic architecture and maximum xylem-specific conductivity

15  ($k_{s,max,x}$). Because $k_{s,max,x}$ relates to the maximum rate at which water can be transported

through the xylem, and $P_{50,x}$ quantifies the xylem water potential at which half of this transport

capacity is lost, these can be thought as representing xylem efficiency and safety, respectively.

"Safety" refers to a property of the xylem alone and is not to be confused with the hydraulic

safety margin (HSM), which is field-observed or modeled minimum leaf water potential relative

20  to $P_{50,x}$. As with PV traits, we seek to determine to what extent xylem efficiency and safety

covary with other plant traits and trade off with each other and/or covary with other plant traits.

**2.1.4 Scaling conductance with tree size**

Tree size exhibits a first-order control over much variation in whole-plant hydraulic conductance

(Sperry et al., 2008); since therefore, capturing this effect in any hydraulics scheme is critical for

25  modeling size-structured communities such as forests. The size effect is driven by an increase in

hydraulic path length increases with tree height (*H*)size (Mencuccini, 2002). For this reason,

whole plant conductance is not a constant parameter in our model. Rather, first principles dictate that, to a first approximation, whole-tree maximum aboveground conductance ($K_{max,tree,ag}$; kg s$^{-1}$ MPa$^{-1}$) may be derived as a function of xylem conductivity ($k_{s,max,x}$; kg m$^{-1}$ s$^{-1}$ MPa$^{-1}$), sapwood area ($A_s$; m$^{-2}$), and $H$ (m) as

$$K_{max,tree,ag} = \frac{k_{s,max,x}A_s}{H} \tag{5}$$

This relation predicts the negative effects of $H$ (increasing $H$ means decreasing $K_{max,tree,ag}$). Whole-tree conductance per unit leaf area ($K_{l,max,tree,ag}$) will determine water status at the level of individual leaves, and thus hydraulic constraints on leaf-level gas exchange. Dividing through by leaf area ($A_l$; m$^2$) gives

$$K_{l,max,tree,ag} = \frac{k_{s,max,x}A_s}{HA_l} \tag{6}$$

In this relation, the leaf:sapwood area ratio ($A_l:A_s$) emerges as a key plant trait controlling $K_{l,max,tree,ag}$. If $A_l:A_s$ decreases with tree height, as has been documented in many tree species (McDowell et al., 2002, but see Calvo-Alvarado et al., 2008), the negative effects of height can be partially overcome. In addition,  the near-universal tendency for xylem conduits to decrease in diameter within trees from  trunk base to stem tips  (Mencuccini et al., 2007; Meinzer et al., 2010; Petit and Anfodillo, 2011; Olson and Rosell, 2013; Olson et al., 2014)  also mitigates the negative effects of height according to  the Hagen-Poiseuille law. Neglecting the effects of xylem taper may thus overestimate the negative hydraulic effects of increasing path length in size-structured forests. Metabolic scaling theory (MST) makes baseline predictions about the optimal degree of xylem conduit taper in trees subject to the constraint of hydraulic safety (which decreases as conduits get larger) and has been validated against observations of conduit diameter across trees of different heights (West et al., 1999; Savage et al., 2010). We therefore use MST to include the effect of xylem taper on $K_{l,max,tree,ag}$ by modifying Equation 6 to include a xylem taper term ($\chi_{tap:notap,ag}$; unitless) representing the ratio of whole-plant conductance with taper to that without:

$$K_{l,max,tree,ag} = \frac{k_{s,max,petiole}A_s}{HA_l}\chi_{tap:notap,ag} \qquad (7)$$

where $k_{s,max,petiole}$ (kg m$^{-1}$ s$^{-1}$ MPa$^{-1}$) is used to reference $K_{l,max,tree,ag}$ to MST predictions.

5   ~~We used these first principles in the model to derive a bottom-up estimate of whole-tree maximum aboveground conductance ($K_{max,tree,ag}$; kg s$^{-1}$ MPa$^{-1}$) from xylem-specific conductivity ($k_{s,max,x}$; kg m$^{-1}$ s$^{-1}$ MPa$^{-1}$). We followed a three-step process, as follows: 1) $k_{s,max,x}$ was first standardized to the petiole ($k_{s,max,petiole}$; kg m$^{-1}$ s$^{-1}$ MPa$^{-1}$); 2) $k_{s,max,petiole}$ was then used to estimate whole-tree maximum aboveground conductance without xylem~~

10   ~~conduit taper ($K_{max,tree,notaper,ag}$; kg s$^{-1}$ MPa$^{-1}$); 3) $K_{max,tree,notaper,ag}$ was multiplied by a nondimensional factor representing the ratio of theoretical whole-tree aboveground conductance with taper to that without taper ($\chi_{tap:notap,ag}$) to derive $K_{max,tree,ag}$. The primary purpose of this component of the model is to capture the size dependency of $K_{max,tree,ag}$, without resorting to a pre-specified relationship between $K_{max,tree,ag}$ and tree size because we wanted to make~~

15    A taper  $\chi_{tap:notap,ag}$ is in the range of 23-50 for trees of heights 10-30 m; thus the benefit of xylem taper for increasing total plant conductance itself increases with tree height. $\chi_{tap:notap}$ is assumed constant across individuals in this study, but parameterizing

20   variation in $A_l$:$A_s$ across species is an outcome of this study. The full details of this approach in addition to the treatment of the belowground component of tree conductance ($K_{max,tree,bg}$) are outlined in Section 2 of the Technical Description (Supplement S1).

**2.1.5 Hydraulic impacts on stomatal conductance**

The fraction of maximum stomatal conductance ($FMC_{gs}$; [0,1]), which is updated at every

25   timestep and is used to down-regulate non-water stressed stomatal conductance

the host model TFS. Finally, we introduced to TFS a formulation for the impact of water supply on stomatal conductance ($g_s$), in which reductions in leaf water potential ($\psi_l$) cause a decline in $g_s$ (sensu Jarvis, 1976), is one of two variables passed from the hydraulics module to the host model TFS. As in Mencuccini et al. (2015), the formulation for $FMC_{gs}$ mimics . Similar to the approach for the loss of xylem conductivity with water potential relation (Eqn 4), this formulation followed a "fraction of maximum conductivity" ($FMC_{gs}$) curve:

$$g_s = g_{s,max} \cdot FMC_{gs} = g_{s,max} \left[ 1 + \left( \frac{\psi_l}{P_{50,gs}} \right)^{a_{gs}} \right]^{-1} \qquad (85)$$

where $P_{50,gs}$ and $a_{gs}$ respectively represent leaf water potential at 50% stomatal closure and the slope of the curve at $\psi_l = P_{50,gs}$ (Manzoni et al., 2013b; Mencuccini et al., 2015), and $g_{s,max}$ is stomatal conductance in the absence of water supply limitation and comes from the host model's stomatal conductance scheme (for TFS, we used Medlyn et al. (2011); see Eqn A25 of Fyllas et al. 2014). The second variable passed from the hydraulics module to the TFS model describes the derivative of $FMC_{gs}$ with leaf water content, which is derived from the $FMC_{gs} \sim \psi_l$ (Equation 8) and leaf pressure-volume (Equations 1-3) relations (see Equations A34a and A40 in the Supplement S1 for derivation). This variable is used as an additional constraint promoting stability within the numerical solution (see Section 5.3 of the Supplement S1 and Supplement S3 Figure S3.3).

-Theory and data suggest that stomata operate in such a way so as to prevent catastrophic xylem embolism (Sperry and Love, 2015); this implies that $P_{50,gs} > P_{50,x}$, which is supported by global datasets (Klein (2014) Manzoni et al. (2013b)). We used a 1:1 relationship between $P_{50,gs}$ and the water potential at 20% loss of xylem hydraulic conductivity ($P_{20,x}$), a relationship suggested which is supported by data from a tropical dry forest (Brodribb et al., 2003). $P_{20,x}$ is easily derived from $P_{50,x}$ and $a_x$ using Equation 4. This current approach ignores the continuum in hydraulic safety (Klein, 2014; Martinez-Vilalta et al., 2014; Skelton et al., 2015) and future work needs to identify how such a continuum maps onto other plant traits. $a_{gs}$ is derived from $P_{50,gs}$ using the same relationship derived for $a_x$ and $P_{50,x}$ (Table 2).

[revised manuscript text omitted]

15   tropical regions and added data from 15 additional species not originally present (Mendez-Alonzo *et al.* 2013).  We then evaluated the hypothesis that variation in hydraulic transport is explained by *WD* explored relationships of these traits with *WD* and $A_{max}$, by prioritizing *WD* reported in the original publications and adding species-average *WD* data from Zanne *et al.* (2009) where it was unreported.  We also evaluated the hypothesis that rates of hydraulic

20   transport coordinate with leaf-level rates of gas exchange.  Because the objective of this paper is to derive hydraulic traits from other non-hydraulic plant traits, we explored the relationship of $k_{s,max,x}$ regressed on the light-saturated photosynthesis rate ($A_{max}$), even though the causality of the relationship could be interpreted in reverse. There is substantial debate surrounding appropriate methods for determining embolism vulnerability (Choat et al., 2010; Cochard et al.,

25   2010; Sperry et al., 2012; Wang et al., 2014) (see Brodribb et al. *in press* for a brief summary). Because there is some consensus that the "gold-standard" for $P_{50,x}$ measurements involves bench dehydration (DH) on long stem segments (Jansen et al., 2015), we explored trait relationships with and without other measurement methods for $P_{50,x}$.

For hydraulic architecture, we used the only study of which we were aware for tropical trees

30   (Calvo-Alvarado et al., 2008) that reports independent measurements of individual total tree leaf

area and sapwood area across a wide range of tree sizes to explore the relationship of $A_l : A_s$ with tree height as well as with *LMA* and verified the latter relationship using a much broader dataset of branch-level measurements of $A_l : A_s$ conducted across the Amazon basin (Patiño et al., 2012). Via literature survey, we also compiled an independent, extensive dataset of $A_s$ as it varied with

5      tree diameter at breast height (*DBH*, cm) in tropical forests.

Finally, we standardized the representation of the "biome" category across all databases, and defined the following categories based on the location of the study (not the species' home range): tropical flooded forest (if identified as such in original publication), tropical wet forest (no months where evaporative demand exceeds precipitation), tropical moist forest (at least 1 month

10     where evaporative demand exceeds precipitation; predominantly evergreen), tropical dry forest (drought-deciduous phenologies make up a substantial fraction of species), tropical savanna (identified as such in original publication), subtropical forest (absolute latitude exceeded 23 degrees), tropical mangrove, and greenhouse.  Syntheses of traits were limited to studies conducted on species growing in native (non-greenhouse) environments in tropical (thus

15     excluding subtropical observations) upland (non-flooded) habitats.  In some cases, it was also necessary to match hydraulic traits for the same species given as multiple (different) records within a database.  As a rule, however, we did not average to the species level, and thus some variation included in regressions is intraspecific, albeit small.

**3.2 Model setup**

20     We used Princeton downscaled meteorological forcing data (Sheffield et al., 2006), observed soil textural properties, and observed tree size and trait distributions (for *WD*, *LMA*, $N_L$ and $P_L$) in the Caxiuana National Forest of east-central Brazilian Amazonia, a seasonal evergreen forest receiving 2100-2500 mm rainfall annually, and the site of an ongoing throughfall exclusion experiment (Meir et al., 2008; da Costa et al., 2010; Meir et al., 2015; Rowland et al., 2015a;

25     Rowland et al., 2015c), to parameterize and run our model for one year.  Spinup was not necessary, as the model was initialized from plot observations as described above, and an initial soil moisture was chosen typical of wet season conditions, which is when the model period began. While actual soil depth at the site is ~10 m, we set soil depth in simulations to 4m, which reflects the effective depth over which the majority of water extraction occurs, based on previous

30     model validation with soil moisture data (Fisher et al., 2007).  This setup applied to all model

simulations in this paper -- both the  model experiments and the simulations for comparison with field data.

**3.3 Model experiments**

[revised manuscript text omitted]

While the Calvo-Alvarado et al. (2008) study showed no significant common relationship of $A_l : A_s$ with *H* or *DBH* across all individuals studied, it identified increasing $A_l : A_s$ with *H within* 4 out of 5 individuals (Fig 6a). When we took species-average $A_l : A_s$ from the Calvo-Alvarado (2008) study and plotted the interspecific differences as a function of species-average *LMA*, the individuals largely fell on the same regression line as that of Patiño *et al.* (2012) (Fig. 6b). Therefore, it appears that for tropical forests leaf-level traits sets the range of $A_l : A_s$ about which variation occurs in response to height and/or light availability (Fig 6c).

**4.1.2 Plant hydraulics parameterization**

Figure 7 shows the parameterization of all components of the plant hydraulics model (combining Table 2 and plant hydraulics model constitutive equations 1-4) as a function of the TFS model input traits and tree height (*H*). Panels a-c demonstrate the distribution of leaf PV, sapwood PV and xylem vulnerability curves for a characteristic TFS model simulation in which each tree is assigned four trait values (*WD*, *LMA*, $N_L$ and $P_L$) according to the observed joint distribution (Taylor & Thompson 1986 resampling algorithm) of these four traits at the focal field site Caxiuana National Forest, which are used to parameterize each tree's constitutive equations (using Table 2), with each tree plotted as separate lines colored by trait values. This forest is representative of others in the region (Baraloto et al., 2010), showing decoupled leaf and stem economics (i.e., lack of any significant correlation between *WD* and any of the three leaf traits used as input for TFS). A consequence of this lack of leaf-stem trait coupling and our

formulation of $k_{l,max,x}$ as empirically derived primarily from leaf traits (Fig 5) while $P_{50,x}$ derives from stem traits (Fig 4a) is that the model input  prescribes only a weak  trade-off between $k_{l,max,x}$ and $P_{50,x}$ (Fig 7c), which is consistent with the data for this relationship (Fig 8a). In contrast, panels d-f present 'idealized' scenarios for a case in which  leaf and stem economics are tightly coupled (specifically, between *WD* and *LMA* or *WD* and $A_{max}$) over a wide range of input values. In this case, the consequence for the relationship between $k_{l,max,x}$ and $P_{50,x}$ is a perfect tradeoff (Fig 7f). Finally, we represented the dual dependency of $A_l : A_s$ on both *LMA* and tree size (Fig 6) as an $A_l : A_s \sim H$ relationship with an *LMA*-dependent slope (Fig 7g). For a prescribed leaf biomass allometry function (as $\sim DBH^2 H$; Lescure et al. 1980), this $A_l : A_s$ formulation predicted *LMA* as the driver of differences in sapwood area ($A_s$) $\sim$ DBH allometry, spanning the range of values given by our independent literature compilation across many different studies, though this relationship warrants further validation.

**4.1.3 Ability of hydraulics model parameterization to represent observed trade-offs and coordination among hydraulic traits**

Based on the pantropical xylem functional traits dataset, evidence for a trade-off between xylem efficiency (as $k_{s,max,x}$) and safety ($P_{50,x}$) was insignificant (p = 0.14 on log-transformed data), which was also the case when limiting the database to the DH method (p = 0.82; Fig. 8a). However, when limiting the dataset to tropical dry forests (not savanna), a significant trade-off between xylem efficiency and safety emerged (p < 0.01; $r^2 = 0.17$) which was even stronger (p < 0.0001; $r^2 = 0.44$) when considering two tropical dry forest studies (see circled points in Fig 8a (Markesteijn et al., 2011a; Markesteijn et al., 2011b; Mendez-Alonzo et al., 2012). Our plant hydraulics model parameterization is able to capture both the lack of a safety-efficiency tradeoff, which occurs when stem and leaf economics are decoupled (Fig 7c), and a strong safety-efficiency tradeoff when stem and leaf economics are tightly coupled (Fig 7f).

When we paired our sapwood PV data with data on xylem hydraulic safety, we found that $C_{ft,x}$ (xylem capacitance) and $P_{50,x}$ demonstrated significant evidence (p = 0.02) for a trade-off in drought avoidance (increasing capacitance) and drought tolerance (increasingly negative $P_{50,x}$) (Fig 8b). The strength of this relationship became marginally insignificant (p = 0.05) when

considering $P_{50,x}$ values obtained with the bench dehydration (DH) method only, but was unaffected by the correction factor that we applied to the sapwood PV curves, remaining significant (p = 0.02 and p = 0.01 for all data and DH-limited data, respectively; Supplement Fig. S2.4). Because $C_{ft,x}$ and $P_{50,x}$ both derive directly from *WD*, our plant hydraulics model parameterization also follows this avoidance-tolerance tradeoff (cf thin line Fig 8b).

Finally, joining of the leaf PV database with the xylem functional traits database demonstrated significant evidence in support of coordination between leaf and xylem drought tolerance, as given by significant relationships between leaf drought tolerance traits $\pi_{o,l}$ and associated turgor loss point $\pi_{tlp,l}$ with xylem drought tolerance traits $P_{50,x}$ and $P_{88,x}$, albeit with $R^2$ values no greater than 0.3 (Fig 8c-f). This cross-tissue coordination is also preserved by our model parameterization, with *WD* driving most of the variation in this coordination space and *LMA* generating residual variation in leaf drought tolerance (thin lines, Fig 8c-f).

**4.2 Model  experiments**

**4.2.1 Impact of plant hydraulics on size- and light-dependency of transpiration**

Figure 9 shows how mean +/- 1 s.d. diurnal cycles of transpiration change as a function of canopy position and wet/dry season for both TFS v.1 and TFS v.1-Hydro. The large increase in simulated transpiration in the dry season for both TFS v.1 and TFS v.1-Hydro (Fig. 9e-f) is driven by the comparatively  large  increase  in incoming solar radiation due to a reduction in cloud cover (Fig 9a-d) (Carswell, 2002; Fisher et al., 2007). At the level of individual tree crowns, canopy position is the dominant control over the amount of absorbed PAR (compare different colored lines in Fig 9a-d).  These trends remain when considering total tree transpiration (Fig 9i, j, k, l), but variation in crown size adds significant variability in total water fluxes, especially for canopy trees (very small trees can still be in the canopy depending on subplot assignment). The

effect of including plant hydraulics in TFS v.1-Hydro was to limit late morning and afternoon transpiration via the depletion of stored water within the canopy and tree stem, which caused midday declines in leaf water potential (see Fig 10) and induced hydraulic limitation to water flux

[revised manuscript text omitted]
  (Fig 13a, b), highlighting how plant hydraulics mediate the biosphere-atmosphere exchange of carbon and water. In sum, TFS v.1-Hydro represents a key and advanced model capability to represent differential performance of individual trees based on hydraulic traits, size and light environments (Figs 9-10). Future work coupling the present scheme with community dynamics (mortality, growth and recruitment) has the potential to predict shifts in community trait distributions under changing moisture regimes, as has been observed or implied in studies of tropical forest species distributions and community dynamics (Engelbrecht et al., 2007; Fauset et al., 2012).

[revised manuscript text omitted]
 Earth System Models (ESMs). Likely some degree of simplification of the present approach will be required upon implementation in ESMs; nonetheless we expect that inclusion of

10   trait-driven plant hydraulics schemes will lead to reduced uncertainty in the future state of tropical forests under climate change.

**Code Availability**

The JAVA source code for TFS v.1-Hydro can be obtained from the corresponding author upon request.

15   **Data Availability**

The Supplement S4 describes supplementary data for this paper. Leaf PV data (Figs 2 and 8) that were newly extracted from publications are available as .csv data files as described under the heading "Leaf PV database". Leaf PV data originating from pre-existing leaf PV databases are available by accessing the original articles cited under the heading "Leaf PV database."

20   Sapwood PV data (Figs 3, S2.1, and S2.3) are available as .csv data files as described under the heading "Sapwood PV database." Sapwood area data (Fig 7h) are available as .csv data files as described under the heading "Sapwood Area database." Xylem hydraulic trait data (Figs 4, 5, 8, and S2.2) that were newly extracted from publications are available as .csv data files as described under the heading "Xylem Functional Traits Database." Xylem hydraulic trait data

25   originating from the TRY archive can be accessed there (www.try-db.org); the references of data from that archive used in analyses here are given under the heading "Xylem Functional Traits Database."

**Acknowledgments**

This research was supported in part by the European Union Seventh Framework Programme,

30   under the project AMAZALERT (Grant Agreement No. 282664 to PM) by the Next-

Generation Ecosystem Experiments (NGEE-Tropics) project, which is funded by the Office of Biological and Environmental Research of the U.S. Department of Energy (DOE), and Los Alamos National Laboratory LDRD program. PM also acknowledges ARC support from FT110100457. This submission is under public release with the approved LA-UR-16-20338. We thank all colleagues who contributed data to the Xylem Functional Traits Database, as well as Rick Meinzer for sharing ancillary *LMA* data and Megan Bartlett for answering clarifying questions.  We additionally thank Brett Wolfe, Rafael Oliveira, Yadvinder Malhi, Dan Johnson and John Sperry for helpful discussions, Crystal Ng for suggesting the pithy term 'ecotransfer function,' and Mark Decker and Guo-Yue Niu for helpful discussions regarding the use of the mass-based solution to the Richards equation. We are grateful for the constructive criticisms of two anonymous referees that improved the overall clarity of this manuscript.

[revised manuscript text omitted]

30   **Table 3**. Properties of simulated and observed trees given in Figure 11.

| Figure 11 sub-panel | Tree ID* | Observed DBH (cm) | Simulated DBH (cm) | Simulated Canopy Layer | Simulated $A_l$:$A_s$ (m² cm⁻²) | Simulated $K_{max,ag}$ (kg s⁻¹ MPa⁻¹) |
|---|---|---|---|---|---|---|
| a,b | C1 | 15.6 | 15.6 | 2 | 0.41 | 0.062 |
| c,d | C2 | 18.7 | 18.8** | 2, 3 | 0.45** | 0.064** |
| e,f | C4 | 43.9 | 43.9** | 1, 2 | 0.74** | 0.076** |
| g,h | C3 | 51.4 | 51.4 | 2 | 0.81 | 0.078 |

31   * as given in Fisher et al. (2006)

32   ** two simulated trees were included in this size class. Value given is the average of the two
33   trees.

[Figure]

**Figure 1**. Overview of TFS v.1-Hydro model.  *Left sub-panel*: inputs are plot observations of the tree size distribution and distributions of each of four plant functional traits (*WD*, *LMA*, $N_L$, $P_L$). *Middle bottom*: the perfect plasticity approximation (PPA) orders trees by decreasing crown area. *Middle top*: TFS assigns physiological traits, such as $V_{c,max}$, to each tree based on functional traits. *Middle*: the PPA is used to estimate light environments of each tree, which influence fast timescale (hourly) biophysics of crown light interception, photosynthesis, and stomatal conductance. Assimilated carbon is allocated to leaves, stems, and fine roots daily. *Top right* (hydrodynamics): Soil-root-stem-canopy water fluxes interact with fast-timescale TFS biophysics by taking transpiration as a boundary condition and passing back the 'fraction of maximum conductance' (*FMC*$_{gs}$) for downregulating the next timestep's stomatal conductance based on leaf water potential. *Top right* (size- and trait-scaling): Hydraulic traits in leaves, stem, and roots are assigned based on each tree's height and trait values according to empirical equations and allometric theory described in this paper. *Bottom right*: TFS predicts a

[revised manuscript text omitted]